# ATG14 targets lipid droplets and acts as an autophagic receptor for syntaxin18-regulated lipid droplet turnover

Zhen Yuan[1,9], Kun Cai [2,9], Jiajia Li[3,9], Ruifeng Chen[1], Fuhai Zhang[1], Xuan Tan[1], Yaming Jiu [4], Haishuang Chang[5], Bing Hu [2], Weiyi Zhang[6] & Binbin Ding [1,7,8] ✉

Lipid droplets (LDs) are dynamic lipid storage organelles that can be degraded by autophagy machinery to release neutral lipids, a process called lipophagy. However, specific receptors and regulation mechanisms for lipophagy remain largely unknown. Here, we identify that ATG14, the core unit of the PI3KC3-C1 complex, also targets LD and acts as an autophagic receptor that facilitates LD degradation. A negative regulator, Syntaxin18 (STX18) binds ATG14, disrupting the ATG14-ATG8 family members interactions and subverting the PI3KC3-C1 complex formation. Knockdown of STX18 activates lipophagy dependent on ATG14 not only as the core unit of PI3KC3-C1 complex but also as the autophagic receptor, resulting in the degradation of LD-associated anti-viral protein Viperin. Furthermore, coronavirus M protein binds STX18 and subverts the STX18-ATG14 interaction to induce lipophagy and degrade Viperin, facilitating virus production. Altogether, our data provide a previously undescribed mechanism for additional roles of ATG14 in lipid metabolism and virus production.

Lipid droplets (LDs), surrounded by a single monolayer of phospholipids, are highly dynamic cellular organelles that are responsible for the storage of neutral lipids, such as triacylglycerols, cholesteryl esters, and retinyl esters[1]. LDs play important roles in multiple cellular processes, including membrane biogenesis, viral packing, and host defense[2–4]. There are two pathways for LD degradation. The canonical pathway for LD catabolism is lipolysis, which is catalyzed by three lipases[5]; the other is degraded by lipophagy, a selective autophagic mechanism in which LDs are engulfed by autophagosomes and

decomposed in lysosomes. Lipophagy includes the formation of pre-autophagosomal structures, recognition and engulfment of LDs by autophagosomes mediated by specific receptors, and fusion of autophagosomes with lysosomes for degradation. A significant body of knowledge has been compiled regarding the regulatory mechanisms of lipolysis; however, the mechanisms by which lipophagy is initiated remain largely unknown.

Lipophagy plays important roles in liver metabolism, virus production, neuroinflammation, cholesterol efflux, thermogenesis, and

[1]Department of Biochemistry and Molecular Biology, School of Basic Medicine, Tongji Medical College, Huazhong University of Science and Technology, Wuhan, Hubei 430030, China. [2]Institute of Health Inspection and Testing, Hubei Provincial Center for Disease Control and Prevention, Wuhan, Hubei 430079, China. [3]School of Pharmacy, Tongji Medical College, Huazhong University of Science and Technology, Wuhan, Hubei 430030, China. [4]Unit of Cell Biology and Imaging Study of Pathogen Host Interaction, The Center for Microbes, Development and Health, Key Laboratory of Molecular Virology and Immunology, Shanghai Institute of Immunity and Infection, Chinese Academy of Sciences, Shanghai 200031, China. [5]Shanghai Institute of Precision Medicine, Shanghai Ninth People's Hospital, Shanghai Jiaotong University School of Medicine, Shanghai, China. [6]Department of Applied Biology, College of Natural Resources and Life Science, Dong-A University, Busan 49315, Republic of Korea. [7]Cell Architecture Research Institute, Huazhong University of Science and Technology, Wuhan, Hubei, China. [8]Guangzhou National Laboratory; State Key Laboratory of Respiratory Disease, Guangzhou, Guangzhou, Guangdong 510000, China. [9]These authors contributed equally: Zhen Yuan, Kun Cai, Jiajia Li. ✉e-mail: dingbinbin1988@163.com

aging[6–11]. In hepatocytes, Rab10 recruits LC3-positive autophagic membrane to the surface of LDs by binding to EHBP1 and EHD2[12]. Rab7 regulates the transport of lysosomes to LDs, thereby thus activating lipophagy[13]. A recent study identified the key role of oxysterol-binding protein (OSBP)-related protein 8 (ORP8), a known ER lipid transfer protein, which localizes on LDs and acts as an autophagic receptor for lipophagy through interaction with ATG8 family members (ATG8s)[14]. However, it remains to be explored whether other LD-localized proteins can act as receptors to mediate lipophagy.

ATG14 is a core component of autophagic initiation complex class III Phosphatidylinositol 3-kinase complex 1 (PI3KC3-C1), which consists of VPS34, Beclin1, p150 and ATG14, and is responsible for the initiation of autophagy by generating phosphatidylinositol 3-phosphate (PI3P). The N-terminal cysteine repeats of ATG14 are essential for the ER localization and recruitment of the Beclin1-Vps34-p150 complex. The C-terminal BATS domain can sense membrane curvature and bind to membranes through an amphipathic alpha helix[15,16]. During final maturation, oligomeric ATG14 directly binds to the STX17–SNAP29 complex to promote membrane tethering and autophagosome-lysosome fusion[17]. In addition to binding to the autophagosome membrane, ATG14 has been implicated in localization to LDs, and its gain and loss results in a fall and increase in triglyceride levels in the liver and serum of mice[18,19]. However, the precise mechanism(s) by which ATG14 regulates LD dynamics remain poorly understood.

In this study, we showed that ATG14 targets LDs, interacts with ATG8s via LC3 interaction region (LIR), and acts as an autophagic receptor to regulate LD turnover via lipophagy. Our results also identified Syntaxin18 (STX18), which inhibits lipophagy by regulating the interaction of ATG14-ATG8s and the formation of PI3KC3-C1 complex. Coronavirus membrane protein M induces lipophagy via binding STX18 and subverting the STX18-ATG14 interaction, resulting in the degradation of the LD-associated anti-viral protein Viperin. Therefore, our data uncover a mechanism by which STX18 regulates ATG14-mediated lipophagy and highlight how coronavirus modulates lipophagy to degrade LD-associated anti-viral proteins.

## Results

### ATG14 targets LDs
To investigate whether ATG14 targets LDs, we first examined the colocalization of boron-dipyrromethene (BODIPY)-labeled LDs with mCherry-tagged ATG14 under steady state or during LD biogenesis induced by oleic acid (OA) treatment for 12 h via confocal microscopy. ATG14 was remarkably targeted to LDs upon OA treatment while failed to associate with LDs under steady stage (Fig. 1a). To confirm that the localization of ATG14 to LDs was not caused by the mCherry-tag at the N-Terminal, we also expressed ATG14 with a mCherry-tag at the C-Terminal (ATG14-mCherry) and found that ATG14-mCherry also showed LD localization (Fig. 1b). Colocalization of ATG14 and LDs was also observed in Vero-E6 cells (Fig. 1c). We further purified LD and found that ATG14 was enriched in LD fraction upon OA treatment (Fig. 1d). Importantly, using immunoelectron microscopy, we found that beside autophagosome localization, GFP-labeled ATG14 also decorated the membrane of LDs, but not GFP vector control (Fig. 1e and Supplementary Fig. 1a). ATG14 failed to colocalize with GPAT4[152–208] (Supplementary Fig. 1b), a membrane marker for nascent LDs under steady stage[20], but targeted to HSD17B11 (Supplementary Fig. 1c), a marker for late LDs under OA treatment[21]. We further analyzed the colocalization between ATG14 and other organelles with or without OA treatment. ATG14 had no obvious colocalization with mitochondrial and Golgi apparatus while partly associated with lysosomes and the ER with or without OA treatment (Supplementary Fig. 1d). Together, these data suggest that ATG14 targets LDs upon OA treatment.

ATG14 functions as a membrane curvature sensor to target pre-autophagic membrane structures through its BATS domain (final 80 amino acids, Barkor autophagosome-targeting sequence)[16]. Our observations showed that disruption of the ATG14 BATS domain (W484R/F485R/Y488R, ATG14[WFY-RRR] or deletion of last 10 amino acids, ATG14[△C10aa]) abolished its LD localization (Fig. 1f–h and Supplementary Fig. 1e), suggesting that LD targeting of ATG14 requires its BATS domain.

Four evolutionarily conserved cysteine repeats, C43/C46/C55/C58, are localized to the N-terminus of ATG14 and are essential for its ER localization, autophagy function, and homo-oligomerization[17,22]. The point mutant ATG14[C46A] failed to dimerize[17], but still showed LD localization in an imaging assay (Supplementary Fig. 1f), suggesting that LD targeting of ATG14 does not require homo-oligomerization.

### ATG14 overexpression induces lipophagy
ATG14 binds Beclin1 and forms the PI3KC3-C1 complex with VPS34 and p150, and plays an essential role in the initiation of autophagy[23]. Next, we sought to determine the functions of LD targeting by ATG14. Knockdown of ATG14 caused the accumulation of LDs in untreated cells (Fig. 1i). Overexpression of ATG14, but not ATG14[WFY-RRR], led to the decreased LD number and triglyceride storage in OA-treated cells (Fig. 1j–l). ATG14[WFY-RRR] was still functional in non-LD related tasks, as they interact with Beclin1 and STX17 and undergoes homo-oligomerization (Supplementary Fig. 2a, b), suggesting that ATG14 negatively regulates cellular LD content depending on its LD localization. We found that the reduction of LD number and triglyceride storage induced by ATG14 overexpression could be reversed by CQ or Bafilomycin A1 (Baf-A1) treatment (Fig. 1j–l), indicating that ATG14 overexpression induces lipophagy. To further explore the importance of LD localization in ATG14-mediated lipophagy, *Atg14* KO cells were generated (Supplementary Fig. 2c), and the LD-resident ATG14[WFY-RRR] was engineered through substitution of the ATG14 mutants signal peptide with the LD-targeting amphipathic helices (from SARS-CoV-2 ORF6 protein[24]) (LD-ATG14[WFY-RRR]) (Supplementary Fig. 2d). We first confirmed the colocalization between LD-ATG14[WFY-RRR] and LDs in OA and CQ co-treated cells (Supplementary Fig. 2e). LD-ATG14[WFY-RRR] rescued the decreased LD number in *Atg14* KO cells, but not ATG14[WFY-RRR] (Supplementary Fig. 2f). Collectively, these observations demonstrated that ATG14 targets LDs and induces lipophagy.

### ATG14 interacts with ATG8s and functions as a receptor
Next, we sought to determine the mechanism by which ATG14 induces lipophagy. Previous studies suggested that ATG14 interacts with LC3[25]. Therefore, we proposed that ATG14 may act as a receptor on LDs. Using in vitro GST pull-down assays and in vivo co-IP assays, we confirmed that ATG14 binds with LC3A, LC3C, GABARAP, and GABARAPL2, but not LC3B or GABARAPL1 (Fig. 2a, b), with a preference for LC3C in vivo. ATG14 was found to interact with endogenous LC3-II, but not LC3-I (Fig. 2c). We then investigated whether ATG14 is required for LD recruitment of LC3. Lipophagy is induced by serum starvation and we found that serum starvation enhanced the interaction between ATG14 and LC3C (Fig. 2d) while OA treatment had no effect on ATG14-LC3 interactions (Supplementary Fig. 2g). Knockdown of ATG14 abolished LD recruitment of LC3 upon serum starvation (Fig. 2e, f). These data suggest that ATG14 may interact with and recruit LC3 to LDs when lipophagy is activated.

A classic LIR, "WxxL", was identified on ATG14 (Fig. 2g). A point mutant, WL/AA (abbreviated as ATG14[LIRm]), was generated to break the LIR. ATG14[LIRm] neither interacted with LC3 (Fig. 2h, i and Supplementary Fig. 2h), nor promoted the conversion of endogenous LC3-I to LC3-II (Fig. 2j, k). ATG14[LIRm] still interacts with Beclin1 and STX17 and undergoes homo-oligomerization (Supplementary Fig. 2i, j), suggesting that the LIR of ATG14 is not required for the formation of PI3KC3-C1 complex and the fusion of autophagosomes with lysosomes. Given that ATG14 interacts with LC3-II and localizes on LDs, we sought to determine whether ATG14 could act as a receptor to mediate lipophagy. Upon serum starvation-induced lipophagy, ATG14 co-localized

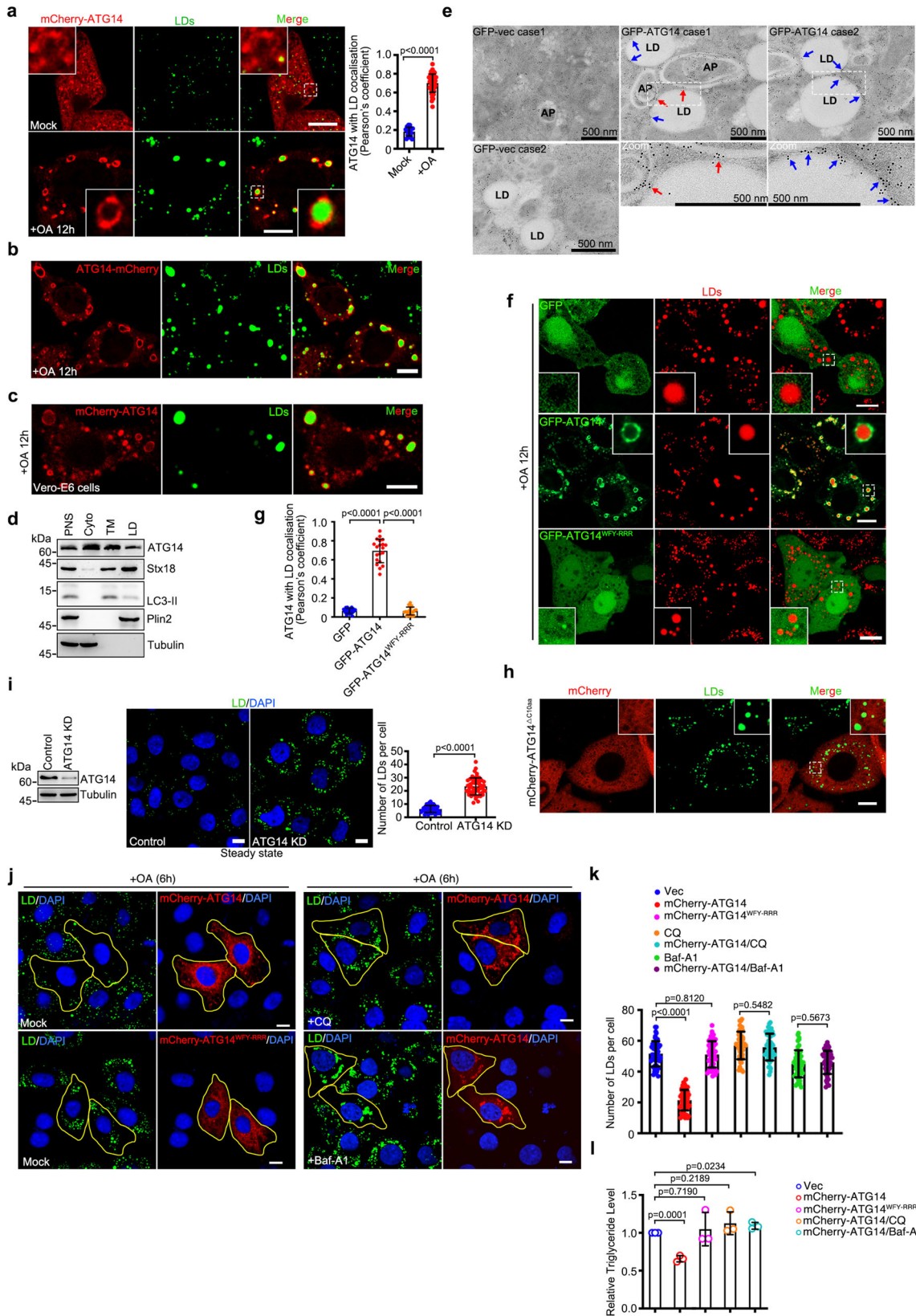

with LD and LC3 (Fig. 2l, m). Notably, ATG14^LIRm was still associated with LDs, but failed to recruit LC3 to LD (Fig. 2l, m). We further found that only wild-type ATG14 could rescue the decreased LD content and triglyceride storage, but not ATG14^LIRm in *Atg14* KO cells (Fig. 2n–p and Supplementary Fig. 2k). Therefore, these data indicate that ATG14 binds ATG8s via the LIR and acts as a receptor for lipophagy.

## STX18 negatively regulates the interactions between ATG14 and ATG8s

Next, we investigated the regulatory mechanism through which ATG14 functions in lipophagy. We focused on STX18, an ER-resident SNARE: (1) Based on siRNA screening to knockdown various SNARE proteins, STX18 knockdown induced autophagy (Supplementary Fig. 3a); (2)

**Fig. 1 | ATG14 targets LDs. a** Co-localization of LDs with mCherry-ATG14 in HeLa cells treated with or without 200 μM OA. Mock (17 cells) and OA-treated cells (51 cells) from three independent experiments were analyzed. **b** Co-localization of LDs with ATG14-mCherry in HeLa cells treated with 200 μM OA. **c** Co-localization of LDs with mCherry-ATG14 in Vero-E6 cells treated with 200 μM OA. **d** HeLa cells were treated with OA for 12 h. PNS, cytoplasm, total membrane and LDs fractions were isolated via ultracentrifugation and were analyzed via western blot. **e** Immunogold electron micrograph of GFP or GFP-ATG14 expressed HeLa cells. Blue arrows mark GFP-ATG14 dots are enriched on the surface of LDs. Red arrows mark GFP-ATG14 dots are enriched on the surface of autophagosomes (AP). **f, g** Co-localization of LDs with GFP, GFP-ATG14, or GFP-ATG14$^{WFY-RRR}$ in HeLa cells treated with 200 μM OA, $n = 20$ cells from three independent experiments were analyzed. **h** Co-localization of LDs with mCherry-ATG14$^{\triangle C10aa}$ in HeLa cells treated with 200 μM OA.

**i** HeLa cells were transfected with or without ATG14 siRNA for 48 h and analyzed via western blot. For imaging assay, LDs were labeled with BODIPY-493/503. The nuclei were stained with DAPI. Number of LDs in each cell was counted from 50 cells of three independent experiments. **j, k** HeLa cells expressing mCherry-ATG14 or its mutant were treated with 200 μM OA. Cells were also treated with 100 μM CQ for 6 h or 400 nM Baf-A1 for 4 h. LDs were labeled with BODIPY-493/503. The nuclei were stained with DAPI. Yellow ROIs indicate cells expressing ATG14. Number of LDs was counted from 50 cells of three independent experiments. **l** The concentration of triglyceride in HeLa cells of three independent experiments was analyzed. Data information: Scar bar represents 500 nm in **e** and represents 10 μm for all fluorescence images. Error bars, mean ± SD of three independent experiments. Two-tailed Unpaired Student's $t$ test. Source data are provided as a Source Data file.

---

STX18 knockdown enhanced PI3P production (Fig. 3a); (3) consistent with previous study[26], STX18 was found to associate with LDs (Supplementary Fig. 3b, c); (4) previous report has shown that the depletion of STX18 resulted in a reduced number of LDs, as well as decreased cellular triglyceride level[26]; and (5) using a co-IP assay, we found that ATG14 interacts with STX18, and this interaction was decreased under serum starvation (Fig. 3b). The direct interaction between STX18 and ATG14 was further confirmed using an in vitro GST pull-down assay (Fig. 3c). OA or CQ treatment, or knockdown of Atg7 had minimal effect on the endogenous interaction between STX18 and ATG14 (Supplementary Fig. 3d, e).

Next, to further explore the importance of STX18 in ATG14-mediated lipophagy, *Stx18* KO cells were generated and high autophagy level was observed (Supplementary Fig. 3f). The colocalization between ATG14 and LD was confirmed in *Stx18* KO cells (Supplementary Fig. 3g), indicating that STX18 deletion caused no effect on ATG14-LD colocalization. To investigate whether STX18 KD has any effect on LD synthesis/expansion, we co-treated cells with ATGL siRNAs and CQ to block LD degradation (lipolysis and lipophagy), and found that STX18 KD caused minor effect on the amount of LDs in CQ-treated ATGL KD cells (Supplementary Fig. 3h). Furthermore, LiveDrop is a widely used probe to label nascent lipid droplets. The results showed that STX18 KD had no effect on the amount of LiveDrop puncta (Supplementary Fig. 3i). Together, these data indicate that loss of STX18 has no direct impact on LD synthesis/expansion.

STX18 overexpression remarkably subverted the interactions between ATG14 and LC3A, LC3C, GABARAP, or GABARAPL2 (Fig. 3d, e, and Supplementary Fig. 4a−c). Consistently, knockdown of STX18 enhanced the interaction between ATG14 and LC3C (Fig. 3f). In addition, obvious colocalization between ATG14-LC3-LDs but not ATG14$^{LIRm}$-LC3-LDs was observed in *Stx18* KO HeLa cells (Fig. 3g, h). Together, these data suggest that STX18 disrupts the interactions between ATG14 and ATG8s.

**STX18 negatively regulates the formation of PI3KC3-C1 complex**
From our in vivo co-IP assay, we found that STX18 also interacts with Beclin1, Vps34, and UVRAG, but not LC3 (Fig. 3i and Supplementary Fig. 4d), suggesting that STX18 binds to both PI3KC3-C1 and PI3KC3-C2 complexes. We then investigated whether STX18 regulated the formation of PI3KC3 complexes. STX18 overexpression remarkably reduced the interaction between ATG14-Beclin1-Vps34 (Fig. 3j and Supplementary Fig. 4e) while having little if any effect on the interactions of UVRAG with Vps34 or Beclin1 (Supplementary Fig. 4f), suggesting that only PI3KC3-C1, but not PI3KC3-C2 formation is disrupted by STX18. Additionally, we purified PI3KC3-C1 complexes from HEK293F cells and confirmed that STX18 overexpression disrupted the formation of PI3KC3-C1 complexes in vitro (Fig. 3k). Furthermore, compared to wild-type cells, stronger interactions between ATG14-Beclin1-Vps34 were observed in *Stx18* KO cells (Fig. 3l and Supplementary Fig. 4g). Thus, STX18 inhibits the formation of PI3KC3-C1 complex.

Next, we sought to determine whether STX18 knockdown promoted LD-associated PI3P production. We employed a fluorescence-labeled PI3P sensor to track PI3P. Compared to control cells, the number of fluorescent PI3P was significantly increased, and more PI3P was attached to the LDs in STX18-depleted cells (Fig. 3m−o). Instead, knockdown of ATG14 abolished STX18 depletion-induced the production and LD association of PI3P (Fig. 3m−o). Together, these data indicate that STX18 acts as a negative regulator of PI3KC3-C1 complex and regulates LD-associated PI3P production dependent on ATG14.

Interestingly, overexpression of ATG14 recruited Beclin1 to LDs, and a remarkable colocalization was observed between ATG14 and Beclin1 on LDs (Supplementary Fig. 4h). Additionally, knockdown of ATG14 abolished the STX18 depletion-induced LD association of Beclin1 (Supplementary Fig. 4i), and STX18 depletion failed to induce LD-associated PI3P production in Beclin1 knockdown cells (Fig. 3o and Supplementary Fig. 4j). These data suggest that ATG14 recruits Beclin1 to LDs for LD-associated PI3P production in STX18 knockdown cells.

Next, we sought to determine whether STX18 negatively regulated the PI3KC3-C1 complex by directly interacting with ATG14. To map the critical region of STX18 necessary for its interaction with ATG14, a series of progressively truncated STX18 mutants were constructed and used for co-IP assay. We found that the mutant STX18$^{\triangle1-80}$-Flag failed to coimmunoprecipitate with ATG14 (Supplementary Fig. 5a). Further truncation assays based on STX18 showed that residues 71-80 were required for its interaction with ATG14 (Supplementary Fig. 5b). STX18$^{\triangle71-80}$ failed to inhibit the interactions of PI3KC3-C1 complex (Supplementary Fig. 5c, d), and consequently, was defective in subverting the ATG14 expression-induced conversion of endogenous LC3-I to LC3-II (Supplementary Fig. 5e). Consistent with previous study[23], ATG14-Beclin1 interaction is mediated by their coiled-coil domain (CCD) (Supplementary Fig. 5f). Notably, we found that STX18 directly binds the CCD of ATG14 (Supplementary Fig. 5g, h). To confirm that STX18 disrupts the formation of PI3KC3-C1 complex by competitively binding the CCD in ATG14, we gradually increased the expression of ATG14 and found that STX18 overexpression-inhibited Torin1-induced p62 degradation was reversed (Supplementary Fig. 5i). Taken together, our data indicate that STX18 competitively binds the CCD in ATG14 via its residues 71-80 to inhibit ATG14-Beclin1 interactions, thus disrupting the formation of PI3KC3-C1 complex, inhibiting autophagy initiation.

Next, we investigated whether STX18 regulated the function of ATG14 in the fusion of autophagosomes with lysosomes. Overexpression of STX18 had a minimal effect on the interaction between ATG14 and STX17 (Supplementary Fig. 5j). Knockdown of STX18 induced autophagic flux (increased LC3-II and decreased p62) under both stressed and unstressed conditions (Supplementary Fig. 5k). The degradation of p62 induced by STX18 depletion was only observed in Atg5 WT MEF cells, but not in *Atg5* KO cells (Supplementary Fig. 5l) or *Atg14* KO cells (Supplementary Fig. 5m), indicating that STX18 depletion activates autophagic flux. Together, these data suggest that STX18 is not involved in ATG14-mediated autophagosome-lysosome fusion.

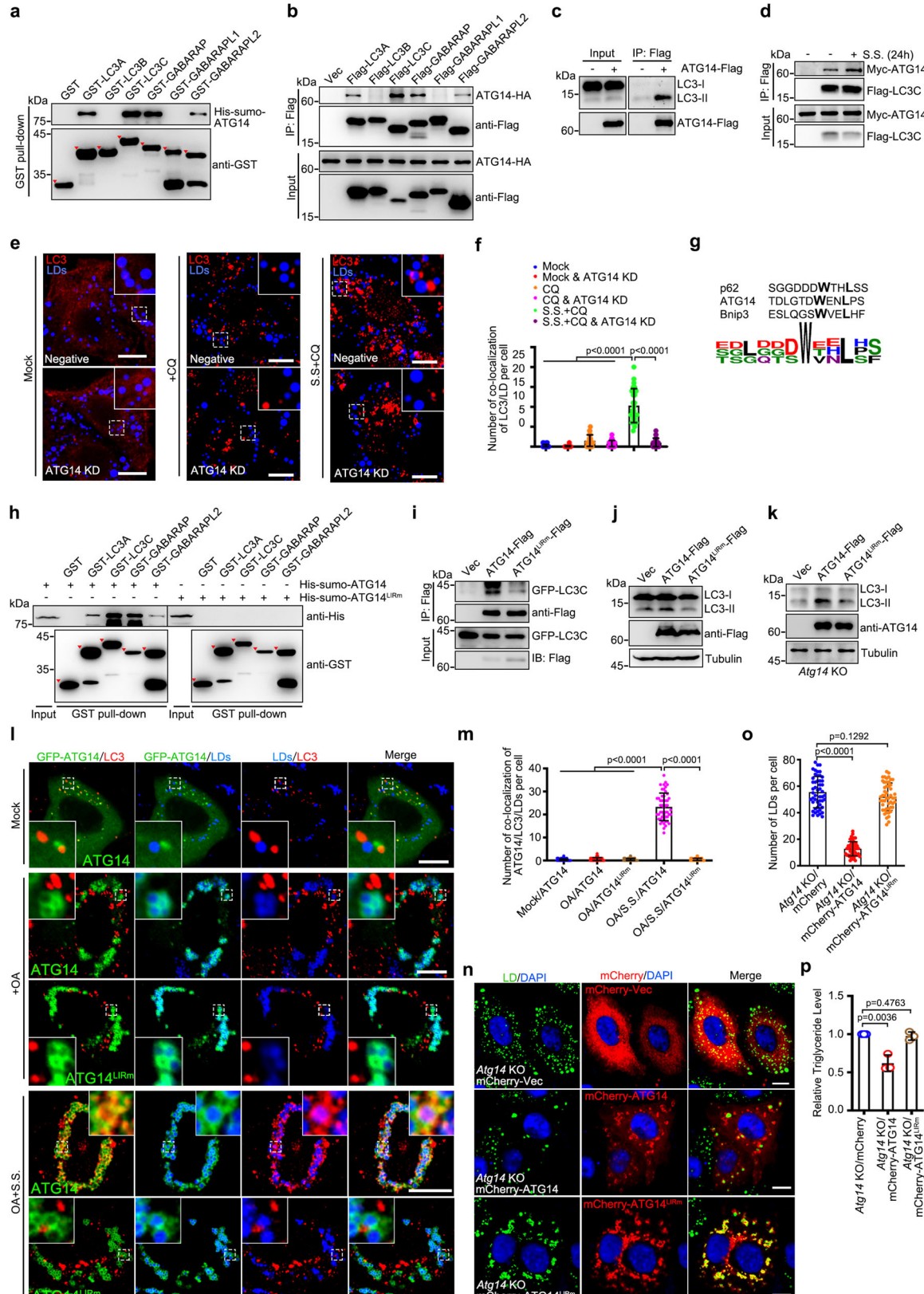

**Depletion of STX18 activates lipophagy dependent on ATG14**

Although previous study has shown that STX18 does not colocalize with LC3 even under starvation conditions[27], given that ATG14 targets LDs, and knockdown of STX18 enhances ATG14-mediated production of PI3P and recruitment of LC3 to LDs, we hypothesized that knockdown of STX18 may induce lipophagy dependent on ATG14.

Consistently, STX18 depletion caused a reduction in LDs number but failed to induce LD degradation in Atg7 knockdown HeLa cells (Fig. 4a). We also observed reduced triglyceride levels in STX18 depletion cells, and this change was dependent on autophagy (Fig. 4b). LDs were decorated with autophagosome/autolysosome marker LC3 in CQ-treated STX18 knockdown cells (Fig. 4c). Using transmission electron

**Fig. 2 | ATG14 acts as a lipophagic receptor. a** GST pull-down analysis of GST-ATG8s with His-sumo-ATG14. **b** Co-immunoprecipitation analysis of Flag-ATG8s with ATG14-HA. **c** Co-immunoprecipitation analysis of ATG14-Flag with endogenous LC3. **d** The effect of serum starvation (S.S.) on Myc-ATG14 and Flag-LC3C interaction was analyzed by co-immunoprecipitation. **e, f** Co-localization of LDs with endogenous LC3 in indicated treated cells. Wile-type or ATG14 KD HeLa cells were treated with 200 µM OA for 12 h, and meanwhile, with or without S.S. for 24 h, with or without 100 µM CQ for 6 h before cells were fixed. The number of colocalization of LC3 with LDs per cell was counted from 25 cells of three independent experiments. **g** Schematic diagram of LIR (WxxL) in p62, ATG14, and Bnip3. **h** GST pull-down analysis of GST-ATG8s with His-sumo-ATG14 or LIR mutant. **i** Co-immunoprecipitation analysis of ATG14-Flag or LIR mutant with GFP-LC3C. **j, k** HEK293T cells were transfected with indicated plasmids for 36 h (**j**), *Atg14* knockout HeLa cells were transfected with indicated plasmids for 24 h (**k**). Cell

lysates were analyzed via western blot. **l, m** Co-localization of LDs, GFP-ATG14, and endogenous LC3. HeLa cells were treated with or without 200 µM OA for 12 h and treated with or without S.S. for 24 h. Meanwhile, 100 µM CQ was added for 6 h before cells were fixed. The number of colocalization of ATG14/LC3/LDs per cell was counted from 50 cells of three independent experiments. **n, o** *Atg14* knockout HeLa cells were transfected with indicated plasmids for 24 h, and treated with 200 µM OA for 6 h. Cells were imaged by confocal microscopy. Number of LDs was counted from 50 cells in three independent experiments. **p** The concentration of triglyceride in *Atg14* knockout HeLa cells of three independent experiments was analyzed according to manufacturer's instructions. Data information: Scar bar represents 10 µm for all fluorescence images. Error bars, mean ± SD of three independent experiments. Two-tailed Unpaired Student's *t* test. Source data are provided as a Source Data file.

microscopy (TEM), we directly visualized "microlipophagy" in which LDs were engulfed into single-membrane autolysosomes in *Stx18* KO HepG2 and STX18 knockdown HeLa cells (Fig. 4d–g). All these evidences support the idea that lipophagy is activated upon STX18 depletion.

Moreover, ATG14 knockdown abolished STX18 depletion-induced LD degradation (Fig. 4h, i). We further verified the lipophagy by applying the RFP-GFP tandem tagging strategy, RFP-GFP-Plin2[12]. The presence of the RFP-GFP-Plin2 reporter enhanced LD levels in these cells. Lipophagic degradation, indicated by the number of RFP + GFP− LDs (the GFP of this tandem reporter is attenuated by lysosomal degradation), was observed in STX18 knockdown cells in comparison to wild-type cells, and knockdown of ATG14 inhibited GFP degradation (Fig. 4j, k). Additionally, unlike the wild-type, STX18$^{\triangle 71\text{-}80}$ (defect in ATG14 interaction) failed to rescue the reductions in LD number and triglyceride storage in *Stx18* KO cells (Fig. 4l–n). These results indicate that knockdown of STX18 induces lipophagy dependent on ATG14 interaction. Notably, ATG14$^{LIRm}$ (an LIR mutant) failed to rescue the degradation of LDs caused by STX18 depletion in *Atg14* KO cells (Fig. 4o, p). Collectively, these results indicate that STX18 regulates lipophagy, which is dependent on the autophagic receptor ATG14.

Besides ATG14, STX18 regulates lipophagy also dependent on Beclin1 (Supplementary Fig. 6a). Five key autophagy receptors KO (Penta KO) cells that lacked p62/SQSTM1, NBR1, NDP52, OPTN, and TAX1BP1 were used to determine whether STX18 depletion induces lipophagy dependent on multiple reported autophagic receptors[28]. Knockdown of STX18 induced the degradation of LDs in Penta KO cells (Supplementary Fig. 6b). Chaperone-mediated autophagy (CMA) was also reported to regulate the degradation of LD-coated proteins (perilipin 2 and perilipin 3) and subsequently precede lipophagy[29]. Depletion of HSC70, the key chaperone receptor in CMA, did not compromise the STX18 depletion-induced LD degradation (Supplementary Fig. 6c), suggesting that STX18 acts independently of multiple autophagic receptors and CMA. We further confirmed that STX18 depletion failed to induce LD degradation in *Fip200* (the core unit of ULK complex) or *Atg5* (the core unit of ATG16 complex) KO MEF cells (Supplementary Fig. 6d), indicating that STX18 regulates lipophagy dependent on ULK complex and ATG16 complex. STX18 was reported as an ER-resident t-SNARE and also as a component of tethering complex between ER and LDs[26,30]. We further used a reporter construct, mCherry-Sec61B, an ER sheet resident protein that produces free mCherry visualized by western blot analysis when ER-phagy (autophagic degradation of ER) was induced because of the partial digestion of Sec61B, to determine the effect of STX18 depletion on ER-phagy. Knockdown of STX18 caused no free mCherry production (Supplementary Fig. 6e), nor degradation of ER membrane proteins CLIMP63 or REEP5 (Supplementary Fig. 6f). Thus, we conclude that STX18 is not involved in ER-phagy.

## STX18 depletion induces Viperin degradation through ATG14-mediated lipophagy

Interestingly, we found that the protein level of the interferon-induced protein Viperin (also known as cig5 and RSAD2), which is LD-associated protein that have been identified to play critical roles in anti-infection[4], was significantly decreased in STX18 knockdown cells upon OA treatment (Fig. 5a). Furthermore, we found that IGTP, TGTP1, and IFI47, which are LD-associated anti-infection proteins were all decreased in STX18 knockdown cells (Supplementary Fig. 7a). Having established that STX18 depletion can induce lipophagy, resulting in LDs and LD-associated proteins being sequestered by autophagosomes, we thought to determine whether STX18 depletion or ATG14 overexpression could degrade LD-associated immune proteins to reduce the anti-viral response. Viperin is active against two viruses that assembled on LDs (HCV and DENV)[31]. We focused on Viperin. We first confirmed that Viperin was partly targeted to LDs upon OA treatment (Supplementary Fig. 7b). Consistently, STX18 depletion-induced degradation of Viperin-HA upon OA treatment, which can be inhibited by CQ treatment, whereas it had minimal effect on Viperin-HA degradation without OA treatment (Fig. 5a). Colocalization of LC3-Viperin-LDs was observed in STX18 depletion cells (Supplementary Fig. 7c). Furthermore, rescued with wild-type STX18, but not STX18$^{\triangle 71\text{-}80}$ (defect in ATG14 interaction) abolished the colocalization of LC3-Viperin-LDs in *Stx18* KO cells (Supplementary Fig. 7d). In addition, obvious colocalization between LC3-Viperin-LDs was observed in ATG14-Flag expressed, but not vector or ATG14$^{LIRm}$-Flag expressed *Atg14* KO cells (Supplementary Fig. 7e). Endogenous Viperin induced by VSV infection was also degraded upon STX18 depletion, and this reduction was reversed by Atg7 KD (Fig. 5b). Overexpression of STX18 resulted in the accumulation of Viperin-GFP (Supplementary Fig. 7f). Similar to STX18 depletion, overexpression of ATG14 also promoted the degradation of Viperin-GFP under OA treatment but not under mock treatment (Fig. 5c). Instead, ATG14$^{LIRm}$ had little effect on the protein levels of Viperin-GFP (Fig. 5d). In *Atg14* KO cells, unlike the wild-type, ATG14$^{LIRm}$ failed to rescue the degradation of Viperin-GFP (Fig. 5e). Endogenous Viperin induced by VSV infection was also degraded upon ATG14 overexpression, and this reduction was reversed by CQ treatment or Atg7 KD (Fig. 5f). Additionally, neither STX18 depletion, nor ATG14 overexpression promoted the degradation of mutant Viperin$^{\triangle 1\text{-}42}$ (Fig. 5g–i), which is defective in LD localization[4,32]. Under STX18 depletion or ATG14 overexpression conditions, compared to wild-type Viperin, stronger degradation was observed for mutant ALDI-Viperin (Fig. 5j–l), which contains a LD-targeting sequence ALDI and showed more LD localization[4]. Therefore, these data indicate that STX18 depletion or ATG14 overexpression induces the degradation of Viperin via lipophagy. Furthermore, in *Stx18* KO cells, STX18$^{\triangle 71\text{-}80}$-Flag was defective in reversing the degradation of Viperin-GFP (Fig. 5m), and ATG14 knockdown abolished STX18 depletion-induced

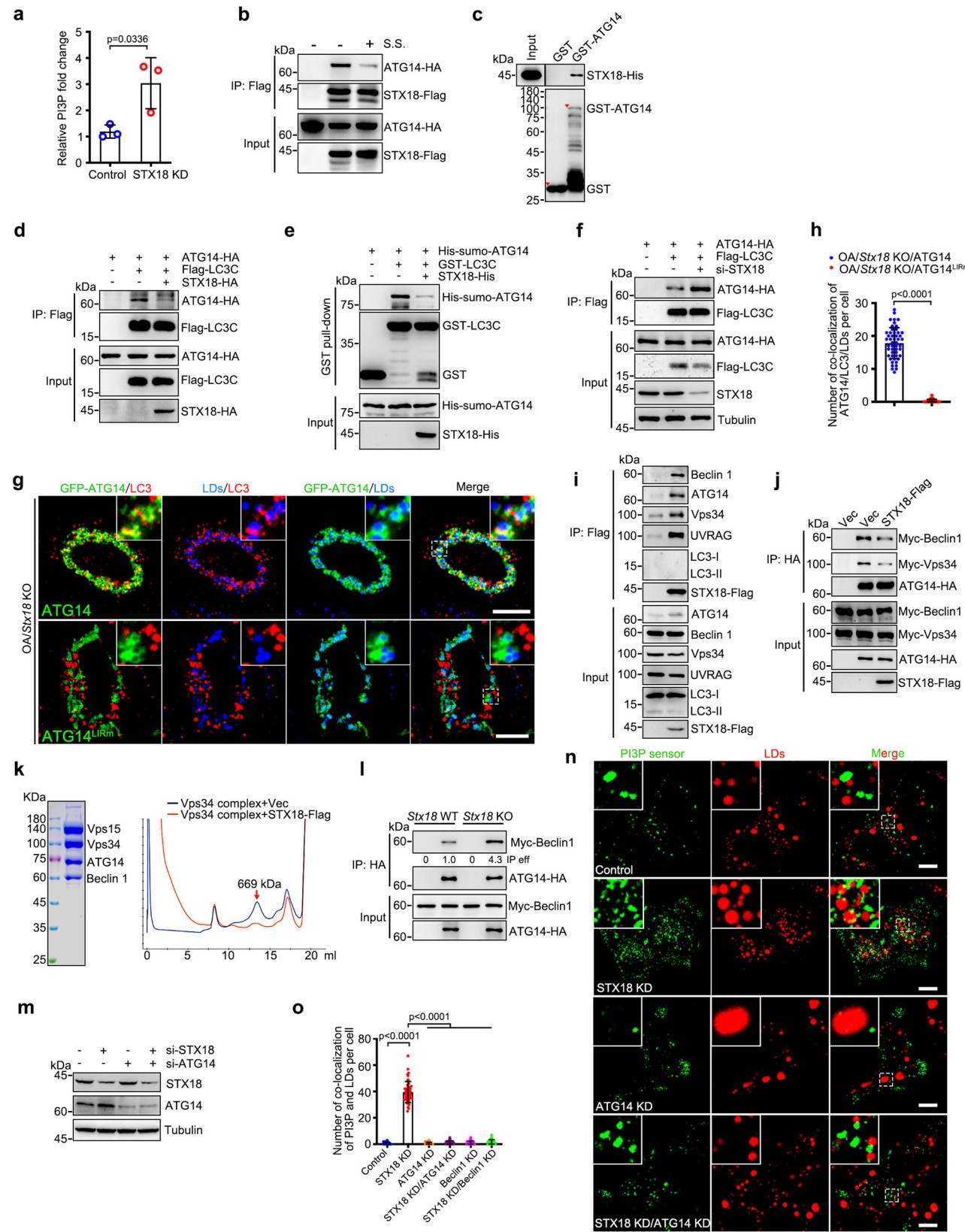

degradation of endogenous Viperin (Fig. 5n). Importantly, ATG14[LIRm] failed to rescue the degradation of Viperin caused by STX18 depletion in *Atg14* KO cells (Fig. 5o). Together, these results indicate that Viperin degradation upon STX18 depletion occurs ATG14-mediated lipophagy.

## SARS-CoV-2 M protein disrupts STX18 for induction of ATG14-mediated lipophagy

Given that STX18 depletion can induce ATG14-mediated lipophagy, resulting in the degradation of anti-viral protein Viperin, we then asked whether SARS-CoV-2 could take advantage of this mechanism to

**Fig. 3 | STX18 negatively regulates the interactions of ATG14-ATG8s and the formation of PI3KC3-C1 complex. a** PI(3)P production was analyzed by the ELISA assay. The fold change was calculated based on the concentration of PI(3)P and normalized against control group. **b** The effect of serum starvation (S.S.) on STX18-Flag and ATG14-HA interaction was analyzed by co-immunoprecipitation. **c** GST pull-down analysis of GST-ATG14 with STX18-His. **d** The effect of STX18-HA expression on the interaction of Flag-LC3C with ATG14-HA was analyzed by immunoprecipitation. **e** The effect of STX18 on the interaction of LC3C with ATG14 was detected by GST pull-down. **f** The effect of STX18 knockdown on the interaction of Flag-LC3C with ATG14-HA was analyzed by immunoprecipitation. **g, h** Co-localization of GFP-ATG14/LC3/LDs in *Stx18* knockout HeLa cells expressing indicated plasmids were treated with 200 μM OA for 12 h. 100 μM CQ was added for 6 h before cells were fixed. The number of colocalization of ATG14/LC3/LDs per cell was counted from 50 cells of three independent experiments. **i** HEK293T cells were transfected with STX18-Flag. Protein interactions were detected by immunoprecipitation with anti-Flag beads and immunoblotting analysis. **j** The effect of STX18-Flag expression on the interaction of ATG14-HA with Myc-Vps34 and Myc-Beclin1 was analyzed by immunoprecipitation. **k** Vps34 complex (Vps15, Vps34, ATG14, Beclin1) was purified from 293 F cells and incubated with or without STX18-Flag and analyzed via Chromatography of C1 on Superose 6 10/300 GL columns. **l** HEK293T *Stx18* wild-type and knockout cells were transfected with indicated plasmids. Protein interactions were detected by immunoprecipitation. **m** HeLa cells were transfected with ATG14 or/and STX18 siRNA for 48 h. Cell lysis were analyzed via western blot. **n, o** HeLa cells expressing GFP-FYVE_SARA were transfected with ATG14 or/and STX18 siRNA for 48 h, and treated with 200 μM OA for 12 h. Co-localization of LDs and GFP-FYVE_SARA. The number of colocalization of PI3P with LDs per cell in (**n** and Supplementary Fig. 4j) was counted from 50 cells of three independent experiments. Data information: Scar bar represents 10 μm for all fluorescence images. Error bars, mean ± SD of three independent experiments. Two-tailed Unpaired Student's *t* test. Source data are provided as a Source Data file.

regulate Viperin, facilitating viral production. First, we sought to determine whether SARS-CoV-2 infection induces lipophagy. The colocalization of LC3 and LD was observed in SARS-CoV-2-infected cells (Fig. 6a). The engulfment of LDs into single-membrane autolysosomes was directly visualized in SARS-CoV-2-infected cells by TEM (Fig. 6b). These evidences support the idea that lipophagy is activated by SARS-CoV-2 infection. Previous studies from our group and other group have indicated that SARS-CoV-2 M expression induces autophagy[33,34]. Here, we found that M expression significantly decreased the number of LDs, and knockdown of Atg7 or CQ treatment abolished M expression-induced LD degradation (Fig. 6c, d). We further confirmed that M expression-induced lipophagy by using the RFP-GFP-Plin2 assay and TEM (Fig. 6e, f and Supplementary Fig. 8a).

Based on our previous IP/MS, STX18 was on the candidate list for the SARS-CoV-2 M interactome[33]. M directly interacted with STX18 by in vivo co-IP and in vitro GST pull-down assays (Fig. 7a, b). M protein 20-100 amino acids are required and sufficient for the interaction with STX18 and M$^{\triangle 20-100}$ overexpression failed to induce autophagy (Supplementary Fig. 8b, c). M$^{\triangle 20-100}$ overexpression failed to decreased the number of LDs (Supplementary Fig. 8d), indicating that M-STX18 interaction is important for M expression-induced lipophagy. We further examined the interactions between M and the autophagic machinery but obtained a negative output (Supplementary Fig. 8e). M did not interact with ER-resident SNARE complexes RINT1, ZW10, USE1, and BNIP1 (Supplementary Fig. 8f). These data indicate that M specifically binds to STX18. We then investigated whether M induces lipophagy dependent on STX18. As expected, the degradation of LDs and RFP + GFP− LDs caused by M expression was prevented in the presence of STX18 overexpression (Fig. 7c–f). M expression did not affect the protein level of endogenous STX18 (Supplementary Fig. 8g), but disrupted the interactions between STX18 and PI3KC3-C1 complex (Fig. 7g), while enhancing the interactions between ATG14-Beclin1-Vps34 and ATG14-LC3C (Supplementary Fig. 8h–j). Furthermore, compared to non-infection, SARS-CoV-2 infection decreased the interaction between ATG14 and STX18 (Fig. 7h). These results indicate that SARS-CoV-2 infection or M expression induces lipophagy by inhibiting the STX18-ATG14 interaction. In SARS-CoV-2-infected *Atg14* KO cells, wild-type ATG14 rescued the colocalization between LC3 and LD, but not ATG14$^{LIRm}$ (Fig. 7i). Taken together, these data indicate that SARS-CoV-2 infection and M expression induce ATG14-mediated lipophagy by disrupting STX18.

Noticeably, we found that M expression also induced the autophagic degradation of Viperin-GFP (Supplementary Fig. 8k, l) but failed to promote the degradation of mutant Viperin$^{\triangle 1-42}$ (Supplementary Fig. 8m). In *Atg14* KO cells, wild-type ATG14 rescued the degradation of Viperin induced by M expression, but not ATG14$^{LIRm}$ (Supplementary Fig. 8n). Together, these data indicate that M expression induces lipophagic degradation of Viperin dependent on ATG14 as the

autophagic receptor. We then sought to examine whether the sequestration of Viperin by lipophagosomes upon SARS-CoV-2 infection would facilitate viral replication. Although Viperin was induced in SARS-CoV-2-infected cells (Supplementary Fig. 8o), we found that Viperin knockdown enhanced SARS-CoV-2 replication (Fig. 7j). We then overexpressed STX18 to inhibit lipophagy and found that overexpression of STX18 significantly decreased the viral replication, whereas Viperin depletion abolished the reduction in viral replication induced by STX18 overexpression (Fig. 7j), suggesting that the reduction in viral replication induced by STX18 overexpression is dependent on Viperin. ATG14 knockdown inhibited SARS-CoV-2 replication (Supplementary Fig. 8p). We further infected *Atg14* KO cells with rescuing wild-type ATG14 or ATG14$^{LIRm}$ by SARS-CoV-2. Consistently, rescued with wild-type ATG14 but not ATG14$^{LIRm}$, led to a reduction in Viperin expression, facilitating viral production (Fig. 7k). Altogether, our data indicate that in the physiological condition, SARS-CoV-2 hijacks STX18 by viral protein M, and subverts the interaction of STX18 with ATG14, promotes the formation of PI3KC3-C1 complex and ATG14-LC3 interaction, resulting in the induction in ATG14-mediated lipophagy to degrade Viperin to evade the anti-viral effect.

## Disscussion

In this study, we identified that the autophagic protein ATG14 targets to LDs and interacts with the core autophagy marker ATG8s via an LIR motif, acting as a receptor regulating LD degradation. In addition, we identified a lipophagy negative regulator, STX18, which interacts with ATG14, disrupting the interactions of ATG14-ATG8s and the formation of PI3KC3-C1 complex by competitively binding to the CCD in ATG14. Knockdown of STX18 induces ATG14-mediated lipophagy, resulting in the reduction of LD-associated anti-viral protein Viperin. SARS-CoV-2 hijacks STX18, inducing ATG14-mediated lipophagy to degrade Viperin for viral replication (Fig. 8).

ATG14, a unique subunit of autophagy-specific PI3KC3-C1 complex, localizes to autophagosomes/autolysosomes upon autophagy activation. The C-terminal BATS domain can sense membrane curvature and bind to membranes through an amphipathic alpha helix[15,16]. It is not surprising that ATG14 localizes to LDs. ATG14 only targets LDs under OA treatment, colocalizes with late LD markers HSD17B11 but fails to colocalize with the nascent LD marker GPAT4$^{152-208}$, indicating that ATG14 traffics to LDs when LDs are produced and accumulated in cells. The mechanisms by which proteins target LDs are not fully understood[35]. Several proteins have been identified to target LDs through amphipathic helices, and depletion of the amphipathic helices abolished the LD association[36-38]. Here we found that disruption of the BATS domain abolished the LD localization of ATG14. In addition to the BATS domain of ATG14, whether the amphipathic helix 303-320 amino acids of ATG14 were also required for LD localization is unknown. Although according to AlphaFold, these internal residues that should not be available for the interaction with LDs. More evidences are

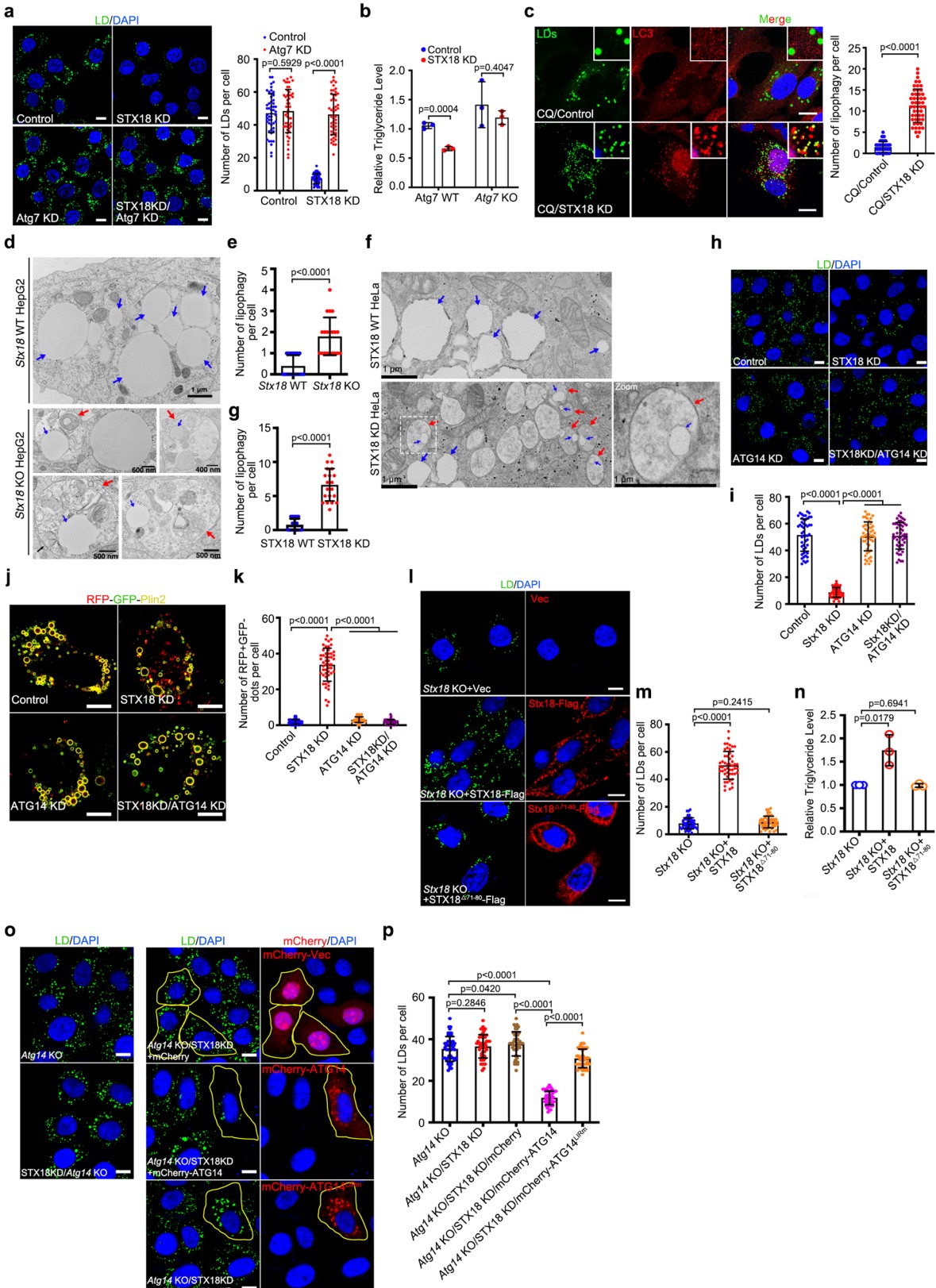

required to investigate the potential mechanism(s) by which ATG14 targets LDs.

ATG14 plays an important role in AMPK activation-triggered lipophagy in yeast[39]. Depletion of ATG14 in mice attenuated hepatic autophagy and elevated triglyceride concentrations, whereas overexpression of ATG14 caused a reduction in triglyceride concentrations[19,40]. Consistent with these findings, we found that knockdown of ATG14 caused the accumulation of LDs in untreated cells, whereas overexpression of ATG14, but not the LD location defect mutants ATG14^{WFY-RRR} or ATG14^{ILI-QQQ}, decreased the LD content and triglyceride storage, and the reductions were abolished by CQ or Baf-A1 treatment. These data lead us to conclude that ATG14 targets LDs

**Fig. 4 | Knockdown of STX18 activates lipophagy dependent on ATG14.**
**a** Representative fluorescence images of HeLa cells transfected with indicated siRNAs and treated with 200 μM OA for 6 h. Number of LDs in each cell was counted from 50 cells of three independent experiments. **b** The concentration of TAG in cells was analyzed according to manufacturer's instructions. **c** HeLa cells were transfected with STX18 siRNA and treated with 200 μM OA for 12 h and 100 μM CQ for 6 h before cells were fixed and immunostained with anti-LC3. Cells were imaged by confocal microscopy. Number of colocalization of LC3/LDs per cell was counted from 50 cells of three independent experiments. **d**–**g** Representative transmission electron micrograph. Blue arrows indicate LDs. Red arrows indicated autolysosome. The number of lipophagy per cell in **d** and **f** was counted from 20 cells of three independent experiments. **h**, **i** Representative fluorescence images of HeLa cells transfected with indicated siRNAs and treated with 200 μM OA for 6 h. Number of LDs per cell was counted from 50 cells of three independent

experiments. **j**, **k** Representative fluorescence images of HeLa cells expressing RFP-GFP-Plin2 transfected with indicated siRNAs and treated with 200 μM OA for 12 h. Number of RFP + GFP- dots per cell was counted from 50 cells of three independent experiments. **l**, **m** Representative fluorescence images of *Stx18* knockout HeLa cells transfected with indicated plasmids and treated with 200 μM OA for 6 h. Number of LDs per cell was counted from 50 cells in three independent experiments. **n** The concentration of triglyceride in cells was analyzed according to manufacturer's instructions. **o**, **p** Representative fluorescence images of *Atg14* knockout HeLa cells expressing indicated proteins transfected with STX18 siRNA and treated with 200 μM OA for 6 h. Number of LDs per cell was counted from 50 cells in three independent experiments. Data information: LDs were labeled with BODIPY-493/503, the nuclei were stained with DAPI, and the scar bar represents 10 μm for all fluorescence images. Error bars, mean ± SD of three independent experiments. Two-tailed Unpaired Student's *t* test. Source data are provided as a Source Data file.

and induces lipophagy. Consistent with the previous study[25], we found that ATG14 contains a functional LIR motif and interacts with the ATG8s, but with a preference for LC3A, LC3C, GABARAP, and GABARAPL2 via in vivo co-IP and in vitro GST pull-down. Although ATG14[LIRm] is still localized on LDs, this mutant failed to bind ATG8s, nor promote the conversion of endogenous LC3-I to LC3-II, or rescue lipophagy. ATG14[LIRm] still interacts with Beclin1 and STX17 and undergoes homo-oligomerization, suggesting that LIR is not required for the formation of PI3K complex and the fusion of autophagosomes with lysosomes. Besides ATG14, a recent study found that ATG9 depletion increases the number and size of LDs in HeLa cells, but not due to a block in autophagic degradation[41]. ATG2A was also found to localize on LDs with ATG14 for unknown functions and mechanisms[18]. Therefore, it would be interesting to investigate whether other autophagic proteins target to LDs and regulate their dynamics in the future.

SNARE complex, the core of the membrane fusion machinery, are membrane-associated trafficking proteins[42]. STX18 (Qa-SNARE) was initially found to localize in the ER and function in transport between the ER and Golgi apparatus[30,43]. The depletion of STX18 resulted in a reduced number and size of LDs, as well as decreased cellular triglyceride levels[26]. The ER-resident SNARE complex (STX18-USE1-BNIP1) functions together with NRZ complex (NAG-RINT1-ZW10) and Rab18 as a bridging complex to maintain ER-LD contact[26]. It is generally anticipated that the smaller LDs are preferentially engulfed by autophagosomes rather than by lipolysis[44]. Thus, it remains to be clarified whether STX18 regulates LD turnover through lipophagy. In our IP results, compared to the control, STX18 showed a weaker interaction with ATG14 under serum starvation. STX18 depletion induces lipophagy dependent on the LIR of ATG14, STX18[△71-80] (defect in ATG14 binding) failed to rescue lipophagy in *Stx18* KO cells. We further noticed that in Atg7 deleted cells, STX18 depletion failed to induce reduction of LD number and triglyceride levels, indicating that STX18 depletion has minimal effect on LD synthesis. Therefore, we draw the conclusion that STX18 directly interacts with ATG14 and inhibits lipophagy via two strategies: (1) blocking the formation of PI3KC3-C1 complex, disrupting the initiation of lipophagy by targeting ATG14; (2) decreasing the interactions of ATG14 with ATG8s, disrupting substrate capture in lipophagy. Although we observed that ATG14 recruited Beclin1 to LDs and that STX18 depletion promoted the LD association of Beclin1 and PI3P, we still had no direct evidence suggesting that ATG14 recruited PI3KC3-C1 complex to LDs, generating PI3P to form pre-autophagosomal structures on LDs in STX18-depleted cells. In addition to SARS-CoV-2, we also found that the M proteins of SARS-CoV, MERS-CoV, HCoV-OC43, HCoV-NL63, and HCoV-229E interact with STX18 (Supplementary Fig. 8q), suggesting that the interactions of coronavirus M proteins with STX18 are conserved. It will be of interest to determine whether other coronaviruses use a similar mechanism to induce lipophagy for viral production.

Viperin is an interferon-inducible protein that is directly expressed in viruses-infected cells. Viperin is one of the powerful antiviral

effectors against a large variety of viral infections, including ZIKV, influenza A virus (IAV), human cytomegalovirus (HCMV), and West Nile virus (WNV)[45–48]. Viperin also catalyzes the conversion of cytidine triphosphate (CTP) to 3′-deoxy-3′,4′-didehydro-CTP (ddhCTP), which acts as a chain terminator for the RNA-dependent RNA-polymerases to inhibit RNA synthesis of the flavivirus family[49]. Viperin contains three distinct domains: an N-terminal amphipathic α-helical domain, a radical S-adenosylmethionine (SAM) domain, and a highly conserved C-terminal domain[50]. The N-terminal amphipathic α-helical domain is necessary and sufficient for Viperin to bind to LD[31]. Mutants lacking the α-helical domain greatly limit antiviral activity, indicating that LD association is important for its antiviral activity[4,32]. During viral infection, the ubiquitin ligase UBE4A was found to stimulate K6-linked polyubiquitination at Lys206 and degradation of Viperin protein[51]. However, whether Viperin can be degraded via autophagy remains unknown. In this study, we found that STX18 depletion or ATG14 overexpression-induced lipophagy led to the degradation of the interferon-induced proteins Viperin. Further analysis showed that other interferon-induced proteins IGTP, TGTP1, and IFI47 were also degraded by lipophagy. In brief, here we report that the LD-located immune proteins can be degraded by lipophagy.

Lipophagy plays an important role in viral replication. DENV infection induces AMPK-dependent lipophagy, correlating with reduced the amounts of LDs and triglyceride levels, and enhanced β-oxidation, to produce energy for viral replication[7,52]. The induction of lipophagy by ZIKV infection significantly reduces the number and volume of LDs[53]. SARS-CoV-2 infection increases LD accumulation by upregulating the expression of key proteins in lipid metabolism[54]. Whether lipophagy is activated and/or important during SARS-CoV-2 infection remains poorly understood. Our data in this study showed that SARS-CoV-2 M expression leads to autophagic degradation of LDs. Mechanically, M interacts with STX18, blocking the interaction between STX18 and PI3KC3-C1 complex, resulting in enhanced associations of ATG14 with LC3 and the formation of PI3KC3-C1 complex, thus inducing lipophagy and Viperin degradation. Although the endogenously synthesized product of the Viperin, ddhCTP, does not affect SARS-CoV-2 replication in cells[55], here we found that Viperin depletion led to the enhanced SARS-CoV-2 replication, indicating that Viperin can negatively regulate SARS-CoV-2 replication. STX18 overexpression or rescued of ATG14[LIRm] in *Atg14* KO conditions inhibited viral replication by blocking the autophagic degradation of Viperin. These data highlight the important roles of STX18 and ATG14 in SARS-CoV-2 replication by regulating lipophagy.

All in all, our study proposes an interesting mechanism in which ATG14 targets LDs and binds to ATG8s to mediate lipophagy. We also identified STX18 as a lipophagy-negative regulator that decreases the interactions of ATG14 with ATG8s and disrupts the formation of PI3KC3-C1 complex by targeting ATG14. SARS-CoV-2 M protein hijacks STX18 to activate lipophagy for Viperin degradation, facilitating viral replication. Our study showed the possible and essential roles

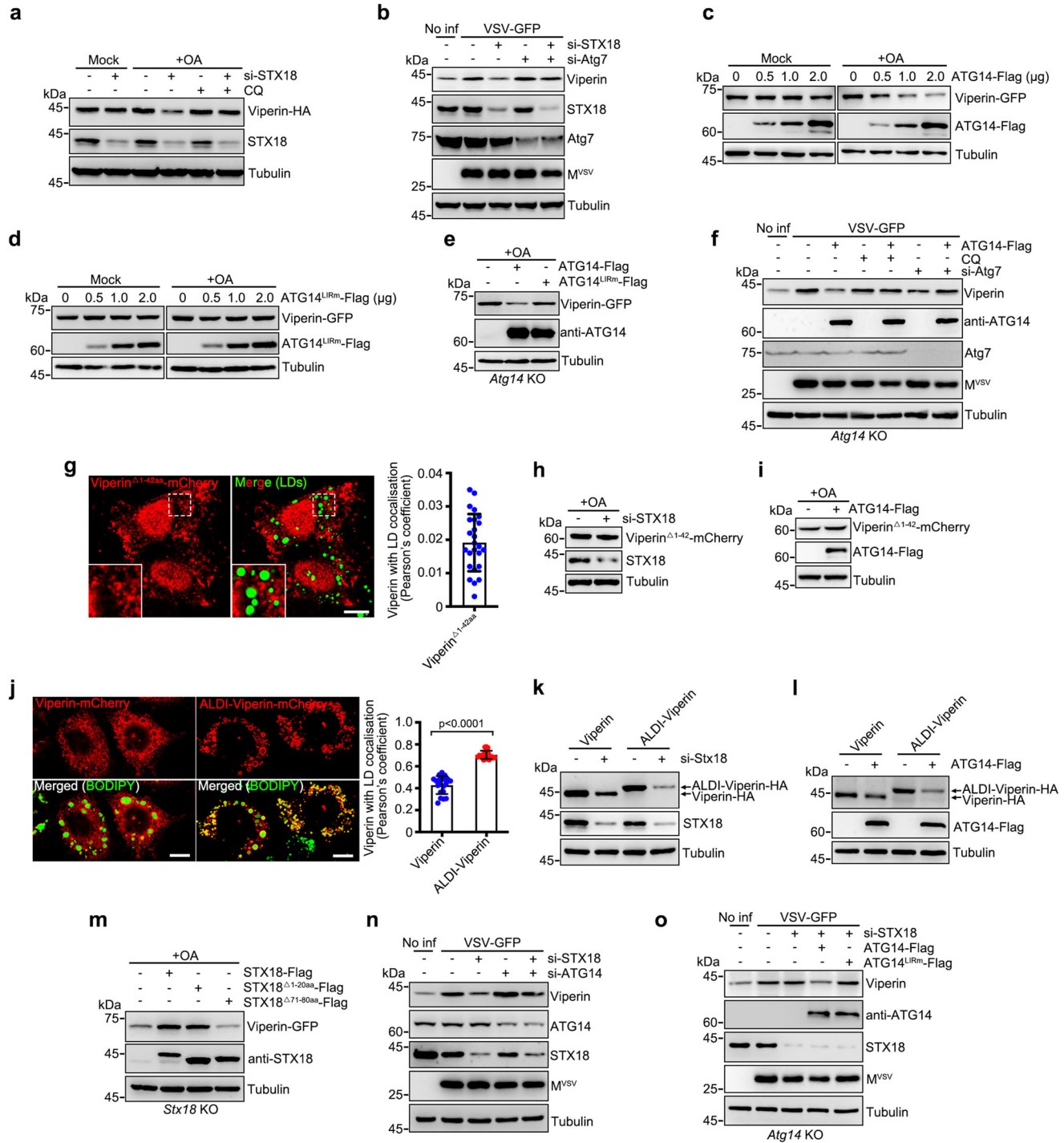

**Fig. 5 | Viperin was degraded by lipophagy. a** HeLa cells expressing Viperin-HA were transfected with STX18 siRNA and treated with or without 200 μM OA for 12 h. 100 μM CQ was added for 6 h before cells were harvested. **b** BHK21 cells were transfected with indicated siRNAs and infected with VSV for 12 h and treated with 200 μM OA for 12 h. **c, d** HeLa cells expressing indicated proteins were treated with or without 200 μM OA for 12 h. **e** *Atg14* knockout HeLa cells expressing indicated proteins were treated with 200 μM OA for 12 h. **f** *Atg14* knockout cells were transfected with Atg7 siRNA and ATG14-Flag for 24 h and then infected with VSV for 12 h and treated with 200 μM OA for 12 h or/and 100 μM CQ for 6 h before cells were harvested. **g** Colocalization of LDs with Viperin^Δ1–42, *n* = 25 cells of three independent experiments. **h** HeLa cells expressing Viperin^Δ1–42-mCherry were transfected with STX18 siRNA and treated with 200 μM OA for 12 h. **i** HeLa cells expressing Viperin^Δ1–42-mCherry were transfected with ATG14-Flag for 24 h and treated with

200 μM OA for 12 h. **j** Colocalization of LDs with Viperin-mCherry or ALDI-Viperin-mCherry, *n* = 20 cells of three independent experiments. **k** HeLa cells expressing Viperin-HA or ALDI-Viperin-HA were transfected with STX18 siRNA and treated with 200 μM OA for 12 h. **l** HeLa cells expressing Viperin-HA or ALDI-Viperin-HA were transfected with ATG14-Flag for 24 h and treated with 200 μM OA for 12 h. **m** *Stx18* knockout HeLa cells expressing Viperin-GFP with STX18-Flag or its mutants were treated with 200 μM OA for 12 h. **n, o** BHK21 cells (**n**) or *Atg14* knockout cells (**o**) were transfected with indicated siRNAs and plasmids and further infected with VSV for 12 h and meanwhile treated with 200 μM OA for 12 h. Data information: Cell lysis were analyzed via western blot (**a–f, h, i, k–o**). LDs were labeled with BODIPY-493/503, and scar bar represents 10 μm (**g, j**). Error bars, mean ± SD of three independent experiments. Two-tailed Unpaired Student's *t* test. Source data are provided as a Source Data file.

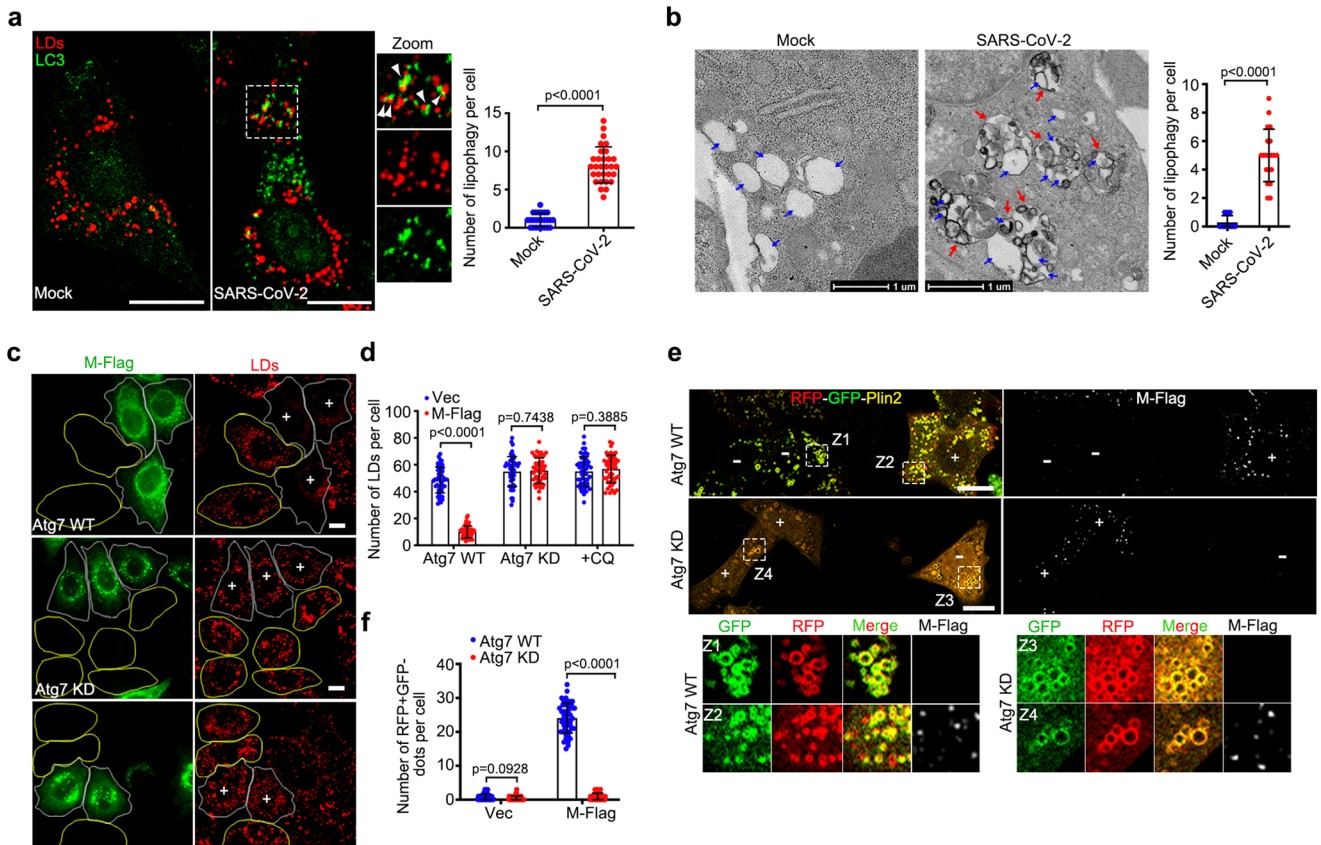

**Fig. 6 | SARS-CoV-2 induces lipophagy. a** SARS-CoV-2-infected Vero cells were treated with 200 μM OA for 12 h and immunostained with anti-LC3. The number of colocalization of LC3/LDs per cell in each group was quantified and shown at the right panel, mock (25 cells) and SARS-COV-2-infected cells (30 cells) of two independent experiments. **b** Representative transmission electron micrograph of non-infected or SARS-CoV-2-infected Vero-E6 cells. Red arrows mark lipophagy. Blue arrows mark LDs. Scar bar represents 1 μm. Quantification of number of lipophagy per LD. 20 cells of two independent experiments were counted. **c** HeLa cells expressing the M-Flag were transfected with Atg7 siRNA for 48 h or treated with 100 μM CQ and 200 μM OA for 6 h, then fixed and immunostained with anti-Flag. Cells were imaged by confocal microscopy. White ROIs and "+" indicate the cells expressing M protein and yellow ROIs indicate the cells without or low M expression as control group. **d** Number of LDs per cell in **c** was counted from 50 cells of three independent experiments. **e** HeLa cells expressing the RFP-GFP-Plin2 and M-Flag were transfected with or without Atg7 siRNA, and cells were treated with 200 μM OA for 12 h, then fixed and immunostained with anti-Flag. Cells were imaged by confocal microscopy. "+" indicate the cells expressing M protein and "−" indicate cells without or low M expression as control group. **f** Number of RFP + GFP-dots per cell in **e** was counted from 50 cells of three independent experiments. Data information: LDs were labeled with LipidTOX Red (red) (**a**, **c**). Scar bar represents 10 μm for all fluorescence images. Error bars, mean ± SD of three independent experiments. Two-tailed Unpaired Student's *t* test. Source data are provided as a Source Data file.

of lipophagy in SARS-CoV-2 replication and suggested potential strategies to block viral replication.

## Methods

### Cell cultures

HEK293T (CRL-1573, ATCC), U2OS (HTB-96, ATCC), HeLa (CCL-2, ATCC), HepG2, MEF, BHK21 (obtained from Mingzhou Chen, Wuhan University), and Vero-E6 (obtained from Kun Cai, Hubei Provincial Center for Disease Control and Prevention) cells were cultured in Dulbecco's modified Eagle's medium (DMEM, Gibco, 11995065) supplemented with 10% fetal bovine serum (Sigma-Aldrich, 12303 C) and 1% penicillin-streptomycin (Gibco, 15140163) at 37 °C with 5% $CO_2$. HepG2, WT MEF, *Fip200* KO MEF, and *Atg5* KO MEF cells were obtained from Qing Zhong (Shanghai Jiao Tong University). HEK293F (obtained from Hongjun Yu, Huazhong University of Science and Technology) suspension cells were cultured in freestyle 293 medium (Union Biotech, Up-1000) at 37 °C under 5% $CO_2$ in shaker. For serum starvation, cells were washed three times with phosphate-buffered saline (PBS, Hyclone SH30256.01) and incubated in DMEM medium for 24 h. *Stx18* WT and KO HEK293T, *Stx18* WT and KO HeLa, *Stx18* WT and KO HepG2, *Atg14* WT and KO HeLa, *Atg14* WT and KO Vero-E6

were constructed in our lab. Penta KO HeLa cells were obtained from Richard J. Youle (NIH).

### OA induced LD accumulation

To analyze the LD turnover, cells were treated with 200 μM OA for 6 h. To examine the colocalization of LD with different proteins, cells were treated with 200 μM OA for 12 h. To measure cellular TAG, cells were treated with 200 μM OA for 12 h.

### Antibodies and reagents

Mouse anti-Flag (AE005, western blot (WB) 1:5000, immuno-fluorescence (IF) 1:200), mouse anti-HA (AE008, WB 1:5000, IF 1:200), mouse anti-GFP (AE012, WB 1:5000), mouse anti-GST (AE001, WB 1:5000), mouse anti-His (AE003, WB 1:5000), mouse anti-mCherry (AE002, WB 1:5000), rabbit anti-ATG14 (A7526, WB 1:2000) and rabbit anti-Beclin1 (A7353, WB 1:2000) were obtained from ABclonal. Mouse anti-Tubulin (E7S, WB 1:10000) was obtained from Developmental Studies Hybridoma Bank. Mouse anti-p62 (H00008878-M01, WB 1:10000) was obtained from Abnova. Mouse anti-Myc (2276, WB 1:5000) and rabbit anti-Atg7 (8558, WB 1:5000) were obtained from Cell Signaling Technology. Rabbit anti-LC3 (PM036, WB 1:1000, IF 1:500) was obtained

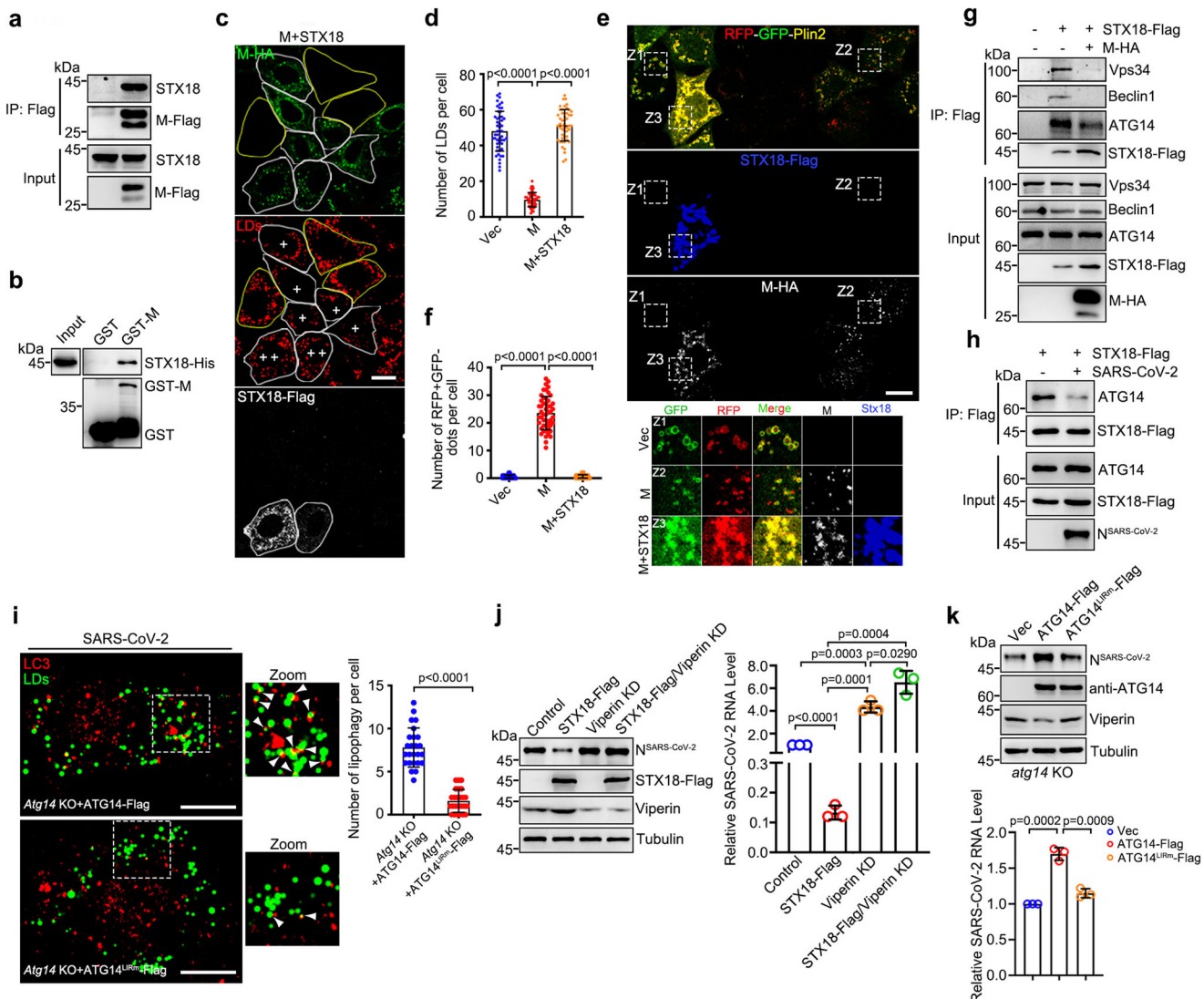

**Fig. 7 | SARS-CoV-2 hijacks STX18 for lipophagy activation. a** Co-immunoprecipitation analysis of M-Flag with endogenous STX18. **b** GST pull-down analysis of GST-M with STX18-His. **c, d** HeLa cells expressing the M-HA and STX18-Flag were treated with 200 μM OA for 6 h, then fixed and immunostained with anti-HA and anti-Flag. Cells were imaged by confocal microscopy. Number of LDs per cell was counted from 50 cells of three independent experiments. **e, f** HeLa cells expressing the RFP-GFP-Plin2 were transfected with M-HA and STX18-Flag for 24 h and were treated with 200 μM OA for 12 h, then fixed and immunostained with anti-HA and anti-Flag. Cells were imaged by confocal microscopy. Number of RFP + GFP-dots per cell was counted from 50 cells of three independent experiments. **g** The effect of M-HA on the interaction of STX18-Flag with ATG14, Beclin1 and Vps34 was detected by immunoprecipitation. **h** The effect of SARS-CoV-2 infection on the interaction of STX18-Flag with ATG14 was detected by immunoprecipitation. **i** *Atg14*

knockout Vero-E6 cells were transfected with indicated plasmids for 24 h and infected with SARS-CoV-2 for 24 h. Meanwhile cells were treated with 200 μM OA for 12 h. Cells were fixed and immunostained with anti-LC3. Cells were imaged by confocal microscopy. Number of colocalization of LC3/LDs per cell was counted from 25 cells of three independent experiments. **j, k** Vero-E6 cells (**j**) or *Atg14* knockout Vero-E6 cells (**k**) were transfected with indicated plasmids for 24 h and infected with SARS-CoV-2 for 24 h. Meanwhile cells were treated with 200 μM OA for 12 h. Cell lysates were analyzed via western blot. Viral RNA level was determined by RT-qPCR. Three independent experiments. Data information: LDs were labeled with LipidTOX Red (**c**) or BODIPY-493/503 (**i**). Scar bar represents 10 μm for all fluorescence images. Error bars, mean ± SD of three independent experiments. Two-tailed Unpaired Student's *t* test. Source data are provided as a Source Data file.

from MBL. Rabbit anti-GFP (ab6556) was obtained from Abcam. Mouse anti-STX18 (sc-293067, WB 1:1000) and mouse anti-GAPDH (sc-365062, WB 1:5000) were obtained from Santa Cruz Biotechnology. Rabbit anti-Vps34 (Z-R015, WB 1:1000) was obtained from Echelon Bioscience. Rabbit anti-ATG14 (19491-1-AP, WB 1:1000), rabbit anti-Viperin (28089-1-AP, WB 1:1000), rabbit anti-CLIMP63 (16686-1-AP, WB 1:1000) and rabbit anti-REEP5 (14643-1-AP, WB 1:1000) were obtained from Proteintech. Mouse anti-SARS-CoV-2 Nucleocapsid (N) (40143-MM05, WB 1:1000) was obtained from Sino Biological Inc. Mouse anti-VSV-M (EB0011, WB 1:1000) was obtained from kerafast. HRP-conjugated goat anti-mouse IgG (H + L) (AS003, WB 1:10000), HRP-conjugated goat anti-rabbit IgG (H + L) (AS014, WB 1:10000) were obtained from Abclonal.

Peroxidase-AffiniPure goat anti-mouse IgG light-chain-specific (115-035-174, WB 1:10000) and peroxidase IgG fraction monoclonal mouse anti-Rabbit IgG, light-chain-specific (211-032-171, WB 1:10000) were obtained from Jackson Immunoresearch. Alexa Fluor 488-conjugated goat anti-mouse IgG (H + L) (A32723, IF 1:500), goat anti-rabbit IgG (H + L, IF 1:500) (A32731), Alexa Fluor 568-conjugated goat anti-mouse IgG (H + L) (A-11031, IF 1:500), goat anti-rabbit IgG (H + L) (A-11036, IF 1:500), Alexa Fluor 647-conjugated goat anti-mouse IgG (H + L) (A-21236, IF 1:500) and goat anti-rabbit IgG (H + L) (A-21244, IF 1:500) were purchased from Thermo Fisher Scientific.

Anti-HA (B26202) and anti-Myc (B26301) magnetic beads were obtained from Bimake. Anti-Flag M2 Affinity Gel (A2220) was obtained

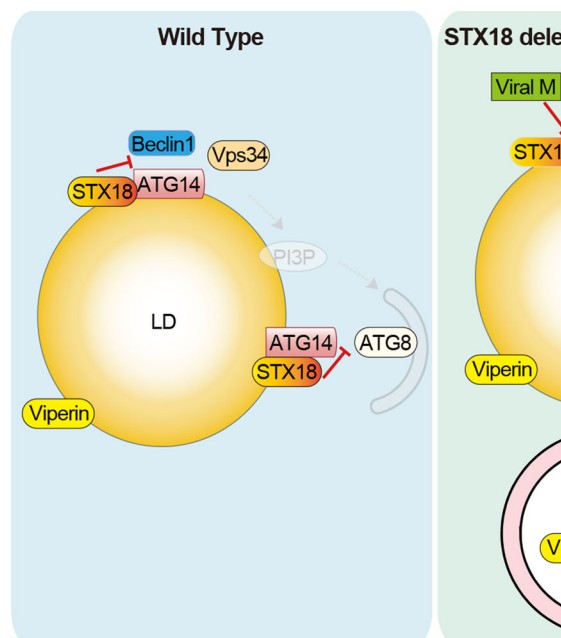

**Fig. 8 | Proposed model for STX18-ATG14 axis regulate lipophagy.** In wild-type cells, STX18 interacts with ATG14 disrupts the interactions of ATG14 with LC3, and subverts the formation of PI3KC3-C1 complex to inhibit lipophagy. In coronavirus infected cells, viral M protein binds STX18 and subverts the STX18-ATG14 interaction to induce lipophagy and degrade the LD-associated anti-viral protein Viperin.

from Sigma-Aldrich. Streptactin beads 4FF was purchased from Smart Lifesciences. Glutathione resin (L00206) was obtained from GeneScript. BODIPY-493/503 (D3922), LipidTOX Red neutral lipid stain (H34476), LipidTOX Deep Red neutral lipid stain (H34477) and disuccinimidyl suberate (DSS, 21555) were purchased from ThermoFisher Scientific. Oleic acid (O1008) and Chloroquine (CQ, C6628) were purchased from Sigma-Aldrich. Bafilomycin A1 (S1413) was purchased from Selleck Chemicals. DAPI stain solution (G1012) was obtained from Servicebio.

### Generation of KO cell line using CRISPR/Cas9
Single guide RNAs (sgRNAs) were designed using synthego (https://www.synthego.com/products/bioinformatics/crispr-design-tool), and annealed and ligated into BsmBI-cut pLenti-CRISPR-V2 vectors. Lentivirus were produced by cotransfecting pLenti-CRISPR-V2 with packaging plasmids pMD2.G and psPAX2 into 293 T cells. The supernatant from transfected 293 T cells were filtered through a syringe with 0.45 μm filters. Cells were infected with Lentivirus for 24 h, and then selected with puromycin. sgRNA targeting for STX18: AAGACGCGGAACAAGGCGCT. sgRNA targeting sequences for ATG14 #1: GCCATCATGGCGTCTCCCAG, ATG14 #2: CCATCATGGCGTCTCCC AGT, ATG14 #3: GCCCGGGCTCCCTTCCCACT, ATG14 #4: CATGGCG TCTCCCAGTGGGA.

### DNA construction
pCDNA4-STX18-Flag and its truncation mutations were cloned into pCDNA4-Flag vector. pCDNA4-STX18-HA was cloned into pCDNA4-HA vector. mCherry-ATG14 and its point mutants were cloned into pCDNA4-mCherry vector. EGFP-ATG14 and its point mutants were cloned into pCDNA4-EGFP vector. pCDNA4-ATG14^LIRm-Flag was cloned into pCDNA4-Flag vector. GFP-STX18, GFP-Beclin1 and GFP-LC3C were cloned into pCDNA4-EGFP vector. mCherry-Beclin1 was cloned into pCDNA4-mCherry vector. GFP-GPAT4^152-208 was generated by cloning sequences encoding the hairpin domain of GPAT4 from pEGFP-N1-GPAT4 into pCDNA4-EGFP vector. GST-ATG8s were generated by cloning sequences encoding the full-length ATG8s from

pLVX-Flag-ATG8s into PEGX-GST vector. GST-M, GST-Beclin1, GST-ATG14 and GST-ATG14^CCD were cloned into PEGX-GST vector. His-sumo-ATG14 and its point mutants were cloned into pCOLD-6×His-sumo vector. pCDNA4-Vperin-HA was cloned into pCDNA4-HA vector. pCDNA4-Vperin-mCherry and its truncation mutations were cloned into pCDNA4-mCherry vector. pLenti-ALDI-Vperin-mCherry was cloned into pLenti-ALDI-mCherry vector. pLenti-Viperin-HA was cloned into pLenti-HA vector. pLenti-ALDI-Vperin-HA was cloned into pLenti-ALDI-HA vector. pCDNA4-Vperin-GFP was cloned into pCDNA4-GFP vector. STX18-His was cloned into PET28-HIS-TEV vector. cDNA encoding HSD17B11, TGTP1, IGTP1 and IFI47 were cloned into pCDNA4-GFP vector. SARS-CoV-M, MERS-CoV-M, HCoV-NL63-M, HCoV-229E-M were synthesized by Sangon Biotech and cloned into pCDNA4-Flag vector. All constructs were confirmed by DNA sequencing.

### DNA transfection and RNA interference
Plasmid DNAs were transfected into 293 T using Polyethylenimine Linear (Yeasen, 40815ES03) or HeLa and Vero cells using Lipofecta-mine 2000 (Invitrogen, 11668019) according to the manufacturer's instructions. siRNAs were introduced into cells using riboFECT CP Transfection Kit (RiboBio, C10511-05). Cells were then cultured for 48 h and harvested. The knockdown efficiency for each protein was evaluated by Western blot analysis. siRNAs used in all experiments were synthesized by GenePharma. siRNA targeting sequences for STX18 #1: GGAGATACATTCCCAGCAA, STX18 #2: GCGTGTGGATCCTCTTCTT, STX18 #3: GGAAGGCAACGAAGACATA. siRNA targeting sequences for Atg7 #1: GCCAACAUCCCUGGAUACAAG, Atg7 #2: CUGUGAACUUCUC UGACGU. siRNA targeting sequences for ATG14 #1: GGCAAAUCU UCGACGAUCCCAUAUA, ATG14 #2: GAUCACAACGGAGACACCAGCA UUA. siRNA targeting sequences for p62: GCATTGAAGTTGATAT CGAT. siRNA targeting sequences for HSC70: GTCCTCATCAAGCGT AATA. ATGL siRNAs were obtained from RiboBio.

### Immunoprecipitation and western blot
Cells were transfected with the indicated plasmids for 36 h, and cellular proteins were extracted with TAP buffer (20 mM Tris-HCl, pH 7.5,

150 mM NaCl, 0.5% NP-40, 1 mM NaF, 1 mM Na3VO4, 1 mM EDTA, protease cocktail) for 30 min on ice. The supernatants were collected by centrifugation at 13000 rpm for 20 min at 4 °C. For endogenous IP, the supernatants were precleared by incubating with protein A/G plus-agaroses for 1 h at 4 °C with rotation, and then the supernatants were collected by centrifugation at 2300 rpm for 2 min at 4 °C. Specific primary antibodies were added in supernatants and incubated overnight with rotation, then the protein A/G plus-agaroses were added in supernatants and incubated for another 4 h with rotation. For Flag, Myc or HA tag IP, the supernatants were incubated with beads overnight. Beads were washed three times with TAP buffer, boiled at 100 °C for 10 min in SDS protein loading buffers, and analyzed by western blot. Equal amounts of protein were separated by on 10% or 12% gradient SDS-PAGE gels and transferred to nitrocellulose membranes. Membranes were blocked with 5% skim milk in PBST (1× PBS and 0.1% Tween20) for 30 min at room temperature and then incubated with the relevant primary antibodies overnight at 4 °C. After washing 3 times with PBST for 5 min each, the membranes were incubated with HRP-conjugated secondary antibodies in 1% skim milk for 1 hour at room temperature. Membranes were washed and signals were developed using ECL Plus western blot substrate. Western blotting images were obtained using Tanon 5200 and analyzed using Bio-Rad Image Lab Software for PC Version 6.1.

### Protein expression and purification

The four plasmids pCAG-strep(2x)-flag-hATG14, pCAG-hP150, pCAG-hVps34 and pCAG-hBeclin1 were transfected into HEK293F cells for 36 h. Cells were harvested in the cold PBS, and lysed in buffer containing 25 mM Tris-HCl (pH 8.0), 150 mM NaCl, 2 mM DTT, 1% Triton X-100 and protease inhibitor cocktail. Cell suspension was collected by centrifugation at 15,000 rpm for 1 h at 4 °C. The supernatants were incubated with Strep-Tactin resin overnight. The resin was washed 3 times. The complex was eluted with 10 mM desthiobiotin and incubated with or without STX18-Flag and further analyzed by gel filtration on Superose 6.

### GST pull-down assay

GST or His tagged proteins were constructed into pGEX-4T vector or pCOLD-6×His vector. Proteins were purified from *E. coli* Rosetta (DE3) or BL21 (DE3). Bacteria were grown at 37 °C to an OD600 of 0.6–0.8, then protein expression was induced with 0.4 mM IPTG for 20 h at 16 °C. Bacterias were harvested by centrifugation and resuspended in lysis buffer (20 mM Tris, pH 7.4, 300 mM NaCl, 0.1% NP-40, 1 mM PMSF) and then lysed by sonication. Lysates were centrifuged at $10,000 \times g$ for 15 min and collected the supernatant. The GST beads were washed three times by lysis buffer, and the indicated protein samples were incubated with GST beads overnight at 4 °C, and then the beads were washed with lysis buffer three times before adding SDS protein loading buffer. Samples were analyzed by western blot.

### Transmission electron microscopy

Cells were fixed in 2.5% gluteraldehyde in 0.1 M sodium cacodylate buffer, pH 7.4, for 2 h at 4 °C and washed 3 times with 0.1 M sodium cacodylate buffer. Cells were then postfixed with cold 1% OsO4, 1.5% potassium ferrocyanide, and 1 mM CaCl2 in 0.1 M sodium cacodylate buffer for 2 h at room temperature and washed 3 times with 0.1 M sodium cacodylate buffer. Next, the cells were dehydrated in a graded series of ethanol, substituted with propylene oxide, and infiltrated with EMbed-812 resin mixed 2:1, 1:1 with propylene oxide and pure EMbed-812 for 8 h each in 37 °C oven. Then samples were placed into molds with fresh resin and polymerized in 60 °C oven for 48 h. Ultrathin sections (80 nm) were prepared with Leica EM UC7 and stained in 2% uranyl acetate in 50% acetone followed by staining in 0.2% lead citrate. Sections were observed with a Hitachi H-7000FA transmission electron microscopy at 80 kV.

### Immunoelectron microscopy

**High-pressure freezing.** The HeLa cells were cultured on 3 mm sapphire discs. Sapphire disc was placed with cells facing up on flat aluminum planchette and another aluminum planchette with 25 μm depth inner space was used as a cover. The spaces between the two aluminum planchettes were filled with 1-hexadecane. Then the samples were frozen immediately using the EM ICE high pressure freezing machine (Leica) and rapidly transferred into liquid nitrogen for storage.

**Freeze substitution and ultrathin section.** After all of the samples were frozen, the samples were transferred into the EM ASF2 (Leica) for substitution. Samples were incubated for 48 h in acetone contained 0.2% UA at −90 °C. Then the temperature was raised to −50 °C in 4 h. After incubated in acetone contained 0.2% UA for another 12 h, the temperature was raised to −30 °C in 4 h. After 2 h incubation at −30 °C, the samples were rinsed three times with pure acetone (15 min each). Then the samples were gradually infiltrated in HM20 resin with grades of 25%, 50%, 75% and pure resin (2 h each) at −30 °C. After infiltrated in pure resin overnight, the samples were embedded in gelatin capsules. The samples were polymerized under UV light for 48 h at −30 °C and 12 h at 25 °C. After polymerization the samples were trimmed and ultrathin sectioned with a microtome (Leica UC7). Serial thin sections (100 nm thick) were collected onto formvar-coated nickel grids.

**Immunogold labeling.** The formvar-coated nickel grids with sections were incubated in 0.01 M PBS contain 1% BSA, 0.05% Triton X-100 and 0.05% Tween20 for 5 min. Then the sections were incubated in the GFP antibody (Abcam, ab6556) diluted in 0.01 M PBS contain 1% BSA and 0.05% Tween20 at 4 °C overnight. After washed 6 times (2 min each) with 0.01 M PBS, the sections were incubated in the secondary antibody (goat anti-rabbit conjugated with 10 nm gold) diluted in 0.01 M PBS contain 1% BSA and 0.05% Tween20 (1:50) for 2 h at RT. After washed 6 times (2 min each) with 0.01 M PBS and 4 times (2 min each) with distill water, the sectioned were dried in the RT and examined in a transmission electron microscopy (Thermo Fisher/FEI Talos L 120 C).

**LD purification.** Cells were treated with 200 μM OA for 12 h. LDs were isolated as described previously with some modifications[56]. Briefly, cells from ten 150 mm plates were collected and washed with PBS buffer. Then samples were suspended in 2 ml buffer A (20 mM tricine and 250 mM sucrose, pH 7.8) plus 0.2 mM PMSF and keep the cells on ice for 20 min. Homogenize the samples by using 5 ml plastic coated tissue grinder for 20 times. The supernatants were collected by centrifugation at $3000 \times g$ for 10 min at 4 °C. The supernatants were transferred into SW40 tube and filled up to 10 ml with buffer A, then loaded 4 ml buffer B (20 mM HEPES, 100 mM KCl and 2 mM MgCl2, pH 7.4) to create a discontinuous gradient. Centrifuged the samples at $38,000 \times g$ for 1 h at 4 °C with slowly acceleration and deceleration. Carefully collected LDs from the top band of the gradient and washed LDs five times with buffer B. LDs were suspended in SDS protein loading buffers and boiled at 100 °C for 10 min and analyzed by western blot.

### Triglyceride quantification assay

Cells were treated with 200 μM OA for 12 h and cellular triglyceride level was measured using the Triglyceride Colorimetric Assay Kit (Cayman Chemical, 10010303). Briefly, cells were collected by centrifugation at $1000 \times g$ for 10 min at 4 °C, then cells were resuspended in 1 ml of cold diluted Standard Diluent and homogenized on ice by ultrasonic. Cell suspension was collected by centrifugation at $10,000 \times g$ for 10 min at 4 °C. For triglyceride assay, 10 μl of sample was added to plate and the reaction was initiated by adding 150 μl diluted Enzyme Mixture solution to each well. Incubate the plate for 30 min at 37 °C and measure the absorbance at 530–550 nm.

### In vitro PI3P ELISA assay

Endogenous VPS34 protein was immunoprecipitated from 293 T cells transfected with or without STX18 siRNA for 36 h and subjected to VPS34 activity assay using Class III PI3K ELISA Kit (Echelon, K-3000) following the manufacturer. Briefly, 40 μl kinase reaction buffer (50 mM Tris-HCl at pH 7.4, 10 mM MgCl2, 10 mM MnCl$_2$), 8 μl of 500 μM PI, 2 μl of 1.25 mM ATP were added to the immune complex and incubated at 37 °C for 1 h. The reaction was terminated by adding 10 μl of 100 mM EDTA. The quenched reaction mixture and PI3P detector protein were added together to the PI3P-coated microplate for competitive binding to the PI3P detector protein. The amount of PI3P detector protein bound to the plate was determined through colorimetric detection of absorbance at 450 nm. The concentration of PI3P in the reaction mixture was calculated as reversed to the amount of PI3P detector protein bound to the plate.

### SARS-CoV-2 virus infection

Vero-E6 cells were infected with SARS-CoV-2 WBP at a multiplicity of infection (MOI) of 0.05 PFU/cell for 1 h at 37 °C with 5% CO$_2$. Cells were cultured with fresh medium supplemented with 2% FBS. All experiments with the SARS-CoV-2 virus were conducted in the BSL-3 laboratory of Hubei Provincial Center for Disease Control and Prevention.

### RT-qPCR

Cells were infected with SARS-CoV-2 WBP at a multiplicity of infection (MOI) of 0.05 PFU/cell for 1 h, cells were cultured with fresh medium supplemented with 2% FBS for 24 h. RNA was extracted using Virus DNA/RNA Extraction Kit (Tianlong, T016). The level of SARS-CoV-2 RNA was measured using COVID-19 Nucleic Acid Test Kit (eDiagnosis) according to manufacturer's instructions.

### Immunofluorescence analysis

For immunostaining, cells cultured on coverslips were washed twice with PBS (135 mM NaCl, 4.7 mM KCl, 10 mM Na2HPO4, 2 mM NaH2PO4), then fixed in 4% paraformaldehyde for 20 min at room temperature, and permeabilized with 0.1% Triton X-100 in PBS for 10 min. After washing twice in PBS, cells were blocked with 1% BSA in PBS for 30 min and incubated with the appropriate primary antibodies in 1% BSA overnight at 4 °C and then with secondary antibodies for 1 h at room temperature. For LD straining, cells were incubated with BODIPY-493/503, LipidTOX Red or LipidTOX Deep Red for 30 min in PBS for another 30 min at room temperature. After washing with PBS three times, samples were mounted on slides with antifade mounting medium. Cells were imaged with a laser scanning confocal microscope (LSM780, Zeiss, Germany) equipped with multiple excitation lasers (405-nm, 458-nm, 488-nm, 514-nm, 561-nm, 633-nm). Fluorescence images were analyzed using ImageJ (2.0.0-rc-59/1.51k) and ZEN 3.0 (black edition, 16.0.1.306).

### Image analysis

All image analysis and processing were performed using ImageJ. Pearson's Coefficient of colocalization analysis were performed using a plugin named JACoP in ImageJ. Total area of LDs in each cell analysis, pictures were adjusted threshold and cells were manually selected, and further analyzed using Analyze Particles in ImageJ.

### Statistics and reproducibility

Experiments in Figs. 1e, 2c, h, 3c, i, m, 6a, b are independently repeated two times. Experiments in Figs. 1b–d, h; 2a, b, d, i–k; 3b, d–f, j, l; 5a–f, h, i, 5k–o; 7a, b, g, h are independently repeated three times. For the rest of experiments, reproducibility is stated in figure legends. No data were excluded.

Statistical parameters including the definition and exact values of *n*, distribution and deviation are reported in the figure legends. Data are expressed as mean ± standard deviation (SD). The significance of the variability between different groups was determined by two-way analyses of variance using GraphPad Prism software (v9.5.1). A *p* value of < 0.05 was considered statistically significant and a *p* value of > 0.05 was considered statistically non-significant. *P* value as exact values were given, ****$p < 0.0001$.

### Reporting summary

Further information on research design is available in the Nature Portfolio Reporting Summary linked to this article.

## Data availability

All data generated or analyzed during this study are included in this article and its supplemental materials. Source data are provided with this paper.

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

## Acknowledgements

We thank professor Qing Zhong (Shanghai Jiao Tong University) for helpful discussion and pCDNA4-ATG14-Flag, pCDNA5-ATG14-HA, pCDNA3-myc-his-ATG14, pCDNA3-myc-his-Beclin1, pCDNA3-myc-his-Vps34 and GFP-LiveDrop; professor Nevan J. Krogan (UCSF) for providing SARS-CoV-2 viral proteins expression plasmids; professor Peng Li (Tsinghua University) for pEGFP-C1-RINT1, pEGFP-C1-ZW10, pEGFP-C1-USE1 and pEGFP-N1-BNIP1; professor Mark A McNiven (Mayo Clinic) for RFP-GFP-Plin2; professor Yueguang Rong (Huazhong University of Science and Technology) for pLVX-Flag-ATG8s; professor Meisheng Ma (Huazhong University of Science and Technology) for pCAG-Strep(2x)-Flag-hATG14, pCAG-hVps34, pCAG-hBeclin1, and pCAG-hP150; professor Shunji Jia (Tsinghua University) for EGFP-FYVE$_{SARA}$; professor Hui Zheng (Soochow University) for shViperin; Pei Zhang (Wuhan Institute of Virology) for TEM assistance; the Electron Microscopy center of Shanghai Institute of Precision Medicine, Shanghai Ninth People's Hospital, Shanghai Jiaotong University School of Medicine, for their technical support and assistance in the electron microscopy. We would like to thank Editage (www.editage.cn) for English language editing. This work was supported by the National Natural Science Foundation of China (32370809, U22A20337) to B.D., the Major Research Plan of the

National Natural Science Foundation of China (92054107) to B.D., and Open Research Fund Program of the State Key Laboratory of Virology of China (2022KF006) to B.D.

## Author contributions

Z.Y., J.L. and B.D. performed most experiments. R.C. contributed to protein purification and repeated the experiments. F.Z. and W.Z. contributed to cloning and repeated the experiments. X.T. repeated the experiments. K.C. and B.H. contributed to SARS-CoV-2 infection experiments. H.C. contributed with Immunoelectron Microscopy. Y.J. provided critical reagents. B.D. conceived the project, designed the experiments, analyzed the data. B.D., Z.Y. and J.L. wrote the manuscript. All authors discussed the results and commented on the manuscript.

## Competing interests

The authors declare no competing interests.
