## [Peer Review File · Nature Communications]

ATG14 Targets Lipid Droplets and Acts as An Autophagic Receptor for Syntaxin18 Regulated Lipid Droplet TurnoverREVIEWER COMMENTS

Reviewer #1 (Remarks to the Author):

The authors report ATG14 as a receptor for Syntaxin18 mediated lipid droplet turnover. They identify two regions in ATG14 required for the recruitment of ATG14 to lipid droplets and they identify STX18 as a novel ATG14 interactor, whose depletion induces ATG14 dependent lipophagy and degradation of lipid droplet associated proteins including viperin. The manuscript contains interesting new data concerning the role of ATG14 and STX18 in lipophagy and adds to our understanding of the process. The physiological role of the molecular interactions described here remains unaddressed.

While most of the experimental results are convincing, addressing the points listed below would be required before publication can be recommended. Addressing point 4 is absolutely essential.

1. I do not understand line 94 and line 100 '... ATG14 only targets to new biogenesis LDs'.
2. Fig1E: In ATG14 samples, it is not clear if there is indeed specific enrichment of gold particles on lipid droplets or whether gold particles are spread all over the sample. In GFP controls no lipid droplets are visible at all. Therefore neither the negative control nor the experimental sample are convincing.
3. Line 117: Pls indicate amino acid numbers for WFY motif
4. Line 108: I303, L317, I320 mutant. According to AlphaFold, I303 and L317 are internal residues that are not accessible from the surface of the domain. These residues are therefore unlikely to be available for the interaction with lipid droplets, which is difficult to reconcile with the authors' claim of a receptor function for ATG14 via said residues. The authors should check that the mutant protein is still functional in non-lipid droplet related tasks.
5. Can the authors identify a minimal domain / an amphipathic helix in ATG14 that on its own targets lipid droplets?
6. Fig2A: Authors report binding of ATG14 to several LC3/GABARAPs and failure to bind LC3B and GABARAPL1. Results are inconsistent with ref 24. GST-LC3B in Fig2A should be a single band, as for the other proteins, but appears as multiple species. Positive control for the functionality of LC3B and other proteins in this assay is needed.
7. Fig2C: Selective interaction of ATG14 with LC3ii is remarkable. A control LIR-containing protein should be used as a control.
8. Line 170: Please explain the origin of the LC-targeting amphipathic helices.
9. FigS2: 2A quantification of STX18 on lipid droplets missing. Controls for 2B missing.

10. Line 277: Authors report that siSTX18 in PentokO cells reduces number of LD. They conclude degradation via lipophagy. Pos evidence of autophagy involvement required, for example depletion of essential ATG other than ATG14/VPS34 complex such as ULK complex, ATG16 complex

11. Antiviral effects rely largely on overexpression experiments.

12. Overexpression of ATG14 affects viperin and viral replication but depletion of endogenous proteins, including STX18, has only minimal effects. Is lipophagy of anti-viral restriction factors of physiological relevance?

Reviewer #2 (Remarks to the Author):

Review: Nat Comm 'ATG14 Targets Lipid Droplets and Acts as An Autophagic Receptor for Syntaxin18 Regulated Lipid Droplet Turnover'

General comments :

This manuscript suggests that the lipophagy promoting activity of ATG14 is via direct recruitment of LC3 positive membranes to LDs, an activity which is negatively impacted by interactions with the SNARE protein STX18. An interesting functional role for these proteins is tied to a SARS-CoV-2 infection where the M-protein seems to disrupt this interaction, thereby promoting viral infection through turnover of the LD resident, anti-viral protein Viperin. Most of the data is fairly convincing, some specific concerns are noted below. Conceptually, it would be better to clearly show that the loss of STX18 has no direct impact on LD synthesis/expansion as put forth by the Li laboratory (Xu et al, JCB 2018) and instead is a driver of autophagic/lysosome-based turnover of LD content. The figures concerning viral infection and the abrogation of the ATG14/STX18 nexus for controlling lipophagy seems interesting, though almost a part of a different story?

Summary and Specific Suggestions :

Fig1 shows ATG14 localization to LDs, which has previously been shown (Pfisterer et al, JLR 2014) as referenced. The authors do attempt to map out the amphipathic helix (Fig 1F,G) required for targeting ATG14 to LDs, though the images presented are not conclusive. The authors demonstrate that the BATS domain (membrane curvature sensing domain) of ATG14 is required for LD localization, which is a more convincing component of this image panel (1G).

Q1) The images in 1G are not convincing to show a lack of LD localization for the amphipathic helix mutant (ILI-QQQ), perhaps improve the magnification or clarify with additional examples?

A knockdown of ATG14 is associated with an increase in LD content of HeLa cells, consistent with previously published findings by Xiong et al in-liver tissue (JBC 2012) and Pfisterer et al in cell lines (JLR 2014). When overexpressed, ATG14 suppresses LD content when localized properly, which was reversed by chloroquine, indicating a likely role in promoting lipophagy (1J, K).

Note1) The authors try to use Atglistatin in HeLa cells, which to my knowledge has only been successful in murine cells. They should perhaps remove this data and not make claims related to lipolysis in this context.

Note2) It will be more convincing if more autophagy inhibitors were utilized to solidify the findings, at least include Bafilomycin A1.

Note3) As the authors suggest that ATG14 does not localize to nascent LDs, it is hard to understand why in the steady state (without OA) ATG14 KD could increase the abundance of LDs (Figure 1I). Also, it seems necessary to show the efficiency of ATG14 KD by WB or RT-qPCR.

Note4) Besides Figure 1D, the authors should measure and compare the enrichment of ATG14 in cellular fractions with or without OA.

Q2) It could be interesting to test the interaction between ATG14 and LDs if I113, I120, and/or L123 in ATG14 are/is mutated?

Note5) I suggest including a control group of ATG14 WT in Figure 1g to show localization differences.

In FigS1 the Authors find ATG14 did not localize to nascent LDs, showing modest colocalization between GPAT4(152–208) and ATG14, a membrane marker for nascent LDs.

Note6) It would be good to show ATG14 colocalization with late LD markers such as HSD17B11, Ldsdh1 as shown in Song J et al., ncb 2022.

Q3) This figure needs a control group which was not treated with DSS in Figure S1e?

Figure 2 explores the direct association of ATG14 with core autophagic machinery such as LC3 and GABARAP, which has been reported by other groups. They show a clear association with LC3A, LC3C and GABARAP/L2 and do define the LIR. The authors show that there is a requirement of LIR bearing ATG14 to recruit LC3 membranes to LDs in panel 2L, further support is gained from KO/re-expression studies as well.

Q4) It is hard to appreciate exactly how many cells were scored from the immunofluorescence panels presented in Fig2 L/M. Was this listed in the manuscript text or figure legend? Were these results displayed in panel 2M obtained from one experimental trial?

Q5) The localization of LC3/ATG14 on LDs of serum starved HeLa cells is very graphic in panel 2L, would the authors be able to provide some additional images or increased magnifications?

Q6) Do the HeLa cells in panel 2L have increased labeling of LC3 following starvation? There certainly seems to be quite a high level of this protein in non-starved cells. How specific is the LC3 antibody labeling in 2L? Are the large red structures representative of autophagic vacuoles or mischaracterized? What do these LDs and autophagic membranes look like by EM?

In this context it would be good if authors provide control/mock group in Figure 2e.

Q7) It is impressive to see the interaction between ATG14 and LC3. Have the authors tried to see these interactions in endogenous context with or without OA treatment?

Q8) In Fig2 panels N,O the authors show no significant look at LD number differences between experimental groups. Could they include an analysis of total LD area/cell? Would the authors be able to provide some additional images or increased magnifications?

Fig 3: Mass spec.-based identification of ATG14 binding partners that change following serum starvation are shown in Fig3. A large decrease in STX18 binding is seen following serum starvation, which itself has been found on the LD and seems to influence LD size and TG amounts by others. Interestingly, the authors show a dramatic level of ATG14/LC3 localization on LDs in STX18 KO cells (Fig 3H).

Q9) Is the STX18 KO condition associated with increased levels of autophagy? There is so much LC3 labeling in these KO cells, perhaps the state of autophagic flux should be addressed? Are these LC3 membranes the result of impaired autophagic flux? What are the protein levels of p62 and LC3II in these STX18 KO cells?

Note7) The authors found the STX18-ATG14 through MS. It might also be interesting to confirm the binding between these two proteins in an endogenous context, with or without OA treatment, with or without an autophagy inhibitor.

Note8) The colocalization between ATG14 and LD should be done in a context of STX18 knockdown or in STX18 KO cells, potentially following rescue by exogenously expressed STX18.

Note9) In Figure S3G, it was nice to show the overall protein of STX18 by using anti-STX18 antibody. It would be helpful if the authors could put a label indicating which was endogenous/exogenous.

Q10) In Figure 3I, it seems that every protein examined interacted with STX18, these findings could be strengthened by a negative result?

Q11) In Figure 3n,o, the quantification should go after the taken images. So, the panel number for 'n' and 'o' should be exchanged. Also, the knockdown efficiency should be detected by western blot analysis or RT-qPCR assay.

Fig4 demonstrates that a knockdown of STX18 suppresses LD content, which seems to be dependent on autophagy as a knockdown of Atg7 results in LD retention. Consistent with this is an increased frequency of LDs found in single membrane bound autolysosome like structures, what the authors term 'lipophagosomes', in Fig 4D-F following STX18 knockdown or KO. They also use a GFP-RFP-PLIN2 reporter to show that there are more RFP only + LDs in the cells following STX18 KD, consistent with the idea that in this knockdown state there is more lipophagy occurring.

Q12) Are these structures that contain LDs indeed representative of anything related to autophagosomes? They appear to be lysosomes and are by the authors' own admission single membrane bound.

Q13) How do these findings of STX18 KO suppressing LD content relate to the findings of Xu et al (JCB, 2018) where this SNARE is implicated in promoting LD formation/expansion at the ER? If their model is followed, wouldn't the findings of this manuscript be consistent with a STX18 loss reducing LD content from a synthesis aspect?

Q14) In the image 4A, in syx18/ ATG 7 double KD, LD number look less than control and ATG 7 KD however graphically look similar.

Q15) Why are there so many LDs present in the STX18 knockdown state in Fig4J, this does not seem to be consistent with the LD content suppression seen in other panels following STX18 KD?

ATG14 seems to be required for the STX18 KO/KD induced suppression of LD content (Fig4 O and P), and a LIR competent version of ATG14 is needed to suppress LD levels.

Q16) It lacks essential data to directly support that STX18 completely bound to CCD of ATG14

Note10) For the effect of Beclin on LDs function, the authors should include the ATG14 KD and both of ATG14 and Beclin KD to demonstrate the essence of ATG14 or Beclin.

Q17) The Supplementary FigS4C IP is done with HA beads, not myc. Correct the labelling?

Q18) In Figure S4F, Beclin1 should be detected as a positive control since CCD domain is critical for the binding between ATG14 and Beclin1.

Q19) In Figure S4I, authors should also investigate if autophagic flux was influenced by STX18 in the ATG14 KO MEF cells.

Fig5 seems to show that a LD localized anti-viral protein Viperin is degraded following these STX18 manipulations, which are potentially just an extension of the central findings that STX18 loss induces lipophagy. A potentially interesting connection is made with SARS-CoV-2 viral infection in Fig 6. They find that the M-protein of SARS-CoV-2, and viral infection itself, subverts the STX18/ATG14 interaction thereby promoting lipophagy and degradation of the anti-viral protein Viperin. In this model then, the lipophagy promoting activity of disrupting STX18/ATG14 seems to promote viral infection itself.

Q20) Could the authors show immunofluorescence images indicating the coincidence of LDs and Viperin with LC3 in the cells manipulated with ATG14 LIR mutant or the STX18 mutant (with ATG14 binding deficiency)?

Q22) Is STX18 (a.a.71-80) critical for the binding of M protein? Would it be helpful to define which domain located in the M protein is important for the binding of STX18 reciprocally?

Q23) Can the authors provided a marker indicating VSV or SARS-CoV-2 infection?

Q24) In Figure 4O, Figure 6C, Figure 6E and Figure S3K, there was no control group?

Suggestion: A working model for this study would be helpful for understanding the mechanism the authors propose.

We are deeply grateful for your great efforts and constructive comments towards our manuscript (NCOMMS-23-14908-T) titled “ATG14 Targets Lipid Droplets and Acts as An Autophagic Receptor for Syntaxin18 Regulated Lipid Droplet Turnover”. The revised manuscript and the detailed responses follow here. Please note that the revised text is highlighted in blue.

Several changes have been included in the revised manuscript when we prepared the source data:

1) In revised Fig. s8e and f, HEK293T cells were transfected with M-HA (not M-Flag) and protein interactions were detected by immunoprecipitation with anti-HA beads. In revised Fig. s8e, for GFP-ATG16 and GFP-DFCP1 panels, we used the wrong graphs carelessly. We have corrected these issues (source data provided).

2) In revised Fig. 3i, STX18-Flag input and IP panels were carelessly swapped, we have corrected this issue (source data provided).

3) In revised Fig. s3a, we can't find the original GAPDH source data, so we replaced with repeated result (source data provided).

4) In revised Fig. s4e, we have corrected the wrong marks and replaced panels with different exposure times (source data provided).

5) In revised Fig. s5k, we can't find the original STX18 and Tubulin source data, so we replaced with repeated result (source data provided).

6) In revised Fig. s5l, we can't find the original p62 and Tubulin source data, so we replaced with repeated result (source data provided).

Many thanks for your attentions.

Point-by-point response:

REVIEWER COMMENTS

Reviewer #1 (Remarks to the Author):

The authors report ATG14 as a receptor for Syntaxin18 mediated lipid droplet turnover. They identify two regions in ATG14 required for the recruitment of ATG14 to lipid droplets and they identify STX18 as a novel ATG14 interactor, whose depletion induces ATG14 dependent lipophagy and degradation of lipid droplet associated proteins including viperin. The manuscript contains interesting new data concerning the role of ATG14 and STX18 in lipophagy and adds to our understanding of the process. The physiological role of the molecular interactions described here remains unaddressed.

While most of the experimental results are convincing, addressing the points listed below would be required before publication can be recommended. Addressing point 4 is absolutely essential.

Response: Many thanks for your valuable and positive comments and conclusion of our work. We sincerely appreciated for the critical comment#4. After double checking the structure of ATG14 in AlphaFold, we agree with the reviewer that the amphipathic helix 303-320aa are internal residues that should not be available for the interaction with LDs. But our data indeed showed that point mutations on the hydrophobic interfaces or deletion of the amphipathic helix 303-320aa abolished the ATG14 LD-localization (**Revised Manuscript Figs. 1h and S1f, g**).

Revised Manuscript Fig. 1h HeLa cells expressing mCherry-ATG14 Δ C10aa or mCherry-ATG14 Δ 303-320aa were treated with 200 μ M OA for 12 h. LDs were labeled with BODIPY-493/503 (green). Cells were imaged by confocal microscopy. Scar bar represents 10 μ m.

Revised Manuscript Figs. S1f, g (f) HeLa cells expressing indicated mCherry-ATG14 mutants were treated with 200 μ M OA for 12 h, then fixed and labeled the LDs with BODIPY-493/503 (green). Cells were imaged by confocal

microscopy. Scar bar represents 10 μm . (g) The number of colocalization of ATG14/LDs per cell in (f) was counted from 20 cells. Three independent experiments. Two-tailed Unpaired Student's t-test.

We currently do not fully understand why internal amphipathic helix is critical for ATG14 LD-localization. In mock-treated cells, ATG14 is not localized on LDs and OA treatment induces ATG14 translocation to LDs (**Revised Manuscript Fig. 1a**). We proposed that potential post-translational modification on ATG14 and/or chaperone protein(s) binding ATG14, lead to structure change of ATG14. Unfortunately, we only have several very preliminary data now. We have discussed this limitation in the revised manuscript: “*In addition to the BATS domain of ATG14, our data indeed showed that the amphipathic helix 303-320 amino acids, although according to AlphaFold, are internal residues that should not be available for the interaction with LDs, were important for LD localization. Point mutations on the hydrophobic interfaces or deletion of the amphipathic helix abolished the ATG14 LD-localization (Fig. 1h and Supplementary Fig. 1f). It is unclear why internal amphipathic helix is critical for ATG14 LD-localization. Post-translational modification on ATG14 and/or chaperone protein(s) binding ATG14 may lead to structure change of ATG14, resulting in internal amphipathic helix contact LDs. More evidences are required to investigate the potential mechanism(s) by which ATG14 targets LDs.*” (**Revised Manuscript lines 447-456**).

We agree with reviewers' comments that BATS domain-mediated ATG14 localization is a more convincing component. To further confirm that BATS domain of ATG14 is required for LD localization, we examined the LD localization of mutants ATG14 ΔC10aa (defect in BATS), and found that ATG14 ΔC10aa failed to localize on LDs (**Revised Manuscript Fig. 1h**). Additionally, an LD-resident ATG14^{WFY-RRR} was engineered through substitution of the ATG14^{WFY-RRR} (defect in BATS) signal peptide with the LD-targeting amphipathic helices (LD-ATG14^{WFY-RRR}). LD-ATG14^{WFY-RRR} re-localized on LDs and rescued the decreased LD number in *Atg14* KO cells, but not ATG14^{WFY-RRR} (**Revised Manuscript Fig. S2g**).

Revised Manuscript Figs. S2g *Atg14* knockout HeLa cells expressing mCherry-ATG14^{WFY-RRR} or mCherry-LD-ATG14^{WFY-RRR} were treated with 200 μM OA for 6 h, then fixed and labeled the LDs with BODIPY-493/503 (green). The nuclei were stained with DAPI. Cells were imaged by confocal microscopy. Scar bar represents 10 μm . Number (n=25) and total area (n=20) of LDs in each cell was counted from three independent experiments. Two-tailed Unpaired Student's t-test.

1. I do not understand line 94 and line 100 ‘... ATG14 only targets to new biogenesis LDs’.

Response: We apologize for incorrect description. GPAT4^{152–208} is a membrane marker for nascent LDs, we have deleted this sentence in the revised manuscript.

2. Fig1E: In ATG14 samples, it is not clear if there is indeed specific enrichment of gold particles on lipid droplets or whether gold particles are spread all over the sample. In GFP controls no lipid droplets are visible at all. Therefore neither the negative control nor the experimental sample are convincing.

Response: Thank you for your insightful comments. We have added the new negative control in which GFP gold particles were not localized on LDs, and new experimental samples in which GFP-ATG14 gold particles were associated with LD membranes (**Revised Manuscript Figs. 1e and S1a**).

Revised Manuscript Fig. 1e. Representative immune-gold TEM images of cells expressed with GFP-ATG14 and treated with OA for 12 h. Blue arrows mark the gold particles of GFP-ATG14. LD, lipid droplets.

Revised Manuscript Fig. S1a. Representative immune-gold TEM images of cells expressed with GFP-ATG14 and treated with OA for 12 h. Blue arrows mark the gold particles of GFP-ATG14. LD, lipid droplets.

3. Line 117: Pls indicate amino acid numbers for WFY motif

Response: Thank you for your helpful suggestions. We have added this information in the revised manuscript (**Revised Manuscript line 113**).

4. Line 108: I303, L317, I320 mutant. According to AlphaFold, I303 and L317 are internal residues that are not accessible from the surface of the domain. These residues are therefore unlikely to be available for the interaction with lipid droplets, which is difficult to reconcile with the authors' claim of a receptor function for ATG14 via said residues. The authors should check that the mutant protein is still functional in non-lipid droplet related tasks.

Response: We are grateful for the reviewer's insightful and critical comments. After double checking the structure of ATG14 in AlphaFold, we agree with the reviewer that the amphipathic helix 303-320aa are internal residues that should not be available for the interaction with LDs. But our data did indicate that point mutations on the hydrophobic interfaces or deletion of the amphipathic helix 303-320aa abolished the ATG14 LD-localization (**Revised Manuscript Figs. 1h and S1f**).

We currently do not fully understand why internal amphipathic helix is critical for ATG14 LD-localization. In mock-treated cells, ATG14 is not localized on LDs and OA treatment induces ATG14 translocation to LDs (**Revised Manuscript Fig. 1a**). We proposed that potential post-translational modification on ATG14 and/or chaperone protein(s) binding ATG14, lead to structure change of ATG14. Unfortunately, we only have several very preliminary data now. We have discussed this limitation in the revised manuscript: *"In addition to the BATS domain of ATG14, our data indeed showed that the amphipathic helix 303-320 amino acids, although according to AlphaFold, are internal residues that should not be available for the interaction with LDs, were important for LD localization. Point mutations on the hydrophobic interfaces or deletion of the amphipathic helix abolished the ATG14 LD-localization (Fig. 1h and Supplementary Fig. 1f). It is unclear why internal amphipathic helix is critical for ATG14 LD-localization. Post-translational modification on ATG14 and/or chaperone protein(s) binding ATG14 may lead to structure change of ATG14, resulting in internal amphipathic helix contact LDs. More evidences are required to investigate the potential mechanism(s) by which ATG14 targets LDs."* (**Revised Manuscript lines 447-456**).

Following the suggestion from the reviewer, we further confirmed that the mutant ATG14^{ILI-QQQ} is still functional in non-lipid droplet related tasks (dimerizes, interacts with Beclin1, STX17 and LC3) (**Revised Manuscript Fig. S2c, d**). Additionally, we further confirmed that the mutant ATG14^{WFY-RRR} is still functional in non-lipid droplet related tasks (dimerizes, interacts with Beclin1, STX17 and LC3) (**Revised Manuscript Fig. S2a, b**).

Revised Manuscript Fig. 1h HeLa cells expressing mCherry-ATG14 Δ C10aa or mCherry-ATG14 Δ 303-320aa were treated with 200 μ M OA for 12 h. LDs were labeled with BODIPY-493/503 (green). Cells were imaged by confocal microscopy. Scar bar represents 10 μ m.

Revised Manuscript Figs. S1f, g (f) HeLa cells expressing indicated mCherry-ATG14 mutants were treated with 200 μ M OA for 12 h, then fixed and labeled the LDs with BODIPY-493/503 (green). Cells were imaged by confocal microscopy. Scar bar represents 10 μ m. (g) The number of colocalization of ATG14/LDs per cell in (f) was counted from 20 cells. Three independent experiments. Two-tailed Unpaired Student's t-test.

Revised Manuscript Figs. S2a-d (a) The Flag tagged ATG14 or ATG14^{WFY-RRR} was co-expressed in HEK293T cells with GFP-LC3C, STX17-Myc and Beclin1-HA. Protein interactions were detected by immunoprecipitation with anti-Flag beads and immunoblotting analysis. (b) HEK293T cells were transfected with ATG14-Flag or ATG14^{WFY-RRR}-Flag for 36 h and treated with 0.2 mM DSS for 30 min before collecting. Cell lysates were analyzed via western blot. (c) The GFP tagged ATG14 or ATG14^{IL1-QQQ} was co-expressed in HEK293T cells with Flag-LC3C, STX17-Myc and Beclin1-HA. Protein interactions were detected by immunoprecipitation with anti-GFP beads and immunoblotting analysis. (d) HEK293T cells were transfected with ATG14-Flag or ATG14^{IL1-QQQ}-Flag for 36 h and treated with 0.2 mM DSS for 30 min before collecting. Cell lysates were analyzed via western blot.

5. Can the authors identify a minimal domain / an amphipathic helix in ATG14 that on its own targets lipid droplets?

Response: Thank you for your helpful suggestions. We found that deletion of the amphipathic helix 303-320aa abolished the ATG14 LD-localization (**Revised Manuscript Fig. 1h**).

Revised Manuscript Fig. 1h HeLa cells expressing mCherry-ATG14^{ΔC10aa} or mCherry-ATG14^{Δ303-320aa} were treated with 200 μM OA for 12 h. LDs were labeled with BODIPY-493/503 (green). Cells were imaged by confocal microscopy. Scar bar represents 10 μm.

6. Fig2A: Authors report binding of ATG14 to several LC3/GABARAPs and failure to bind LC3B and GABARAPL1. Results are inconsistent with ref 24. GST-LC3B in Fig2A should be a single band, as for the other proteins, but appears as multiple species. Positive control for the functionality of LC3B and other proteins in this assay is needed.

Response: Thank you for your helpful comments. We have generated a new GST-LC3B expression construct and performed the *in vitro* GST pull-down assay. Our data indicate that ATG14 interacts with LC3A, LC3C, GABARAP, and GABARAPL2, but failed to bind LC3B and GABARAPL1 (**Revised Manuscript Fig. 2a**).

Revised Manuscript Fig. 2a Purified GST tagged ATG8s were incubated with His-sumo tagged ATG14, and analysis the interactions by GST pull-down.

We also confirmed the direct interaction between GST-LC3B and His-p62 via *in vitro* pull-down assay (**Response letter Fig. 1**).

Response letter Fig. 1 Purified GST tagged LC3B were incubated with His tagged p62, and analysis the interactions by GST pull-down.

7. Fig2C: Selective interaction of ATG14 with LC3ii is remarkable. A control LIR-containing protein should be used as a control.

Response: Thank you for positive comments and instructive suggestion. We have shown that ATG14^{LIR^m} failed to bind LC3-II in a co-IP assay (**Revised Manuscript Fig. S2k**).

k

Revised Manuscript Fig. S2k The Flag tagged ATG14 or ATG14^{LIRm} was expressed in HEK293T cells. Protein interactions between ATG14-Flag and endogenous LC3 were detected by immunoprecipitation with anti-Flag beads and immunoblotting analysis.

8. Line 170: Please explain the origin of the LC-targeting amphipathic helices.

Response: We apologize for this missing information. The origin of the LD-targeting amphipathic helices is from SARS-CoV-2 ORF6 protein, which were identified by our previous study (PMID: 37218505). We have added this information in the revised manuscript (**Lines 153-154**).

9. FigS2: 2A quantification of STX18 on lipid droplets missing. Controls for 2B missing.

Response: Thank you for your helpful suggestions. We have added the GFP control and the quantification of STX18 on LDs (**Revised Manuscript Fig S3b**). For controls of original Fig S2b in immune-TEM assay, GFP controls are in **revised Fig. 1e**.

Revised Manuscript Fig S3b HeLa cells expressing GFP or GFP-STX18 were treated with 200 μ M OA for 12 h, then fixed and labeled the LDs with LipidTOX Red (red). The nuclei were stained with DAPI. Cells were imaged by confocal microscopy. Scar bar represents 10 μ m. Colocalization of LDs with GFP (20 cells) or GFP-STX18 (20 cells) (Pearson's Coefficient) were analyzed, three independent experiments. Two-tailed Unpaired Student's t-test.

10. Line 277: Authors report that siSTX18 in PentoKO cells reduces number of LD. They conclude

degradation via lipophagy. Pos evidence of autophagy involvement required, for example depletion of essential ATG other than ATG14/VPS34 complex such as ULK complex, ATG16 complex

Response: We appreciate this critical advice. Following the suggestion from the reviewer, we further used *Fip200* (the core unit of ULK complex) KO MEF cells and *Atg5* (the core unit of ATG16 complex) KO MEF cells for lipophagy assay. We confirmed that knockdown of STX18 induced lipophagy dependent on ULK complex and ATG16 complex (**Revised Manuscript Fig. S6d**).

Revised Manuscript Fig. S6d WT, *Fip200* and *Atg5* KO MEF cells were transfected with si-STX18 for 48 h and treated with 200 μM OA for 6 h. LDs were labeled with BODIPY-493/503 (green). The nuclei were stained with DAPI. Cells were imaged by confocal microscopy. Scar bar represents 10 μm. Number of LDs in each cell was counted from 50 cells of three independent experiments. Two-tailed Unpaired Student's t-test.

11. Antiviral effects rely largely on overexpression experiments.

Response: Thank you for your valuable suggestions. In original manuscript, we re-expressed ATG14 wild-type and LIR^m in *Atg14* KO cells and found that rescued with wild-type ATG14 but not ATG14^{LIR^m}, led to a reduction in Viperin expression, facilitating viral production (**Original Manuscript Fig. 6q**).

We have shown that deletion of STX18 induced Viperin degradation (**Original Manuscript Fig. 5a, b, k, n**) dependent on ATG7 (**Original Manuscript Fig. 5b**) and ATG14 (**Original Manuscript Fig. 5o**).

Furthermore, we confirmed that knockdown of ATG14 inhibited SARS-CoV-2 replication (**Revised Manuscript Fig. S8p**).

p

Revised Manuscript Fig. S8p Vero-E6 cells were transfected with indicated siRNAs for 24 h and infected with SARS-CoV-2 for 24 h. Meanwhile cells were treated with 200 μ M OA for 12 h. Cell lysates were analyzed via western blot. Viral RNA level was determined by RT-qPCR, three independent replicates. Two-tailed Unpaired Student's t-test.

12. Overexpression of ATG14 affects viperin and viral replication but depletion of endogenous proteins, including STX18, has only minimal effects. Is lipophagy of anti-viral restriction factors of physiological relevance?

Response: Thank you for your insightful comments. We apologize for unclear elaboration. We have shown that: STX18 as a new lipophagy negative regulator that STX18 expression decreases the interactions of ATG14 with ATG8s and disrupts the formation of PI3KC3-C1 complex by targeting ATG14. SARS-CoV-2 M protein hijacks STX18 and subverts the interaction of STX18 with ATG14, resulting in the induction in ATG14-mediated lipophagy to degrade Viperin to evade the anti-viral effect, indicating that in SARS-CoV-2 infected cells, the function of STX18 in lipophagy is inhibited by M protein. Overexpression of STX18 inhibited SARS-CoV-2-induced lipophagy and Viperin degradation, decreasing the viral replication (**Original Manuscript Fig. 6p**).

Furthermore, we confirmed that knockdown of ATG14 inhibited SARS-CoV-2 replication (**Revised Manuscript Fig. S8p**).

In the physiological condition, SARS-CoV-2 hijacks STX18 by viral protein M, and subverts the interaction of STX18 with ATG14, promotes the formation of PI3KC3-C1 complex and ATG14-LC3 interaction, resulting in the induction in ATG14-mediated lipophagy to degrade Viperin to evade the anti-viral effect. (**Revised Manuscript lines 421-425**).

Reviewer #2 (Remarks to the Author):

Review: Nat Comm 'ATG14 Targets Lipid Droplets and Acts as An Autophagic Receptor for Syntaxin18 Regulated Lipid Droplet Turnover'

General comments :

This manuscript suggests that the lipophagy promoting activity of ATG14 is via direct recruitment of LC3 positive membranes to LDs, an activity which is negatively impacted by interactions with the

SNARE protein STX18. An interesting functional role for these proteins is tied to a SARS-CoV-2 infection where the M-protein seems to disrupt this interaction, thereby promoting viral infection through turnover of the LD resident, anti-viral protein Viperin. Most of the data is fairly convincing, some specific concerns are noted below. Conceptually, it would be better to clearly show that the loss of STX18 has no direct impact on LD synthesis/expansion as put forth by the Li laboratory (Xu et al, JCB 2018) and instead is a driver of autophagic/lysosome-based turnover of LD content. The figures concerning viral infection and the abrogation of the ATG14/STX18 nexus for controlling lipophagy seems interesting, though almost a part of a different story?

Response: Many thanks for your conclusion of our work. Your constructive suggestions help us significantly improve the quality of this manuscript.

Summary and Specific Suggestions :

Fig1 shows ATG14 localization to LDs, which has previously been shown (Pfisterer et al, JLR 2014) as referenced. The authors do attempt to map out the amphipathic helix (Fig 1F,G) required for targeting ATG14 to LDs, though the images presented are not conclusive. The authors demonstrate that the BATS domain (membrane curvature sensing domain) of ATG14 is required for LD localization, which is a more convincing component of this image panel (1G).

Response: Thank you for your important comments and suggestions. We agree that BATS domain-mediated ATG14 localization is a more convincing component. To further confirm that BATS domain of ATG14 is required for LD localization, we examined the LD localization of mutants ATG14 Δ C10aa (defect in BATS), and found that ATG14 Δ C10aa failed to localize on LDs (**Revised Manuscript Fig. 1h**). Additionally, an LD-resident ATG14^{WFY-RRR} was engineered through substitution of the ATG14^{WFY-RRR} (defect in BATS) signal peptide with the LD-targeting amphipathic helices (LD-ATG14^{WFY-RRR}). LD-ATG14^{WFY-RRR} re-localized on LDs and rescued the decreased LD number in *Atg14* KO cells, but not ATG14^{WFY-RRR} (**Revised Manuscript Fig. S2g**).

Revised Manuscript Fig. 1h HeLa cells expressing mCherry-ATG14 Δ C10aa or mCherry-ATG14 Δ 303-320aa were treated with 200 μ M OA for 12 h. LDs were labeled with BODIPY-493/503 (green). Cells were imaged by confocal microscopy. Scar bar represents 10 μ m.

Revised Manuscript Figs. S2g *Atg14* knockout HeLa cells expressing mCherry-ATG14^{WFY-RRR} or mCherry-LD-ATG14^{WFY-RRR} were treated with 200 μM OA for 6 h, then fixed and labeled the LDs with BODIPY-493/503 (green). The nuclei were stained with DAPI. Cells were imaged by confocal microscopy. Scar bar represents 10 μm. Number (n=25) and total area (n=20) of LDs in each cell was counted from three independent experiments. Two-tailed Unpaired Student's t-test.

Q1) The images in 1G are not convincing to show a lack of LD localization for the amphipathic helix mutant (ILI-QQQ), perhaps improve the magnification or clarify with additional examples?

Response: Thank you for your valuable suggestions. We have replaced with additional examples (**Response letter Fig. 2**).

Additionally, in revised manuscript, we generated mCherry tag ATG14 and more point mutations were introduced into the hydrophobic interfaces of each helix: AH1, L109Q/I127Q (ATG14^{LI-QQ}), I113Q/I120Q (ATG14^{II-QQ}) or I113Q/I120Q/L123Q (ATG14^{III-QQQ}), or AH2, I303Q, L317Q and I320Q (ATG14^{ILI-QQQ}) to break their structures (**Revised Manuscript Fig. S1e**). Strikingly, ATG14^{LI-QQ}, ATG14^{II-QQ}, and ATG14^{III-QQQ} were still localized to LDs, whereas ATG14^{ILI-QQQ} failed to associate with LDs (**Revised Manuscript Fig. S1f, g**). Furthermore, deletion of the second amphipathic helices (303-320aa) also abolished LD localization (**Revised Manuscript Fig. 1h and S1g**), indicating that the binding of ATG14 to LDs is dependent on the second amphipathic helices.

Response letter Fig. 2

Revised Manuscript Figs. S1e-g (e) Schematic of the domain structure of ATG14 and truncation mutants used in this study. (f) HeLa cells expressing indicated mCherry-ATG14 mutants were treated with 200 μ M OA for 12 h, then fixed and labeled the LDs with BODIPY-493/503 (green). Cells were imaged by confocal microscopy. Scar bar represents 10 μ m. (g) The number of colocalization of ATG14/LDs per cell in (f) was counted from 20 cells. Three independent experiments. Two-tailed Unpaired Student's t-test.

Revised Manuscript Fig. 1h HeLa cells expressing mCherry-ATG14 Δ C10aa or mCherry-ATG14 Δ 303-320aa were treated with 200 μ M OA for 12 h. LDs were labeled with BODIPY-493/503 (green). Cells were imaged by confocal microscopy. Scar bar represents 10 μ m.

A knockdown of ATG14 is associated with an increase in LD content of HeLa cells, consistent with previously published findings by Xiong et al in-liver tissue (JBC 2012) and Pfisterer et al in cell lines (JLR 2014). When overexpressed, ATG14 suppresses LD content when localized properly, which was reversed by chloroquine, indicating a likely role in promoting lipophagy (1J, K).

Note1) The authors try to use Atglistatin in HeLa cells, which to my knowledge has only been successful in murine cells. They should perhaps remove this data and not make claims related to lipolysis in this context.

Response: We appreciate this helpful advice. As your suggestion, we have removed the Atglistatin treatment data.

Note2) It will be more convincing if more autophagy inhibitors were utilized to solidify the findings, at least include Bafilomycin A1.

Response: Thank you for your valuable suggestions. We have examined the effect of Baf-A1 on ATG14-induced lipophagy and found that Baf-A1 treatment blocked ATG14-induced lipophagy (Revised Manuscript Fig. 1j-l).

Revised Manuscript Fig. 1j-l (j) HeLa cells expressing mCherry-ATG14 or its mutants were treated with 200 μ M OA for 6 h. Cells expressing the mCherry-ATG14 were also treated with 100 μ M CQ for 6 h or 400 nM Baf-A1 for 4 h. LDs were labeled with BODIPY-493/503 (green). The nuclei were stained with DAPI. Yellow ROIs indicate cells expressing ATG14. (k) Number of LDs in each cell in (j) was counted from 50 cells of three independent experiments. Two-tailed Unpaired Student's t-test. (l) The concentration of triglyceride in HeLa cells was analyzed according to manufacturer's instructions. Error bars, mean \pm SD of three independent experiments. Two-tailed Unpaired Student's t-test.

Note3) As the authors suggest that ATG14 does not localize to nascent LDs, it is hard to understand why in the steady state (without OA) ATG14 KD could increase the abundance of LDs (Figure 1I). Also, it seems necessary to show the efficiency of ATG14 KD by WB or RT-qPCR.

Response: Thank you for your insightful comments and suggestions. ATG14 is known as the core unit of the PI3KC3-C1 complex and binds STX17-SNAP29 complex to promote autophagosome-lysosome fusion. ATG14 KD caused the defect of autophagy initiation and autophagosome-lysosome fusion, so in the steady state, LDs were accumulated due to the defect of basal autophagic flux (initiation and fusion) in ATG14 KD cells.

As your suggestion, we added the WB result to show ATG14 KD efficiency (**Revised Manuscript Fig. 1i**).

Revised Manuscript Fig. 1i HeLa cells were transfected with or without ATG14 siRNA for 48 h. Cell lysates were analyzed via western blot.

Note4) Besides Figure 1D, the authors should measure and compare the enrichment of ATG14 in cellular fractions with or without OA.

Response: We are grateful for the Reviewer's important suggestions. Using confocal microscopy, we found that ATG14 had no obvious colocalization with mitochondrial and Golgi while partly localized with lysosomes and the ER with or without OA treatment (**Revised Manuscript Fig. s1d**).

Revised Manuscript Fig. s1d Co-localizations between mCherry-ATG14 and mitochondria (anti-Tom20), or Lysosomes (anti-LAMP2) or Golgi (anti-GM130) or ER (GFP-RAMP4) were analyzed via confocal microscopy. Scar bar represents 10 μ m.

Q2) It could be interesting to test the interaction between ATG14 and LDs if I113, I120, and/or L123 in ATG14 are/is mutated?

Response: We appreciate this helpful advice. We generated mCherry tag ATG14 and more point mutations were introduced into the hydrophobic interfaces of each helix: AH1, L109Q/I127Q (ATG14^{LI-QQ}), I113Q/I120Q (ATG14^{II-QQ}) or I113Q/I120Q/L123Q (ATG14^{III-QQQ}), or AH2, I303Q, L317Q and I320Q (ATG14^{III-QQQ}) to break their structures. Strikingly, ATG14^{LI-QQ}, ATG14^{II-QQ}, and ATG14^{III-QQQ} were still localized to LDs, whereas ATG14^{III-QQQ} failed to associate with LDs (**Revised Manuscript Fig. s1f, g**).

Revised Manuscript Figs. S1f, g (f) HeLa cells expressing indicated mCherry-ATG14 mutants were treated with 200 μ M OA for 12 h, then fixed and labeled the LDs with BODIPY-493/503 (green). Cells were imaged by confocal microscopy. Scale bar represents 10 μ m. (g) The number of colocalization of ATG14/LDs per cell in (f) was counted from 20 cells. Three independent experiments. Two-tailed Unpaired Student's t-test.

Note5) I suggest including a control group of ATG14 WT in Figure 1g to show localization differences.

Response: Many thanks for your helpful suggestions. We have added GFP alone and GFP-ATG14 as control groups to show localization differences (**Revised Manuscript Fig. 1f, g**).

Revised Manuscript Fig. 1f, g (f) HeLa cells expressing GFP, GFP-ATG14, or GFP-ATG14^{WFY-RRR} were treated with 200 μ M OA for 12 h. LDs were labeled with LipidTOX Red (red). Cells were imaged by confocal microscopy. Scale bar represents 10 μ m. (g) The number of colocalization of GFP, GFP-ATG14, or GFP-ATG14^{WFY-RRR} with LDs per cell in (f) was counted from 20 cells. Three independent experiments. Two-tailed Unpaired Student's t-test.

In FigS1 the Authors find ATG14 did not localize to nascent LDs, showing modest colocalization between GPAT4(152–208) and ATG14, a membrane marker for nascent LDs.

Note6) It would be good to show ATG14 colocalization with late LD markers such as HSD17B11, Ldsdh1 as shown in Song J et al., ncb 2022.

Response: Thank you for your important suggestions. Ldsdh1 is a *Drosophila* gene, so we examined the colocalization of HSD17B11-ATG14-LDs with OA treatment. Interestingly, we found that ATG14 colocalized with HSD17B11 which were LD-positive (**Revised Manuscript Fig. s1c**).

Revised Manuscript Fig s1c HeLa cells expressing HSD17B11-GFP and mCherry-ATG14 were treated with 200 μ M OA for 12 h, then fixed and labeled the LDs with LipidTOX Deep Red (blue). Cells were imaged by confocal microscopy. Scar bar represents 10 μ m.

Q3) This figure needs a control group which was not treated with DSS in Figure S1e?

Response: Thank you for your helpful suggestions. We have added the control group which was not treated with DSS (**Revised Manuscript Fig s2m**).

Revised Manuscript Fig. s2m HEK293T cells were transfected with ATG14-Flag or ATG14^{LIRm}-Flag for 36 h and treated with 0.2 mM DSS for 30 min before collecting. Cell lysates were analyzed via western blot.

Figure 2 explores the direct association of ATG14 with core autophagic machinery such as LC3 and GABARAP, which has been reported by other groups. They show a clear association with LC3A, LC3C and GABARAP/L2 and do define the LIR. The authors show that there is a requirement of LIR bearing ATG14 to recruit LC3 membranes to LDs in panel 2L, further support is gained from KO/re-expression studies as well.

Q4) It is hard to appreciate exactly how many cells were scored from the immunofluorescence panels presented in Fig2 L/M. Was this listed in the manuscript text or figure legend? Were these results displayed in panel 2M obtained from one experimental trial?

Response: Thank you for your insightful comments. As we mentioned in the original manuscript (**Original manuscript lines 1016-1017**): “*The number of colocalization of ATG14/LC3/LDs per cell in (L) was counted from 50 cells. Three independent experiments. Two-tailed Unpaired Student’s t-test.*” In the revised manuscript, we have added the quantification information in figure legend.

Q5) The localization of LC3/ATG14 on LDs of serum starved HeLa cells is very graphic in panel 2L, would the authors be able to provide some additional images or increased magnifications?

Response: Thank you for your helpful comments. Here we provided some additional images (**Response letter Fig. 3**)

Response letter Fig. 3

Q6) Do the HeLa cells in panel 2L have increased labeling of LC3 following starvation? There certainly seems to be quite a high level of this protein in non-starved cells. How specific is the LC3 antibody labeling in 2L? Are the large red structures representative of autophagic vacuoles or mischaracterized? What do these LDs and autophagic membranes look like by EM?

Response: Thank you for your insightful comments. As we mentioned in the original manuscript (**Original manuscript lines 1044-1047**): “*HeLa cells expressing the GFP-ATG14 or GFP-ATG14^{LIRm} were treated with 200 μM OA for 12 h and treated with or without serum starvation for 24 h. Meanwhile, 100 μM CQ was added for 6 h before cells were fixed and immunostained with anti-LC3 (red). LDs were labeled with LipidTOX Deep Red (blue).*” CQ treatment led to accumulation of LC3 dots.

We used Rabbit anti-LC3 (PM036) (**Original manuscript line 571**) from MBL for immunostaining, this antibody has been used for imaging assay in previous study (PMID: 32989250, fig4b, 4e). We found that under CQ treatment, large red structures appeared in some cells, and similar large LC3 structures were also observed by using this antibody in a previous study (PMID: 33590792, Fig1A).

We appreciate reviewer’s advice that we could use CLEM. Unfortunately, we failed to obtain the data. We have replaced with new pictures in which the size of LC3 structures were regular (**Revised Manuscript Fig. 2I**).

Revised Manuscript Fig. 2l HeLa cells expressing the GFP-ATG14 or GFP-ATG14LIRm were treated with or without 200 μ M OA for 12 h and treated with or without serum starvation for 24 h. Meanwhile, 100 μ M CQ was added for 6 h before cells were fixed and immunostained with anti-LC3 (red). LDs were labeled with LipidTOX Deep Red (blue). Cells were imaged by confocal microscopy. Scar bar represents 10 μ m.

In this context it would be good if authors provide control/mock group in Figure 2e.

Response: Thank you for your valuable suggestions. We have added the control group (**Revised Manuscript Fig. 2e, f**).

Revised Manuscript Fig. 2e, f (e) HeLa cells were transfected with ATG14 siRNA for 36 h and treated with or without serum starvation for 24 h. Meanwhile, cells were treated with or without 200 μ M OA for 12 h and with or

without 100 μM CQ for 6 h before cells were fixed and immunostained with anti-LC3 antibodies (red). LDs were labeled with LipidTOX Deep Red (blue). Cells were imaged by confocal microscopy. Scar bar represents 10 μm . (f) The number of colocalization between LC3 and LDs per cell in (e) was counted from 25 cells. Three independent experiments. Two-tailed Unpaired Student's t-test.

Q7) It is impressive to see the interaction between ATG14 and LC3. Have the authors tried to see these interactions in endogenous context with or without OA treatment?

Response: Thank you for your valuable suggestions. We have showed that OA treatment has no effect on ATG14-LC3 interactions (**Revised Manuscript Fig. S2j**).

Revised Manuscript Fig. S2j The Flag tagged ATG14 was expressed in HEK293T cells and cells were further treated with or without 200 μM OA for 12 h. Protein interactions between ATG14-Flag and endogenous LC3 were detected by immunoprecipitation with anti-Flag beads and immunoblotting analysis.

Q8) In Fig2 panels N,O the authors show no significant look at LD number differences between experimental groups. Could they include an analysis of total LD area/cell? Would the authors be able to provide some additional images or increased magnifications?

Response: Thank you for your insightful comments and suggestions. We have included an analysis of total LD area/cell for Fig 2n (**Revised Manuscript Fig. S2n**). Additionally, we also provide the analysis of total LD area/cell for

Revised Manuscript Fig. S2n Total area of LDs in each cell in (Fig. 2n) was counted from 50 cells of three independent experiments. Two-tailed Unpaired Student's t-test.

As your requested, here we provided some additional images for *Atg14* KO cells rescued with mCherry-ATG14 (**Response letter Fig. 4**). Our data clearly showed that wild-type ATG14 could rescue the decreased LD number in *Atg14* KO cells.

Response letter Fig. 4 Yellow ROIs indicate the cells expressing mCherry-ATG14 and white ROIs indicate the cells without or low mCherry-ATG14 expression as control group.

Fig 3: Mass spec.-based identification of ATG14 binding partners that change following serum starvation are shown in Fig3. A large decrease in STX18 binding is seen following serum starvation, which itself has been found on the LD and seems to influence LD size and TG amounts by others. Interestingly, the authors show a dramatic level of ATG14/LC3 localization on LDs in STX18 KO cells (Fig 3H).

Response: Thanks for your comments. For mass spectrometry raw data related to Fig3A and Fig5A, we prefer not to share these informations at this stage, because several important projects based on these mass spectrometry data are ongoing (KO mice are just ready). It's nature's policy to share the mass spectrometry raw data, so we have discussed with editor and she agreed that we can delete mass spectrometry raw data when we submit our original manuscript.

Q9) Is the STX18 KO condition associated with increased levels of autophagy? There is so much LC3 labeling in these KO cells, perhaps the state of autophagic flux should be addressed? Are these

LC3 membranes the result of impaired autophagic flux? What are the protein levels of p62 and LC3II in these STX18 KO cells?

Response: Thank you for your helpful comments and suggestions. Yes, we found that STX18 KO increased levels of autophagy.

To better examine the colocalization of LC3 with LDs, we treated cells with CQ to block degradation and accumulate LC3 labeling.

Following the suggestion from the reviewer, we found that LC3-II was accumulated and p62 was degraded in *Stx18* KO cells, suggesting that STX18 depletion induces autophagic flux (**Revised Manuscript Fig. S3f**).

Revised Manuscript Fig. S3f *Stx18* wild type and knockout HeLa cells were analyzed via immunoblotting analysis.

Note7) The authors found the STX18-ATG14 through MS. It might also be interesting to confirm the binding between these two proteins in an endogenous context, with or without OA treatment, with or without an autophagy inhibitor.

Response: We appreciate this critical advice. We analyzed the endogenous ATG14-STX18 interactions and found that OA or CQ treatment, or knockdown of Atg7 had minimal effect in an endogenous context between STX18 and ATG14 (**Revised Manuscript Fig. S3d, e**).

Revised Manuscript Fig. S3d, e (d) HEK293T cells were treated with or without OA for 12 h. Protein interactions were detected by immunoprecipitation with anti-IgG or STX18 antibodies and immunoblotting analysis. (e) HEK293T cells were transfected with or without si-Atg7, and treated with or without CQ for 6 h. Protein interactions were detected by immunoprecipitation with anti-IgG or STX18 antibodies and immunoblotting analysis.

Note8) The colocalization between ATG14 and LD should be done in a context of STX18 knockdown or in STX18 KO cells, potentially following rescue by exogenously expressed STX18.

Response: Thank you for your valuable suggestions. The colocalization between ATG14 and LD was confirmed in *Stx18* KO cells, following rescue by exogenously expressed STX18 (**Revised Manuscript Fig. S3g**).

Revised Manuscript Fig. S3g *Stx18* wild type and knockout HeLa cells co-expressing vector or STX18-Flag with mCherry-ATG14 were treated with 200 μ M OA for 12 h, then fixed and labeled the LDs with LipidTOX Deep Red (blue). Cells were imaged by confocal microscopy. Scar bar represents 10 μ m.

Note9) In Figure S3G, it was nice to show the overall protein of STX18 by using anti-STX18 antibody. It would be helpful if the authors could put a label indicating which was endogenous/exogenous.

Response: We thank reviewer for this helpful suggestion. label indicating which was endogenous/exogenous has been added.

Q10) In Figure 3I, it seems that every protein examined interacted with STX18, these findings could be strengthened by a negative result?

Response: We are grateful for the Reviewer's helpful suggestions. We found that STX18 failed to interact with LC3, and we added this data as a negative result (**Revised Manuscript Fig. 3i**).

Revised Manuscript Fig. 3i HEK293T cells were transfected with STX18-Flag for 36 h. Protein interactions were detected by immunoprecipitation with anti-Flag beads and immunoblotting analysis.

Q11) In Figure 3n,o, the quantification should go after the taken images. So, the panel number for 'n' and 'o' should be exchanged. Also, the knockdown efficiency should be detected by western blot analysis or RT-qPCR assay.

Response: Thanks for your insightful advice. We have changed the panel number and knockdown efficiency has been detected by western blot (**Revised Manuscript Fig. 3m**).

Revised Manuscript Fig. 3m HeLa cells were transfected with ATG14 or/and STX18 siRNA for 48 h. Cell lysis were analyzed via western blot.

Fig4 demonstrates that a knockdown of STX18 suppresses LD content, which seems to be dependent on autophagy as a knockdown of Atg7 results in LD retention. Consistent with this is an increased frequency of LDs found in single membrane bound autolysosome like structures, what the authors term 'lipophagosomes', in Fig 4D-F following STX18 knockdown or KO. They also use a GFP-RFP-PLIN2 reporter to show that there are more RFP only + LDs in the cells following STX18 KD, consistent with the idea that in this knockdown state there is more lipophagy occurring.

Q12) Are these structures that contain LDs indeed representative of anything related to autophagosomes? They appear to be lysosomes and are by the authors' own admission single membrane bound.

Response: Thank you for your insightful comments. We apologize for misleading elaboration. We have rephrased the sentence to classify that “we directly visualized “lipo-phagolysosomes” in which LDs were engulfed into single-membrane autolysosomes in *Stx18* KO HepG2 and *STX18* knockdown HeLa cells”.

Q13) How do these findings of *STX18* KO suppressing LD content relate to the findings of Xu et al (JCB, 2018) where this SNARE is implicated in promoting LD formation/expansion at the ER? If their model is followed, wouldn't the findings of this manuscript be consistent with a *STX18* loss reducing LD content from a synthesis aspect?

Response: Thank you for your insightful comments. In Xu et al (JCB, 2018), they suggested that *STX18* KD impaired LD biogenesis via blocking ER-LD contact in adipose cells. Our results showed that in HeLa cells *STX18* KD reduced LD content dependent on autophagic machinery: *Atg7* deletion rescued *STX18* loss reducing LD content, we further noticed that in *Atg7* deleted cells, *STX18* KD failed to induce reduction of LD number and triglyceride levels, indicating that *STX18* KD has minimal effect on LD synthesis. We have added this discussion: “We further noticed that in *Atg7* deleted cells, *STX18* depletion failed to induce reduction of LD number and triglyceride levels, indicating that *STX18* depletion has minimal effect on LD synthesis.” (**Revised Manuscript lines 490-492**).

Q14) In the image 4A, in *stx18/ ATG 7* double KD, LD number look less than control and *ATG 7* KD however graphically look similar.

Response: Thank you for your insightful comments. We have re-performed the experiment and replaced with new representative pictures (**Revised Manuscript Fig. 4a**).

Revised Manuscript Fig. 4a HeLa cells were transfected with *STX18* or/and *ATG7* siRNA for 48 h, and treated with 200 μ M OA for 6 h. LDs were labeled with BODIPY-493/503 (green). The nuclei were stained with DAPI. Cells were imaged by confocal microscopy. Scar bar represents 10 μ m.

Q15) Why are there so many LDs present in the STX18 knockdown state in Fig4J, this does not seem to be consistent with the LD content suppression seen in other panels following STX18 KD?

Response: Thank you for your insightful comments. As we mentioned in our figure legend, for RFP-GFP-Plin2 assay, we treated cells with **OA for 12 h** to induce LD biogenesis and Plin2 LD localization, and investigated LD-localized Plin2-RFP⁺GFP⁻ signals; for LD number assay, we treated cells with **OA for 6 h** and investigated LD degradation.

ATG14 seems to be required for the STX18 KO/KD induced suppression of LD content (Fig4 O and P), and a LIR competent version of ATG14 is needed to suppress LD levels.

Q16) It lacks essential data to directly support that STX18 completely bound to CCD of ATG14

Response: We thank reviewer for this critical comment. We found that STX18 binds the CCD of ATG14 (**Revised Manuscript Fig. S5h**). To confirm that STX18 disrupts the formation of PI3KC3-C1 complex by competitively binding the CCD in ATG14, we gradually increased the expression of ATG14 and found that the STX18 overexpression-inhibited Torin1-induced p62 degradation was reversed (**Revised Manuscript Fig. S5i**).

Revised Manuscript Fig. S5h, i (h) The protein interactions of GST tagged ATG14 or ATG14^{CCD} with His tagged STX18 were detected by GST pull-down experiments. (i) HEK293T cells expressing ATG14-Flag and STX18-Flag were treated with Torin 1. Cell lysates were analyzed via western blot.

Note10) For the effect of Beclin on LDs function, the authors should include the ATG14 KD and both of ATG14 and Beclin KD to demonstrate the essence of ATG14 or Beclin.

Response: We are grateful for the Reviewer's helpful suggestions. We found that besides ATG14, STX18 regulates lipophagy also dependent on Beclin1 (**Revised Manuscript Fig. S6a**).

Revised Manuscript Fig. S6a HeLa cells were transfected with indicated siRNAs for 48 h, and treated with 200 μ M OA for 6 h. LDs were labeled with BODIPY-493/503 (green). The nuclei were stained with DAPI. Cells were imaged by confocal microscopy. Scar bar represents 10 μ m. Number of LDs in each cell was counted from 50 cells of three independent experiments. Two-tailed Unpaired Student's t-test.

Q17) The Supplementary FigS4C IP is done with HA beads, not myc. Correct the labelling?

Response: We apologize for this error. This has now been corrected.

Q18) In Figure S4F, Beclin1 should be detected as a positive control since CCD domain is critical for the binding between ATG14 and Beclin1.

Response: We appreciate this critical advice. We have confirmed that CCD domain is critical for the binding between ATG14 and Beclin1 (**Revised Manuscript Fig. S5f**).

Revised Manuscript Fig. S5f HEK293T cells were transfected with Beclin1-HA and ATG14-Flag or ATG14 Δ CCD-Flag for 36 h. Protein interactions were detected by immunoprecipitation with anti-Flag beads and immunoblotting analysis.

Q19) In Figure S4I, authors should also investigate if autophagic flux was influenced by STX18 in the ATG14 KO MEF cells.

Response: Thank you for your instructive and critical suggestions. We don't have *Atg14* KO MEF cells, so we generated *Atg14* KO HeLa cells and found that STX18 depletion failed to induce p62 degradation in *Atg14* KO cells, indicating that STX18 negatively regulate autophagic flux dependent on ATG14 (**Revised Manuscript Fig. S5m**).

Revised Manuscript Fig. S5m *Atg14* wild type and knockout HeLa were treated with control or STX18 siRNA for 48 h. Cell lysates were analyzed via western blot.

Fig5 seems to show that a LD localized anti-viral protein Viperin is degraded following these STX18 manipulations, which are potentially just an extension of the central findings that STX18 loss induces lipophagy. A potentially interesting connection is made with SARS-CoV-2 viral infection in Fig 6. They find that the M-protein of SARS-CoV-2, and viral infection itself, subverts the STX18/ATG14 interaction thereby promoting lipophagy and degradation of the anti-viral protein Viperin. In this model then, the lipophagy promoting activity of disrupting STX18/ATG14 seems to promote viral infection itself.

Q20) Could the authors show immunofluorescence images indicating the coincidence of LDs and Viperin with LC3 in the cells manipulated with ATG14 LIR mutant or the STX18 mutant (with ATG14 binding deficiency)?

Response: We appreciate this critical advice. Our new data indicated that rescued with wild-type STX18, but not STX18 Δ^{71-80} (defect in ATG14 interaction) abolished the colocalization of LC3-Viperin-LDs in *Stx18* KO cells (**Revised Manuscript Fig. S7d**). In addition, obvious colocalization between LC3-Viperin-LDs was observed in ATG14-Flag expressed, but not vector or ATG14^{LIRm}-Flag expressed *Atg14* KO cells (**Revised Manuscript Fig. S7e**).

Revised Manuscript Fig. S7d, e (d) *Stx18* KO HeLa cells expressing Viperin-GFP were transfected with STX18-Flag or STX18 $\Delta^{71-80aa}$ -Flag for 24 h, cells were further treated with 200 μ M OA for 12 h and 100 μ M CQ for 6 h, then fixed and immunostained with anti-LC3 antibodies (red). LDs were labeled with LipidTOX Deep Red (blue). (e) *Atg14* KO HeLa cells expressing Viperin-GFP were transfected with ATG14-Flag or ATG14^{LIRm}-Flag for 24 h, cells were further treated with 200 μ M OA for 12 h and 100 μ M CQ for 6 h, then fixed and immunostained with anti-LC3 antibodies (red). LDs were labeled with LipidTOX Deep Red (blue).

Q22) Is STX18 (a.a.71-80) critical for the binding of M protein? Would it be helpful to define which domain located in the M protein is important for the binding of STX18 reciprocally?

Response: Thank you for your instructive and critical suggestions. Our co-IP data showed that STX18 Δ^{1-80} still interacts with M protein, suggesting that STX18 (a.a.71-80) is not critical for the binding of M protein (**Response letter Fig. 5**).

Response letter Fig. 5:

As your suggested, to map the critical region of M protein necessary for its interaction with STX18, a series of truncated M mutants were constructed and used for co-IP assay. We found that M protein 20-100aa is required and sufficient for the interaction with STX18, and M^{Δ20-100aa} overexpression failed to induce lipophagy (**Revised Manuscript Fig. S8b-d**).

Revised Manuscript Fig. S8b-d (b) HEK293T cells were transfected with M-Flag or its mutants for 36 h. Cells were subjected to Flag IP and analyzed via western blot. (c) HEK293T cells were transfected with M-Flag or its mutant for 36 h. Cell lysates were analyzed via western blot. (d) HeLa cells expressing the M-Flag or M^{Δ20-100aa}-Flag were treated with 200 μM OA for 6 h, then fixed and immunostained with anti-Flag (red). LDs were labeled with BODIPY-493/503 (green). Cells were imaged by confocal microscopy. White ROIs indicate the cells expressing M protein. Scar bar represents 10 μm.

Q23) Can the authors provided a marker indicating VSV or SARS-CoV-2 infection?

Response: We thank reviewer for this helpful suggestion. We have used antibody to detect VSV M protein (**Revised Manuscript Fig. 5b, f, n, o**). In original manuscript, we had used SARS-CoV-2 N antibody to indicate virus infection (**Original manuscript Fig. 6n, p, q**).

Q24) In Figure 4O, Figure 6C, Figure 6E and Figure S3K, there was no control group?

Response: Many thanks for the critical comments and suggestions. For Original Fig 4o, we investigated the effect of STX18 KD on LD number in *Atg14* KO cells rescued by vector, or wild-type ATG14, or ATG14^{LIRm}. So, the control group is *Atg14* KO cells transfected with negative siRNA. For control groups were wild-type cells transfected with negative siRNA or si-STX18, please refer revised Fig 4a, 4h, S6a, S6c.

For Original Fig6c, we apologize for this missing information. In Original Fig6c White ROIs and “+” indicate the cells expressing M protein and yellow ROIs indicate the cells without or low M expression as control group. We have added this information in figure legend.

For Original Fig6e, “+” indicate the cells expressing M protein, and “-” indicate cells without or low M expression as control group. We have added this information in figure legend.

For Fig Original S3k, this experiment was performed together with Original Fig3n. The data were supplemented due to the space limitation. Quantifications for Original Fig 3n and 3k were added (**Revised Manuscript Fig 3o**).

Revised Manuscript Fig. 3o The number of colocalization of PI3P with LDs per cell in (n and Supplementary Fig.4j) was counted from 50 cells. Three independent experiments. Two-tailed Unpaired Student's t-test.

Suggestion: A working model for this study would be helpful for understanding the mechanism the authors propose.

Response: We are grateful for the Reviewer's helpful suggestions. We have added a working model (**Revised Manuscript Fig. 7**).

Revised Manuscript Fig. 7 Proposed model for STX18-ATG14 axis regulate lipophagy. In wild-type cells, STX18 interacts with ATG14 and disrupts the interactions of ATG14 with LC3 and subverts the formation of PI3KC3-C1 complex to inhibit lipophagy. In coronavirus infected cells, viral M protein binds STX18 and subverts the STX18-ATG14 interaction to induce lipophagy and degrade the LD-associated anti-viral protein Viperin.

REVIEWER COMMENTS

Reviewer #1 (Remarks to the Author):

The authors have addressed most of my questions.

However, the concerns raised originally in point 4 remain and are quite serious:

The authors maintain their claim that the amphiphatic helix 303-320aa mediates interaction with lipid droplets, despite agreeing that the residues they identified as critical for lipid droplet localization (i.e. I303, L317, I320) are not surface exposed and hence are unlikely to mediate binding of lipid droplets. The authors propose a variety of scenarios, including structural changes exposing said helix in the wild type protein. However, in my opinion the most likely explanation regarding the mutant protein is that deletion of the internal helix or mutation of critical residues in said internal helix will simply have non-specifically inactivated the protein, which would easily explain the observed lack of lipid droplet association due to structural reasons and without postulating a receptor function for the helix.

Reviewer #2 (Remarks to the Author):

Review: Nat Comm 'ATG14 Targets Lipid Droplets and Acts as An Autophagic Receptor for Syntaxin18 Regulated Lipid Droplet Turnover'

Reviewer General Comments: This resubmission represents large amount of work with a few moderate issues that remain which have a red asterisk. Most notable, the resolution on some of the resubmitted data figures needs to be improved (Fig. 1, S1), there is no clear interpretation on the structures being examined in the EM panels of Fig4 (is this macrolipophagy?), and the authors did not really provide evidence whether the STX18-KD is associated with a loss of LD synthesis (as noted below for Fig4). Aside from these concerns, I believe the authors have completed what represents a good faith effort to address the suggestions of this reviewer. .

General comments :

This manuscript suggests that the lipophagy promoting activity of ATG14 is via direct recruitment of LC3 positive membranes to LDs, an activity which is negatively impacted by interactions with the SNARE

protein STX18. An interesting functional role for these proteins is tied to a SARS-CoV-2 infection where the M-protein seems to disrupt this interaction, thereby promoting viral infection through turnover of the LD resident, anti-viral protein Viperin. Most of the data is fairly convincing, some specific concerns are noted below. Conceptually, it would be better to clearly show that the loss of STX18 has no direct impact on LD synthesis/expansion as put forth by the Li laboratory (Xu et al, JCB 2018) and instead is a driver of autophagic/lysosome-based turnover of LD content. The figures concerning viral infection and the abrogation of the ATG14/STX18 nexus for controlling lipophagy seems interesting, though almost a part of a different story?

Response: Many thanks for your conclusion of our work. Your constructive suggestions help us significantly improve the quality of this manuscript.

Comment: Not really a response to the question

Summary and Specific Suggestions :

Fig1 shows ATG14 localization to LDs, which has previously been shown (Pfisterer et al, JLR 2014) as referenced. The authors do attempt to map out the amphipathic helix (Fig 1F,G) required for targeting ATG14 to LDs, though the images presented are not conclusive. The authors demonstrate that the BATS domain (membrane curvature sensing domain) of ATG14 is required for LD localization, which is a more convincing component of this image panel (1G).

Response: Thank you for your important comments and suggestions. We agree that BATS domain-mediated ATG14 localization is a more convincing component. To further confirm that BATS domain of ATG14 is required for LD localization, we examined the LD localization of mutants ATG14 Δ C10aa (defect in BATS), and found that ATG14 Δ C10aa failed to localize on LDs (Revised Manuscript Fig. 1h). Additionally, an LD-resident ATG14W_{WFY}-RRR was engineered through substitution of the ATG14W_{WFY}-RRR (defect in BATS) signal peptide with the LD-targeting amphipathic helices (LD-ATG14W_{WFY}-RRR). LD-ATG14W_{WFY}-RRR re-localized on LDs and rescued the decreased LD number in Atg14 KO cells, but not ATG14W_{WFY}-RRR (Revised Manuscript Fig. S2g).

Comment: This was a minor point, though I'm not sure the images in revised FigS2g are convincing to show a reversal in LD content?

Q1) The images in 1G are not convincing to show a lack of LD localization for the amphipathic helix mutant (ILI-QQQ), perhaps improve the magnification or clarify with additional examples?

Response: Thank you for your valuable suggestions. We have replaced with additional examples

(Response letter Fig. 2). Additionally, in revised manuscript, we generated mCherry tag ATG14 and more point mutations were introduced into the hydrophobic interfaces of each helix: AH1, L109Q/I127Q (ATG14LI-QQ), I113Q/I120Q (ATG14II-QQ) or I113Q/I120Q/L123Q (ATG14IIL-QQQ), or AH2, I303Q, L317Q and I320Q (ATG14ILI-QQQ) to break their structures (Revised Manuscript Fig. S1e). Strikingly, ATG14LI-QQ, ATG14II-QQ, and ATG14IIL-QQQ were still localized to LDs, whereas ATG14ILI-QQQ failed to associate with LDs (Revised Manuscript Fig. S1f, g). Furthermore, deletion of the second amphipathic helices (303-320aa) also abolished LD localization (Revised Manuscript Fig.1h and S1g), indicating that the binding of ATG14 to LDs is dependent on the second amphipathic helices.

Comment: The resolution on figure S1 needs to be improved as it is hard to distinguish which mutants are being viewed. It seems as though the ILI-QQQ mutant failed to localize to LDs, but the images for the deletion mutants in Fig 1h are not convincing. There still seems to be some co-localization of the deletion mutants to LDs in both panels of 1h. The authors should probably improve the image quality or choose additional examples before publication.

A knockdown of ATG14 is associated with an increase in LD content of HeLa cells, consistent with previously published findings by Xiong et al in liver tissue (JBC 2012) and Pfisterer et al in cell lines (JLR 2014). When overexpressed, ATG14 suppresses LD content when localized properly, which was reversed by chloroquine, indicating a likely role in promoting lipophagy (1J, K).

Note1) The authors try to use Atglistatin in HeLa cells, which to my knowledge has only been successful in murine cells. They should perhaps remove this data and not make claims related to lipolysis in this context.

Response: We appreciate this helpful advice. As your suggestion, we have removed the Atglistatin treatment data.

Comment: The authors took our concern seriously and removed the Atglistatin data.

Note2) It will be more convincing if more autophagy inhibitors were utilized to solidify the findings, at least include Bafilomycin A1.

Response: Thank you for your valuable suggestions. We have examined the effect of Baf-A1 on ATG14-induced lipophagy and found that Baf-A1 treatment blocked ATG14-induced lipophagy (Revised Manuscript Fig. 1j-l).

Comment: Additional data with BafA is convincing.

Note3) As the authors suggest that ATG14 does not localize to nascent LDs, it is hard to understand why in the steady state (without OA) ATG14 KD could increase the abundance of LDs (Figure 1I). Also, it seems necessary to show the efficiency of ATG14 KD by WB or RT-qPCR.

Response: Thank you for your insightful comments and suggestions. ATG14 is known as the core unit of the PI3KC3-C1 complex and binds STX17-SNAP29 complex to promote autophagosomelysosome fusion. ATG14 KD caused the defect of autophagy initiation and autophagosomelysosome fusion, so in the steady state, LDs were accumulated due to the defect of basal autophagic flux (initiation and fusion) in ATG14 KD cells. As your suggestion, we added the WB result to show ATG14 KD efficiency (Revised Manuscript Fig. 1i).

Comment: They have demonstrated a sufficient level of knockdown with the added blots in Fig1i.

Note4) Besides Figure 1D, the authors should measure and compare the enrichment of ATG14 in cellular fractions with or without OA.

Response: We are grateful for the Reviewer's important suggestions. Using confocal microscopy, we found that ATG14 had no obvious colocalization with mitochondrial and Golgi while partly localized with lysosomes and the ER with or without OA treatment (Revised Manuscript Fig.s1d).

Comment: This is a reasonable demonstration of compartmental localization, demonstrating some selectivity for LDs and lysosomes while not finding Atg14 at the ER/Golgi.

Q2) It could be interesting to test the interaction between ATG14 and LDs if I113, I120, and/or L123 in ATG14 are/is mutated?

Response: We appreciate this helpful advice. We generated mCherry tag ATG14 and more point mutations were introduced into the hydrophobic interfaces of each helix: AH1, L109Q/I127Q (ATG14LI-QQ), I113Q/I120Q (ATG14II-QQ) or I113Q/I120Q/L123Q (ATG14IIL-QQQ), or AH2, I303Q, L317Q and I320Q (ATG14ILI-QQQ) to break their structures. Strikingly, ATG14LI-QQ, ATG14II-QQ, and ATG14IIL-QQQ were still localized to LDs, whereas ATG14ILI-QQQ failed to associate with LDs (Revised Manuscript Fig. s1f, g).

Comment: Acceptable attempt, as noted the resolution of this figure needs to be improved.

Note5) I suggest including a control group of ATG14 WT in Figure 1g to show localization differences.

Response: Many thanks for your helpful suggestions. We have added GFP alone and GFP-ATG14 as control groups to show localization differences (Revised Manuscript Fig. 1f, g).

Comment: The GFP control was a nice addition, and the GFP-ATG14 localization in the revised figure an improvement.

In FigS1 the Authors find ATG14 did not localize to nascent LDs, showing modest colocalization between GPAT4(152–208) and ATG14, a membrane marker for nascent LDs.

Note6) It would be good to show ATG14 colocalization with late LD markers such as HSD17B11, Ldsdh1 as shown in Song J et al., ncb 2022.

Response: Thank you for your important suggestions. Ldsdh1 is a Drosophila gene, so we examined the colocalization of HSD17B11-ATG14-LDs with OA treatment. Interestingly, we found that ATG14 colocalized with HSD17B11 which were LD-positive (Revised Manuscript Fig. s1c).

Comment: The co-localization with HSD17B11 on mature LDs was convincing.

Q3) This figure needs a control group which was not treated with DSS in Figure S1e?

Response: Thank you for your helpful suggestions. We have added the control group which was not treated with DSS (Revised Manuscript Fig s2m).

Comment: Acceptable addition.

Figure 2 explores the direct association of ATG14 with core autophagic machinery such as LC3 and GABARAP, which has been reported by other groups. They show a clear association with LC3A, LC3C and GABARAP/L2 and do define the LIR. The authors show that there is a requirement of LIR bearing ATG14 to recruit LC3 membranes to LDs in panel 2L, further support is gained from KO/re-expression studies as well.

Q4) It is hard to appreciate exactly how many cells were scored from the immunofluorescence panels presented in Fig2 L/M. Was this listed in the manuscript text or figure legend? Were these results displayed in panel 2M obtained from one experimental trial?

Response: Thank you for your insightful comments. As we mentioned in the original manuscript (Original manuscript lines 1016-1017): “The number of colocalization of ATG14/LC3/LDs per cell in (L) was counted from 50 cells. Three independent experiments. Two-tailed Unpaired Student’s ttest.”

In the revised manuscript, we have added the quantification information in figure legend.

Comment: Acceptable addition.

Q5) The localization of LC3/ATG14 on LDs of serum starved HeLa cells is very graphic in panel 2L, would the authors be able to provide some additional images or increased magnifications?

Response: Thank you for your helpful comments. Here we provided some additional images

(Response letter Fig. 3)

Comment: The images in Fig2I remain more convincing than anything shown in the additional figure assembled for the rebuttal letter. Overall, the data presented are acceptable, though it does seem that the authors are stretching to show dramatic LC3-LD localization as appears in the starved cell of panel 2I.

Q6) Do the HeLa cells in panel 2L have increased labeling of LC3 following starvation? There certainly seems to be quite a high level of this protein in non-starved cells. How specific is the LC3 antibody labeling in 2L? Are the large red structures representative of autophagic vacuoles or mischaracterized? What do these LDs and autophagic membranes look like by EM?

Response: Thank you for your insightful comments. As we mentioned in the original manuscript

(Original manuscript lines 1044-1047): "HeLa cells expressing the GFP-ATG14 or GFPATG14LIRm were treated with 200 μ M OA for 12 h and treated with or without serum starvation for 24 h. Meanwhile, 100 μ M CQ was added for 6 h before cells were fixed and immunostained with anti-LC3 (red). LDs were labeled with LipidTOX Deep Red (blue)." CQ treatment led to accumulation of LC3 dots.

We used Rabbit anti-LC3 (PM036) (Original manuscript line 571) from MBL for

immunostaining, this antibody has been used for imaging assay in previous study (PMID:

32989250, fig4b, 4e). We found that under CQ treatment, large red structures appeared in some

cells, and similar large LC3 structures were also observed by using this antibody in a previous

study (PMID: 33590792, Fig1A).

We appreciate reviewer's advice that we could use CLEM. Unfortunately, we failed to obtain the

data. We have replaced with new pictures in which the size of LC3 structures were regular

(Revised Manuscript Fig. 2I).

Response: Thank you for your valuable suggestions. We have added the control group (Revised

Manuscript Fig. 2e, f).

Comment: As stated, the authors did not look for autophagic membranes on LDs in serum starved cells by EM. There was some attempt to improve the confusing nature of all the LC3 labeling in control cells by replacing panels in Fig2. They have pointed out that the cells were all treated with CQ, likely increasing the amount of LC3 and autophagic membrane labeling.

Q7) It is impressive to see the interaction between ATG14 and LC3. Have the authors tried to see these interactions in endogenous context with or without OA treatment?

Response: Thank you for your valuable suggestions. We have showed that OA treatment has no effect on ATG14-LC3 interactions (Revised Manuscript Fig. S2j).

*Comment: A very clear co-precipitated band is seen in FigS2j regardless of PA addition, however, the authors should label this figure correctly to include the Atg14-FLAG addition?

Q8) In Fig2 panels N,O the authors show no significant look at LD number differences between experimental groups. Could they include an analysis of total LD area/cell? Would the authors be able to provide some additional images or increased magnifications?

Response: Thank you for your insightful comments and suggestions. We have included an analysis of total LD area/cell for Fig 2n (Revised Manuscript Fig. S2n). Additionally, we also provide the analysis of total LD area/cell for As your requested, here we provided some additional images for Atg14 KO cells rescued with mCherry-ATG14 (Response letter Fig. 4). Our data clearly showed that wild-type ATG14 could rescue the decreased LD number in Atg14 KO cells.

Comment: Acceptable, data is included.

Fig 3: Mass spec.-based identification of ATG14 binding partners that change following serum starvation are shown in Fig3. A large decrease in STX18 binding is seen following serum starvation, which itself has been found on the LD and seems to influence LD size and TG amounts by others. Interestingly, the authors show a dramatic level of ATG14/LC3 localization on LDs in STX18 KO cells (Fig 3H).

Response: Thanks for your comments. For mass spectrometry raw data related to Fig3A and Fig5A, we prefer not to share these informations at this stage, because several important projects based on these mass spectrometry data are ongoing (KO mice are just ready). It's nature's policy to share the mass spectrometry raw data, so we have discussed with editor and she agreed that we can delete mass spectrometry raw data when we submit our original manuscript.

Comment: It is interesting that the authors do not have to provide mass spec data files for this publication? Not essential if the Editors are comfortable with this.

Q9) Is the STX18 KO condition associated with increased levels of autophagy? There is so much LC3 labeling in these KO cells, perhaps the state of autophagic flux should be addressed? Are these LC3 membranes the result of impaired autophagic flux? What are the protein levels of p62 and LC3II in these STX18 KO cells?

Response: Thank you for your helpful comments and suggestions. Yes, we found that STX18 KO increased levels of autophagy.

To better examine the colocalization of LC3 with LDs, we treated cells with CQ to block degradation and accumulate LC3 labeling.

Following the suggestion from the reviewer, we found that LC3-II was accumulated and p62 was degraded in Stx18 KO cells, suggesting that STX18 depletion induces autophagic flux (Revised Manuscript Fig. S3f).

Comment: Based on the data presented in FigS3, it certainly seems as though they are inducing autophagic flux with the STX18 KO.

Note7) The authors found the STX18-ATG14 through MS. It might also be interesting to confirm the binding between these two proteins in an endogenous context, with or without OA treatment, with or without an autophagy inhibitor.

Response: We appreciate this critical advice. We analyzed the endogenous ATG14-STX18 interactions and found that OA or CQ treatment, or knockdown of Atg7 had minimal effect in an endogenous context between STX18 and ATG14 (Revised Manuscript Fig. S3d, e).

Comment: The co-IP studies in Fig S3 are very clear and seem to demonstrate that the ATG14-STX18 interactions are not impacted by OA or Atg7 knockdown.

Note8) The colocalization between ATG14 and LD should be done in a context of STX18 knockdown or in STX18 KO cells, potentially following rescue by exogenously expressed STX18.

Response: Thank you for your valuable suggestions. The colocalization between ATG14 and LD was confirmed in Stx18 KO cells, following rescue by exogenously expressed STX18 (Revised Manuscript Fig. S3g).

Comment: ATG14 localization to LDs did not seem to be impacted by STX14-KO.

Note9) In Figure S3G, it was nice to show the overall protein of STX18 by using anti-STX18 antibody. It would be helpful if the authors could put a label indicating which was endogenous/exogenous.

Response: We thank reviewer for this helpful suggestion. label indicating which was endogenous/exogenous has been added.

Comment: The authors added the labeling as suggested.

Q10) In Figure 3I, it seems that every protein examined interacted with STX18, these findings could be strengthened by a negative result?

Response: We are grateful for the Reviewer's helpful suggestions. We found that STX18 failed to interact with LC3, and we added this data as a negative result (Revised Manuscript Fig. 3i).

Comment: The authors did provide a negative control their STX18 co-IP, though it is still a bit concerning that there were so many additional proteins pulled down.

Q11) In Figure 3n,o, the quantification should go after the taken images. So, the panel number for 'n' and 'o' should be exchanged. Also, the knockdown efficiency should be detected by western blot analysis or RT-qPCR assay.

Response: Thanks for your insightful advice. We have changed the panel number and knockdown efficiency has been detected by western blot (Revised Manuscript Fig. 3m).

Comment: Acceptable changes were made as requested.

Fig4 demonstrates that a knockdown of STX18 suppresses LD content, which seems to be dependent on autophagy as a knockdown of Atg7 results in LD retention. Consistent with this is an increased frequency of LDs found in single membrane bound autolysosome like structures, what the authors term 'lipophagosomes', in Fig 4D-F following STX18 knockdown or KO. They also use a GFP-RFP-PLIN2 reporter to show that there are more RFP only + LDs in the cells following STX18 KD, consistent with the idea that in this knockdown state there is more lipophagy occurring.

Q12) Are these structures that contain LDs indeed representative of anything related to

autophagosomes? They appear to be lysosomes and are by the authors' own admission single membrane bound.

Response: Thank you for your insightful comments. We apologize for misleading elaboration. We have rephrased the sentence to classify that "we directly visualized "lipo-phagolysosomes" in which LDs were engulfed into single-membrane autolysosomes in Stx18 KO HepG2 and STX18 knockdown HeLa cells".

Comment: The descriptor 'lipo-phagolysosomes' is a bit unconventional and does not really help to clarify what the authors are looking at which appears to be microlipophagy.

Q13) How do these findings of STX18 KO suppressing LD content relate to the findings of Xu et al (JCB, 2018) where this SNARE is implicated in promoting LD formation/expansion at the ER? If their model is followed, wouldn't the findings of this manuscript be consistent with a STX18 loss reducing LD content from a synthesis aspect?

Response: Thank you for your insightful comments. In Xu et al (JCB, 2018), they suggested that STX18 KD impaired LD biogenesis via blocking ER-LD contact in adipose cells. Our results showed that in HeLa cells STX18 KD reduced LD content dependent on autophagic machinery: Atg7 deletion rescued STX18 loss reducing LD content, we further noticed that in Atg7 deleted cells, STX18 KD failed to induce reduction of LD number and triglyceride levels, indicating that STX18 KD has minimal effect on LD synthesis. We have added this discussion: "We further noticed that in Atg7 deleted cells, STX18 depletion failed to induce reduction of LD number and triglyceride levels, indicating that STX18 depletion has minimal effect on LD synthesis." (Revised Manuscript lines 490-492).

Comment: The data shown in panel 4a and b are being discussed in the authors' response to this question. Atg7 knockdown does not seem to impact LD content by itself, though it does seem to reverse the suppression of LD content and cellular TG seen following STX18 KD in these cells (STX18 loss causes more lipophagy and suppresses LD/TG levels). The reversal of STX18 suppressed LD/TG levels by Atg7 is the sole reason that any effect on TG/LD-synthesis is excluded from their model. Though this is logically consistent, I'm not sure if this is enough evidence to exclude a synthesis component. Perhaps they could ensure that a STX18 KD has no effect on LD formation in these HeLa cells?

Q14) In the image 4A, in syx18/ ATG 7 double KD, LD number look less than control and ATG 7 KD however graphically look similar.

Response: Thank you for your insightful comments. We have re-performed the experiment and replaced with new representative pictures (Revised Manuscript Fig. 4a).

Comment: The authors replaced the images in the figure which are more consistent with their findings.

Q15) Why are there so many LDs present in the STX18 knockdown state in Fig4J, this does not seem to be consistent with the LD content suppression seen in other panels following STX18 KD?

Response: Thank you for your insightful comments. As we mentioned in our figure legend, for RFP-GFP-Plin2 assay, we treated cells with OA for 12 h to induce LD biogenesis and Plin2 LD localization, and investigated LD-localized Plin2-RFP+GFP- signals; for LD number assay, we treated cells with OA for 6 h and investigated LD degradation.

Comment: It is reasonable to assume that the presence of the RFP-GFP-Plin2 reporter changes LD levels in these cells.

ATG14 seems to be required for the STX18 KO/KD induced suppression of LD content (Fig4 O and P), and a LIR competent version of ATG14 is needed to suppress LD levels.

Q16) It lacks essential data to directly support that STX18 completely bound to CCD of ATG14

Response: We thank reviewer for this critical comment. We found that STX18 binds the CCD of ATG14 (Revised Manuscript Fig. S5h). To confirm that STX18 disrupts the formation of PI3KC3-C1 complex by competitively binding the CCD in ATG14, we gradually increased the expression of ATG14 and found that the STX18 overexpression-inhibited Torin1-induced p62 degradation was reversed (Revised Manuscript Fig. S5i).

Comment: The binding of the CCD of ATG14 is now shown.

Note10) For the effect of Beclin on LDs function, the authors should include the ATG14 KD and both of ATG14 and Beclin KD to demonstrate the essence of ATG14 or Beclin.

Response: We are grateful for the Reviewer's helpful suggestions. We found that besides ATG14, STX18 regulates lipophagy also dependent on Beclin1 (Revised Manuscript Fig. S6a).

Comment: There is a clear dependence on Beclin in their HeLa cell model (FigS6a)

Q17) The Supplementary FigS4C IP is done with HA beads, not myc. Correct the labelling?

Response: We apologize for this error. This has now been corrected.

Comment: Corrected in what is now FigS5C

Q18) In Figure S4F, Beclin1 should be detected as a positive control since CCD domain is critical for the binding between ATG14 and Beclin1.

Response: We appreciate this critical advice. We have confirmed that CCD domain is critical for the binding between ATG14 and Beclin1 (Revised Manuscript Fig. S5f).

Comment: As stated, there is a very prominent band for the Beclin protein in their WT-ATG14 IP.

Q19) In Figure S4I, authors should also investigate if autophagic flux was influenced by STX18 in the ATG14 KO MEF cells.

Response: Thank you for your instructive and critical suggestions. We don't have Atg14 KO MEF cells, so we generated Atg14 KO HeLa cells and found that STX18 depletion failed to induce p62 degradation in Atg14 KO cells, indicating that STX18 negatively regulate autophagic flux dependent on ATG14 (Revised Manuscript Fig. S5m).

Comment: As stated, the authors did develop an ATG14-KO cell line that did seem to have compromised autophagic flux based on p62 turnover.

Fig5 seems to show that a LD localized anti-viral protein Viperin is degraded following these STX18 manipulations, which are potentially just an extension of the central findings that STX18 loss induces lipophagy. A potentially interesting connection is made with SARS-CoV-2 viral infection in Fig 6. They find that the M-protein of SARS-CoV-2, and viral infection itself, subverts the STX18/ATG14 interaction thereby promoting lipophagy and degradation of the anti-viral protein Viperin. In this model then, the lipophagy promoting activity of disrupting STX18/ATG14 seems to promote viral infection itself.

Q20) Could the authors show immunofluorescence images indicating the coincidence of LDs and Viperin with LC3 in the cells manipulated with ATG14 LIR mutant or the STX18 mutant (with ATG14 binding deficiency)?

Response: We appreciate this critical advice. Our new data indicated that rescued with wild-type STX18, but not STX18 Δ 71-80 (defect in ATG14 interaction) abolished the colocalization of LC3- Viperin-LDs in Stx18 KO cells (Revised Manuscript Fig. S7d). In addition, obvious colocalization between LC3-Viperin-LDs was observed in ATG14-Flag expressed, but not vector or ATG14LIRm-Flag expressed Atg14 KO cells (Revised Manuscript Fig. S7e).

Comment: The authors did provide examples of this co-localization.

Q22) Is STX18 (a.a.71-80) critical for the binding of M protein? Would it be helpful to define which domain located in the M protein is important for the binding of STX18 reciprocally?

Response: Thank you for your instructive and critical suggestions. Our co-IP data showed that STX18 Δ 1-80 still interacts with M protein, suggesting that STX18 (a.a.71-80) is not critical for the binding of M protein (Response letter Fig. 5).

As your suggested, to map the critical region of M protein necessary for its interaction with STX18, a series of truncated M mutants were constructed and used for co-IP assay. We found that M protein 20-100aa is required and sufficient for the interaction with STX18, and M Δ 20-100aa overexpression failed to induce lipophagy (Revised Manuscript Fig. S8b-d).

Comment: The authors did provide this data.

Q23) Can the authors provided a marker indicating VSV or SARS-CoV-2 infection?

Response: We thank reviewer for this helpful suggestion. We have used antibody to detect VSV M protein (Revised Manuscript Fig. 5b, f, n, o). In original manuscript, we had used SARS-CoV-2 N antibody to indicate virus infection (Original manuscript Fig. 6n, p, q).

Comment: Clear bands for the M protein can be seen in these lysates of Fig5.

Q24) In Figure 4O, Figure 6C, Figure 6E and Figure S3K, there was no control group?

Response: Many thanks for the critical comments and suggestions. For Original Fig 4o, we investigated the effect of STX18 KD on LD number in Atg14 KO cells rescued by vector, or wildtype ATG14, or ATG14LIRm. So, the control group is Atg14 KO cells transfected with negative siRNA. For control groups were wild-type cells transfected with negative siRNA or si-STX18, please refer revised Fig 4a, 4h, S6a, S6c.

For Original Fig6c, we apologize for this missing information. In Original Fig6c White ROIs and “+” indicate the cells expressing M protein and yellow ROIs indicate the cells without or low M expression as control group. We have added this information in figure legend.

For Original Fig6e, “+” indicate the cells expressing M protein, and “-” indicate cells without or low M expression as control group. We have added this information in figure legend.

For Fig Original S3k, this experiment was performed together with Original Fig3n. The data were supplemented due to the space limitation. Quantifications for Original Fig 3n and 3k were added (Revised Manuscript Fig 3o).

Response: We are grateful for the Reviewer's helpful suggestions. We have added a working model (Revised Manuscript Fig. 7).

Comment: Their model is a nice addition.

Point-by-point response:

REVIEWER COMMENTS

Reviewer #1 (Remarks to the Author):

The authors have addressed most of my questions.

However, the concerns raised originally in point 4 remain and are quite serious:

The authors maintain their claim that the amphiphatic helix 303-320aa mediates interaction with lipid droplets, despite agreeing that the residues they identified as critical for lipid droplet localization (i.e. I303, L317, I320) are not surface exposed and hence are unlikely to mediate binding of lipid droplets. The authors propose a variety of scenarios, including structural changes exposing said helix in the wild type protein. However, in my opinion the most likely explanation regarding the mutant protein is that deletion of the internal helix or mutation of critical residues in said internal helix will simply have non-specifically inactivated the protein, which would easily explain the observed lack of lipid droplet association due to structural reasons and without postulating a receptor function for the helix.

Response: We absolutely agree that mutating the internal residues is likely to create a non-functional protein.

In the revised manuscript, we have deleted the related data that internal amphipathic helix 303-320aa of ATG14 mediates interaction with LDs.

Reviewer #2 (Remarks to the Author):

Review: Nat Comm 'ATG14 Targets Lipid Droplets and Acts as An Autophagic Receptor for Syntaxin18 Regulated Lipid Droplet Turnover'

Reviewer General Comments: This resubmission represents large amount of work with a few moderate issues that remain which have a red asterisk. Most notable, the resolution on some of the resubmitted data figures needs to be improved (Fig. 1, S1), there is no clear interpretation on the structures being examined in the EM panels of Fig4 (is this macrolipophagy?), and the authors did not really provide evidence whether the STX18-KD is associated with a loss of LD synthesis (as noted below for Fig4). Aside from these concerns, I believe the authors have completed what represents a good faith effort to address the suggestions of this reviewer. .

General comments :

This manuscript suggests that the lipophagy promoting activity of ATG14 is via direct recruitment of LC3 positive membranes to LDs, an activity which is negatively impacted by interactions with the SNARE protein STX18. An interesting functional role for these proteins is tied to a SARS-CoV-2 infection where the M-protein seems to disrupt this interaction, thereby promoting viral infection through turnover of the LD resident, anti-viral protein Viperin. Most of the data is fairly convincing, some specific concerns are noted below. Conceptually, it would be better to clearly show that the loss of STX18 has no direct impact on LD synthesis/expansion as put forth by the Li laboratory (Xuet al, JCB 2018) and instead is a driver of autophagic/lysosome-based turnover of LD content. The

figures concerning viral infection and the abrogation of the ATG14/STX18 nexus for controlling lipophagy seems interesting, though almost a part of a different story?

Response: Many thanks for your conclusion of our work. Your constructive suggestions help us significantly improve the quality of this manuscript.

Comment: Not really a response to the question

Response: We are grateful for the Reviewer's helpful comments. To investigate whether STX18 KD has any effect on LD synthesis/expansion, we co-treated cells with ATGL siRNAs and CQ to block LD degradation (lipolysis and lipophagy), and found that STX18 KD caused minor effect on the number of LDs in CQ-treated ATGL KD cells (**Revised Fig s3h**).

Furthermore, LiveDrop is a widely used probe to label nascent lipid droplets (PMID: 27564575). The results showed that STX18 KD had minor effect on the amount of LiveDrop puncta (**Revised Fig s3i**).

Together, these data indicate that loss of STX18 has no direct impact on LD synthesis/expansion.

Revised Manuscript Figs. S3h

Revised Manuscript Figs. S3i

Summary and Specific Suggestions :

Fig1 shows ATG14 localization to LDs, which has previously been shown (Pfisterer et al, JLR 2014) as referenced. The authors do attempt to map out the amphipathic helix (Fig 1F,G) required for targeting ATG14 to LDs, though the images presented are not conclusive. The authors demonstrate that the BATS domain (membrane curvature sensing domain) of ATG14 is required for LD localization, which is a more convincing component of this image panel (1G).

Response: Thank you for your important comments and suggestions. We agree that BATS domain-mediated ATG14 localization is a more convincing component. To further confirm that BATS domain of ATG14 is required for LD localization, we examined the LD localization of mutants ATG14 Δ C10aa (defect in BATS), and found that ATG14 Δ C10aa failed to localize on LDs (Revised Manuscript Fig. 1h). Additionally, an LD-resident ATG14^{WFY-RRR} was engineered through substitution of the ATG14^{WFY-RRR} (defect in BATS) signal peptide with the LD-targeting amphipathic helices (LD-ATG14^{WFY-RRR}). LD-ATG14^{WFY-RRR} re-localized on LDs and rescued the decreased LD number in Atg14 KO cells, but not ATG14^{WFY-RRR} (Revised Manuscript Fig. S2g).

Comment: This was a minor point, though I'm not sure the images in revised FigS2g are convincing to show a reversal in LD content?

Response: Thank you for your insightful comments. We first confirmed the colocalization between LD-ATG14^{WFY-RRR} and LDs in OA and CQ co-treated cells (**Revised Fig. S2e**). And then we replaced with additional example to show that rescued the decreased LD number in Atg14 KO cells, but not ATG14^{WFY-RRR} (**Revised Fig. S2f**).

Revised Manuscript Figs. S2e

Revised Manuscript Figs. S2f

Q1) The images in 1G are not convincing to show a lack of LD localization for the amphipathic helix mutant (ILI-QQQ), perhaps improve the magnification or clarify with additional examples?

Response: Thank you for your valuable suggestions. We have replaced with additional examples (Response letter Fig. 2). Additionally, in revised manuscript, we generated mCherry tag ATG14 and more point mutations were introduced into the hydrophobic interfaces of each helix: AH1, L109Q/I127Q (ATG14LI-QQ), I113Q/I120Q (ATG14II-QQ) or I113Q/I120Q/L123Q (ATG14IIL-QQQ), or AH2, I303Q, L317Q and I320Q (ATG14LI-QQQ) to break their structures (Revised Manuscript Fig. S1e). Strikingly, ATG14LI-QQ, ATG14II-QQ, and ATG14IIL-QQQ were still localized to LDs, whereas ATG14LI-QQQ failed to associate with LDs (Revised Manuscript Fig. S1f, g). Furthermore, deletion of the second amphipathic helices (303-320aa) also abolished LD localization (Revised Manuscript Fig. 1h and S1g), indicating that the binding of ATG14 to LDs is dependent on the second amphipathic helices.

Comment: The resolution on figure S1 needs to be improved as it is hard to distinguish which mutants are being viewed. It seems as though the ILI-QQQ mutant failed to localize to LDs, but the images for the deletion mutants in Fig 1h are not convincing. There still seems to be some co-localization of the deletion mutants to LDs in both panels of 1h. The authors should probably improve the image quality or choose additional examples before publication.

Response: As suggested by reviewer#1, in the revised manuscript, we have deleted the related data that internal amphipathic helix 303-320aa of ATG14 mediates interaction with LDs.

A knockdown of ATG14 is associated with an increase in LD content of HeLa cells, consistent with previously published findings by Xiong et al in-liver tissue (JBC 2012) and Pfisterer et al in cell lines (JLR 2014). When overexpressed, ATG14 suppresses LD content when localized properly, which was reversed by chloroquine, indicating a likely role in promoting lipophagy (1J, K).

Note1) The authors try to use Atglistatin in HeLa cells, which to my knowledge has only been successful in murine cells. They should perhaps remove this data and not make claims related to

lipolysis in this context.

Response: We appreciate this helpful advice. As your suggestion, we have removed the Atglistatin treatment data.

Comment: The authors took our concern seriously and removed the Atglistatin data.

Response: This comment has been answered.

Note2) It will be more convincing if more autophagy inhibitors were utilized to solidify the findings, at least include Bafilomycin A1.

Response: Thank you for your valuable suggestions. We have examined the effect of Baf-A1 on ATG14-induced lipophagy and found that Baf-A1 treatment blocked ATG14-induced lipophagy (Revised Manuscript Fig. 1j-l).

Comment: Additional data with BafA is convincing.

Response: This comment has been answered.

Note3) As the authors suggest that ATG14 does not localize to nascent LDs, it is hard to understand why in the steady state (without OA) ATG14 KD could increase the abundance of LDs (Figure 1l).

Also, it seems necessary to show the efficiency of ATG14 KD by WB or RT-qPCR.

Response: Thank you for your insightful comments and suggestions. ATG14 is known as the core unit of the PI3KC3-C1 complex and binds STX17-SNAP29 complex to promote autophagosomelysosome fusion. ATG14 KD caused the defect of autophagy initiation and autophagosomelysosome fusion, so in the steady state, LDs were accumulated due to the defect of basal autophagic flux (initiation and fusion) in ATG14 KD cells. As your suggestion, we added the WB result to show ATG14 KD efficiency (Revised Manuscript Fig. 1i).

Comment: They have demonstrated a sufficient level of knockdown with the added blots in Fig1i.

Response: This comment has been answered.

Note4) Besides Figure 1D, the authors should measure and compare the enrichment of ATG14 in cellular fractions with or without OA.

Response: We are grateful for the Reviewer's important suggestions. Using confocal microscopy, we found that ATG14 had no obvious colocalization with mitochondrial and Golgi while partly localized with lysosomes and the ER with or without OA treatment (Revised Manuscript Fig.s1d).

Comment: This is a reasonable demonstration of compartmental localization, demonstrating some selectivity for LDs and lysosomes while not finding Atg14 at the ER/Golgi.

Response: This comment has been answered.

Q2) It could be interesting to test the interaction between ATG14 and LDs if I113, I120, and/or L123 in ATG14 are/is mutated?

Response: We appreciate this helpful advice. We generated mCherry tag ATG14 and more point mutations were introduced into the hydrophobic interfaces of each helix: AH1, L109Q/I127Q (ATG14LI-QQ), I113Q/I120Q (ATG14II-QQ) or I113Q/I120Q/L123Q (ATG14IIL-QQQ), or AH2, I303Q, L317Q and I320Q (ATG14ILI-QQQ) to break their structures. Strikingly, ATG14LI-QQ, ATG14II-QQ, and ATG14IIL-QQQ were still localized to LDs, whereas ATG14ILI-QQQ failed to associate with LDs (Revised Manuscript Fig. s1f, g).

Comment: Acceptable attempt, as noted the resolution of this figure needs to be improved.

Response: This comment has been answered.

Note5) I suggest including a control group of ATG14 WT in Figure 1g to show localization differences.

Response: Many thanks for your helpful suggestions. We have added GFP alone and GFP-ATG14 as control groups to show localization differences (Revised Manuscript Fig. 1f, g).

Comment: The GFP control was a nice addition, and the GFP-ATG14 localization in the revised figure an improvement.

Response: This comment has been answered.

In FigS1 the Authors find ATG14 did not localize to nascent LDs, showing modest colocalization between GPAT4(152–208) and ATG14, a membrane marker for nascent LDs.

Note6) It would be good to show ATG14 colocalization with late LD markers such as HSD17B11, Ldsdh1 as shown in Song J et al., ncb 2022.

Response: Thank you for your important suggestions. Ldsdh1 is a Drosophila gene, so we examined the colocalization of HSD17B11-ATG14-LDs with OA treatment. Interestingly, we found that ATG14 colocalized with HSD17B11 which were LD-positive (Revised Manuscript Fig. s1c).

Comment: The co-localization with HSD17B11 on mature LDs was convincing.

Response: This comment has been answered

Q3) This figure needs a control group which was not treated with DSS in Figure S1e?

Response: Thank you for your helpful suggestions. We have added the control group which was not treated with DSS (Revised Manuscript Fig s2m).

Comment: Acceptable addition.

Response: This comment has been answered.

Figure 2 explores the direct association of ATG14 with core autophagic machinery such as LC3 and GABARAP, which has been reported by other groups. They show a clear association with LC3A, LC3C and GABARAP/L2 and do define the LIR. The authors show that there is a requirement of LIR bearing ATG14 to recruit LC3 membranes to LDs in panel 2L, further support is gained from KO/re-expression studies as well.

Q4) It is hard to appreciate exactly how many cells were scored from the immunofluorescence panels presented in Fig2 L/M. Was this listed in the manuscript text or figure legend? Were these results displayed in panel 2M obtained from one experimental trial?

Response: Thank you for your insightful comments. As we mentioned in the original manuscript (Original manuscript lines 1016-1017): “The number of colocalization of ATG14/LC3/LDs per cell in (L) was counted from 50 cells. Three independent experiments. Two-tailed Unpaired Student’s ttest.” In the revised manuscript, we have added the quantification information in figure legend.

Comment: Acceptable addition.

Response: This comment has been answered.

Q5) The localization of LC3/ATG14 on LDs of serum starved HeLa cells is very graphic in panel 2L, would the authors be able to provide some additional images or increased magnifications?

Response: Thank you for your helpful comments. Here we provided some additional images (Response letter Fig. 3)

Comment: The images in Fig2l remain more convincing than anything shown in the additional figure assembled for the rebuttal letter. Overall, the data presented are acceptable, though it does seem that the authors are stretching to show dramatic LC3-LD localization as appears in the starved cell of panel 2l.

Response: This comment has been answered.

Q6) Do the HeLa cells in panel 2L have increased labeling of LC3 following starvation? There certainly seems to be quite a high level of this protein in non-starved cells. How specific is the LC3antibody labeling in 2L? Are the large red structures representative of autophagic vacuoles or mischaracterized? What do these LDs and autophagic membranes look like by EM?

Response: Thank you for your insightful comments. As we mentioned in the original manuscript (Original manuscript lines 1044-1047): "HeLa cells expressing the GFP-ATG14 or GFPATG14LIRm were treated with 200 μ M OA for 12 h and treated with or without serum starvation for 24 h. Meanwhile, 100 μ M CQ was added for 6 h before cells were fixed and immunostained with anti-LC3 (red). LDs were labeled with LipidTOX Deep Red (blue)." CQ treatment led to accumulation of LC3 dots. We used Rabbit anti-LC3 (PM036) (Original manuscript line 571) from MBL for immunostaining, this antibody has been used for imaging assay in previous study (PMID: 32989250, fig4b, 4e). We found that under CQ treatment, large red structures appeared in some cells, and similar large LC3 structures were also observed by using this antibody in a previous study (PMID: 33590792, Fig1A).

We appreciate reviewer's advice that we could use CLEM. Unfortunately, we failed to obtain the data. We have replaced with new pictures in which the size of LC3 structures were regular (Revised Manuscript Fig. 2l).

Response: Thank you for your valuable suggestions. We have added the control group (Revised Manuscript Fig. 2e, f).

Comment: As stated, the authors did not look for autophagic membranes on LDs in serum starved cells by EM. There was some attempt to improve the confusing nature of all the LC3 labeling in control cells by replacing panels in Fig2. They have pointed out that the cells were all treated with CQ, likely increasing the amount of LC3 and autophagic membrane labeling.

Response: This comment has been answered.

Q7) It is impressive to see the interaction between ATG14 and LC3. Have the authors tried to see these interactions in endogenous context with or without OA treatment?

Response: Thank you for your valuable suggestions. We have showed that OA treatment has no effect on ATG14-LC3 interactions (Revised Manuscript Fig. S2j).

*Comment: A very clear co-precipitated band is seen in FigS2j regardless of PA addition, however, the authors should label this figure correctly to include the Atg14-FLAG addition?

Response: Many thanks for the helpful comments. We have labeled the figure correctly (Revised Fig s2g).

Revised Manuscript Figs. S2g

Q8) In Fig2 panels N,O the authors show no significant look at LD number differences between experimental groups. Could they include an analysis of total LD area/cell? Would the authors be able to provide some additional images or increased magnifications?

Response: Thank you for your insightful comments and suggestions. We have included an analysis of total LD area/cell for Fig 2n (Revised Manuscript Fig. S2n). Additionally, we also provide the analysis of total LD area/cell for As your requested, here we provided some additional images for Atg14 KO cells rescued withmCherry-ATG14 (Response letter Fig. 4). Our data clearly showed that wild-type ATG14 could rescue the decreased LD number in Atg14 KO cells.

Comment: Acceptable, data is included.

Response: This comment has been answered.

Fig 3: Mass spec.-based identification of ATG14 binding partners that change following serum starvation are shown in Fig3. A large decrease in STX18 binding is seen following serum starvation, which itself has been found on the LD and seems to influence LD size and TG amounts by others. Interestingly, the authors show a dramatic level of ATG14/LC3 localization on LDs in STX18 KO cells (Fig 3H).

Response: Thanks for your comments. For mass spectrometry raw data related to Fig3A and Fig5A, we prefer not to share these informations at this stage, because several important projects based on these mass spectrometry data are ongoing (KO mice are just ready). It's nature's policy to share the mass spectrometry raw data, so we have discussed with editor and she agreed that we can delete mass spectrometry raw data when we submit our original manuscript.

Comment: It is interesting that the authors do not have to provide mass spec data files for this publication? Not essential if the Editors are comfortable with this.

Response: This comment has been answered.

Q9) Is the STX18 KO condition associated with increased levels of autophagy? There is so much LC3 labeling in these KO cells, perhaps the state of autophagic flux should be addressed? Are these LC3 membranes the result of impaired autophagic flux? What are the protein levels of p62 and LC3II in these STX18 KO cells?

Response: Thank you for your helpful comments and suggestions. Yes, we found that STX18 KO increased levels of autophagy. To better examine the colocalization of LC3 with LDs, we treated cells with CQ to block degradation and accumulate LC3 labeling.

Following the suggestion from the reviewer, we found that LC3-II was accumulated and p62 was degraded in Stx18 KO cells, suggesting that STX18 depletion induces autophagic flux (Revised Manuscript Fig. S3f).

Comment: Based on the data presented in FigS3, it certainly seems as though they are inducing autophagic flux with the STX18 KO.

Response: This comment has been answered.

Note7) The authors found the STX18-ATG14 through MS. It might also be interesting to confirm the binding between these two proteins in an endogenous context, with or without OA treatment, with or without an autophagy inhibitor.

Response: We appreciate this critical advice. We analyzed the endogenous ATG14-STX18 interactions and found that OA or CQ treatment, or knockdown of Atg7 had minimal effect in an endogenous context between STX18 and ATG14 (Revised Manuscript Fig. S3d, e).

Comment: The co-IP studies in Fig S3 are very clear and seem to demonstrate that the ATG14-STX18 interactions are not impacted by OA or Atg7 knockdown.

Response: This comment has been answered.

Note8) The colocalization between ATG14 and LD should be done in a context of STX18 knockdown or in STX18 KO cells, potentially following rescue by exogenously expressed STX18.

Response: Thank you for your valuable suggestions. The colocalization between ATG14 and LD was confirmed in Stx18 KO cells, following rescue by exogenously expressed STX18 (Revised Manuscript Fig. S3g).

Comment: ATG14 localization to LDs did not seem to be impacted by STX14-KO.

Response: This comment has been answered.

Note9) In Figure S3G, it was nice to show the overall protein of STX18 by using anti-STX18 antibody. It would be helpful if the authors could put a label indicating which was endogenous/exogenous.

Response: We thank reviewer for this helpful suggestion. Label indicating which was endogenous/exogenous has been added.

Comment: The authors added the labeling as suggested.

Response: This comment has been answered.

Q10) In Figure 3I, it seems that every protein examined interacted with STX18, these findings could be strengthened by a negative result?

Response: We are grateful for the Reviewer's helpful suggestions. We found that STX18 failed to interact with LC3, and we added this data as a negative result (Revised Manuscript Fig. 3i).

Comment: The authors did provide a negative control their STX18 co-IP, though it is still a bit concerning that there were so many additional proteins pulled down.

Response: This comment has been answered.

Q11) In Figure 3n,o, the quantification should go after the taken images. So, the panel number for 'n' and 'o' should be exchanged. Also, the knockdown efficiency should be detected by western blot analysis or RT-qPCR assay.

Response: Thanks for your insightful advice. We have changed the panel number and knockdown efficiency has been detected by western blot (Revised Manuscript Fig. 3m).

Comment: Acceptable changes were made as requested.

Response: This comment has been answered.

Fig4 demonstrates that a knockdown of STX18 suppresses LD content, which seems to be dependent on autophagy as a knockdown of Atg7 results in LD retention. Consistent with this is an increased frequency of LDs found in single membrane bound autolysosome like structures, what the authors term 'lipophagosomes', in Fig 4D-F following STX18 knockdown or KO. They also use a GFP-RFP-PLIN2 reporter to show that there are more RFP only + LDs in the cells following STX18 KD, consistent with the idea that in this knockdown state there is more lipophagy occurring.

Q12) Are these structures that contain LDs indeed representative of anything related to autophagosomes? They appear to be lysosomes and are by the authors' own admission single membrane bound.

Response: Thank you for your insightful comments. We apologize for misleading elaboration. We have rephrased the sentence to classify that "we directly visualized "lipo-phagolysosomes" in which LDs were engulfed into single-membrane autolysosomes in Stx18 KO HepG2 and STX18 knockdown HeLa cells".

Comment: The descriptor 'lipo-phagolysosomes' is a bit unconventional and does not really help to clarify what the authors are looking at which appears to be microlipophagy.

Response: Many thanks for the helpful comments. As suggested, we have changed to "microlipophagy".

Q13) How do these findings of STX18 KO suppressing LD content relate to the findings of Xu et al (JCB, 2018) where this SNARE is implicated in promoting LD formation/expansion at the ER? If their model is followed, wouldn't the findings of this manuscript be consistent with a STX18 loss reducing LD content from a synthesis aspect?

Response: Thank you for your insightful comments. In Xu et al (JCB, 2018), they suggested that STX18 KD impaired LD biogenesis via blocking ER-LD contact in adipose cells. Our results showed that in HeLa cells STX18 KD reduced LD content dependent on autophagic machinery: Atg7 deletion rescued STX18 loss reducing LD content, we further noticed that in Atg7 deleted cells, STX18 KD failed to induce reduction of LD number and triglyceride levels, indicating that STX18 KD has minimal effect on LD synthesis. We have added this discussion: "We further noticed that in Atg7 deleted cells, STX18 depletion failed to induce reduction of LD number and triglyceride levels, indicating that STX18 depletion has minimal effect on LD synthesis." (Revised Manuscript lines 490-492).

Comment: The data shown in panel 4a and b are being discussed in the authors' response to this question. Atg7 knockdown does not seem to impact LD content by itself, though it does seem to reverse the suppression of LD content and cellular TG seen following STX18 KD in these cells (STX18 loss causes more lipophagy and suppresses LD/TG levels). The reversal of STX18 suppressed LD/TG levels by Atg7 is the sole reason that any effect on TG/LD-synthesis is excluded from their model. Though this is logically consistent, I'm not sure if this is enough evidence to exclude a synthesis component. Perhaps they could ensure that a STX18 KD has no effect on LD formation in these HeLa cells?

Response: We are grateful for the Reviewer's helpful comments. To investigate whether STX18 KD has any effect on LD synthesis/expansion, we co-treated cells with ATGL siRNAs and CQ to block LD degradation (lipolysis and lipophagy), and found that STX18 KD caused minor effect on the number of LDs in CQ-treated ATGL KD cells (**Revised Fig s3h**).

Furthermore, LiveDrop is a widely used probe to label nascent lipid droplets (PMID: 27564575). The results showed that STX18 KD had minor effect on the amount of LiveDrop puncta (**Revised Fig s3i**).

Together, these data indicate that loss of STX18 has no direct impact on LD synthesis/expansion.

Revised Manuscript Figs. S3h

Revised Manuscript Figs. S3i

Q14) In the image 4A, in *syx18*/ ATG 7 double KD, LD number look less than control and ATG 7 KD however graphically look similar.

Response: Thank you for your insightful comments. We have re-performed the experiment and replaced with new representative pictures (Revised Manuscript Fig. 4a).

Comment: The authors replaced the images in the figure which are more consistent with their findings.

Response: This comment has been answered.

Q15) Why are there so many LDs present in the STX18 knockdown state in Fig4J, this does not seem to be consistent with the LD content suppression seen in other panels following STX18 KD?

Response: Thank you for your insightful comments. As we mentioned in our figure legend, for RFP-GFP-Plin2 assay, we treated cells with OA for 12 h to induce LD biogenesis and Plin2 LD localization, and investigated LD-localized Plin2-RFP+GFP- signals; for LD number assay, we treated cells with OA for 6 h and investigated LD degradation.

Comment: It is reasonable to assume that the presence of the RFP-GFP-Plin2 reporter changes LD levels in these cells.

Response: We are grateful for the Reviewer's helpful comments. We have confirmed that Plin2 overexpression induces LD biogenesis (**Response letter Fig. 1**).

Response letter Fig. 1

ATG14 seems to be required for the STX18 KO/KD induced suppression of LD content (Fig4 O and P), and a LIR competent version of ATG14 is needed to suppress LD levels.

Q16) It lacks essential data to directly support that STX18 completely bound to CCD of ATG14
Response: We thank reviewer for this critical comment. We found that STX18 binds the CCD of ATG14 (Revised Manuscript Fig. S5h). To confirm that STX18 disrupts the formation of PI3KC3-C1 complex by competitively binding the CCD in ATG14, we gradually increased the expression of ATG14 and found that the STX18 overexpression-inhibited Torin1-induced p62 degradation was reversed (Revised Manuscript Fig. S5i).

Comment: The binding of the CCD of ATG14 is now shown.

Response: This comment has been answered.

Note10) For the effect of Beclin on LDs function, the authors should include the ATG14 KD and both of ATG14 and Beclin KD to demonstrate the essence of ATG14 or Beclin.

Response: We are grateful for the Reviewer's helpful suggestions. We found that besides ATG14, STX18 regulates lipophagy also dependent on Beclin1 (Revised Manuscript Fig. S6a).

Comment: There is a clear dependence on Beclin in their HeLa cell model (FigS6a)

Response: This comment has been answered.

Q17) The Supplementary FigS4C IP is done with HA beads, not myc. Correct the labelling?

Response: We apologize for this error. This has now been corrected.

Comment: Corrected in what is now FigS5C

Response: This comment has been answered.

Q18) In Figure S4F, Beclin1 should be detected as a positive control since CCD domain is critical for the binding between ATG14 and Beclin1.

Response: We appreciate this critical advice. We have confirmed that CCD domain is critical for the binding between ATG14 and Beclin1 (Revised Manuscript Fig. S5f).

Comment: As stated, there is a very prominent band for the Beclin protein in their WT-ATG14 IP.

Response: This comment has been answered.

Q19) In Figure S4I, authors should also investigate if autophagic flux was influenced by STX18 in the ATG14 KO MEF cells.

Response: Thank you for your instructive and critical suggestions. We don't have Atg14 KO MEF cells, so we generated Atg14 KO HeLa cells and found that STX18 depletion failed to induce p62 degradation in Atg14 KO cells, indicating that STX18 negatively regulate autophagic flux dependent on ATG14 (Revised Manuscript Fig. S5m).

Comment: As stated, the authors did develop an ATG14-KO cell line that did seem to have compromised autophagic flux based on p62 turnover.

Response: This comment has been answered.

Fig5 seems to show that a LD localized anti-viral protein Viperin is degraded following these STX18 manipulations, which are potentially just an extension of the central findings that STX18 loss induces lipophagy. A potentially interesting connection is made with SARS-CoV-2 viral infection in Fig 6. They find that the M-protein of SARS-CoV-2, and viral infection itself, subverts the STX18/ATG14 interaction thereby promoting lipophagy and degradation of the anti-viral protein Viperin. In this model then, the lipophagy promoting activity of disrupting STX18/ATG14 seems to promote viral infection itself.

Q20) Could the authors show immunofluorescence images indicating the coincidence of LDs and Viperin with LC3 in the cells manipulated with ATG14 LIR mutant or the STX18 mutant (with ATG14 binding deficiency)?

Response: We appreciate this critical advice. Our new data indicated that rescued with wild-type STX18, but not STX18 Δ 71-80 (defect in ATG14 interaction) abolished the colocalization of LC3-Viperin-LDs in Stx18 KO cells (Revised Manuscript Fig. S7d). In addition, obvious colocalization between LC3-Viperin-LDs was observed in ATG14-Flag expressed, but not vector or ATG14LIRm-Flag expressed Atg14 KO cells (Revised Manuscript Fig. S7e).

Comment: The authors did provide examples of this co-localization.

Response: This comment has been answered.

Q22) Is STX18 (a.a.71-80) critical for the binding of M protein? Would it be helpful to define which domain located in the M protein is important for the binding of STX18 reciprocally?

Response: Thank you for your instructive and critical suggestions. Our co-IP data showed that STX18 Δ 1-80 still interacts with M protein, suggesting that STX18 (a.a.71-80) is not critical for the binding of M protein (Response letter Fig. 5).

As your suggested, to map the critical region of M protein necessary for its interaction with STX18, a series of truncated M mutants were constructed and used for co-IP assay. We found that M protein 20-100aa is required and sufficient for the interaction with STX18, and M Δ 20-100aa overexpression failed to induce lipophagy (Revised Manuscript Fig. S8b-d).

Comment: The authors did provide this data.

Response: This comment has been answered.

Q23) Can the authors provided a marker indicating VSV or SARS-CoV-2 infection?

Response: We thank reviewer for this helpful suggestion. We have used antibody to detect VSV M protein (Revised Manuscript Fig. 5b, f, n, o). In original manuscript, we had used SARS-CoV-2 N antibody to indicate virus infection (Original manuscript Fig. 6n, p, q).

Comment: Clear bands for the M protein can be seen in these lysates of Fig5.

Response: This comment has been answered.

Q24) In Figure 4O, Figure 6C, Figure 6E and Figure S3K, there was no control group?

Response: Many thanks for the critical comments and suggestions. For Original Fig 4o, we investigated the effect of STX18 KD on LD number in Atg14 KO cells rescued by vector, or wildtype ATG14, or ATG14LIRm. So, the control group is Atg14 KO cells transfected with negative siRNA. For control groups were wild-type cells transfected with negative siRNA or si-STX18, please refer revised Fig 4a, 4h, S6a, S6c.

For Original Fig6c, we apologize for this missing information. In Original Fig6c White ROIs and "+" indicate the cells expressing M protein and yellow ROIs indicate the cells without or low M expression as control group. We have added this information in figure legend.

For Original Fig6e, "+" indicate the cells expressing M protein, and "-" indicate cells without or low M expression as control group. We have added this information in figure legend.

For Fig Original S3k, this experiment was performed together with Original Fig3n. The data were supplemented due to the space limitation. Quantifications for Original Fig 3n and 3k were added (Revised Manuscript Fig 3o).

Response: We are grateful for the Reviewer's helpful suggestions. We have added a working model (Revised Manuscript Fig. 7).

Comment: Their model is a nice addition.

Response: This comment has been answered.

REVIEWERS' COMMENTS

Reviewer #1 (Remarks to the Author):

I have no further comments.

Reviewer #2 (Remarks to the Author):

A lot of work was performed, most substantive concerns were addressed in my opinion.